# The source of the Black Death in fourteenth-century central Eurasia

Maria A. Spyrou[1,2,3✉], Lyazzat Musralina[2,3,4,5], Guido A. Gnecchi Ruscone[2,3], Arthur Kocher[2,3,6], Pier-Giorgio Borbone[7], Valeri I. Khartanovich[8], Alexandra Buzhilova[9], Leyla Djansugurova[4], Kirsten I. Bos[2,3], Denise Kühnert[2,3,6,10], Wolfgang Haak[2,3], Philip Slavin[11✉] & Johannes Krause[2,3✉]

The origin of the medieval Black Death pandemic (AD 1346–1353) has been a topic of continuous investigation because of the pandemic's extensive demographic impact and long-lasting consequences[1,2]. Until now, the most debated archaeological evidence potentially associated with the pandemic's initiation derives from cemeteries located near Lake Issyk-Kul of modern-day Kyrgyzstan[1,3–9]. These sites are thought to have housed victims of a fourteenth-century epidemic as tombstone inscriptions directly dated to 1338–1339 state 'pestilence' as the cause of death for the buried individuals[9]. Here we report ancient DNA data from seven individuals exhumed from two of these cemeteries, Kara-Djigach and Burana. Our synthesis of archaeological, historical and ancient genomic data shows a clear involvement of the plague bacterium *Yersinia pestis* in this epidemic event. Two reconstructed ancient *Y. pestis* genomes represent a single strain and are identified as the most recent common ancestor of a major diversification commonly associated with the pandemic's emergence, here dated to the first half of the fourteenth century. Comparisons with present-day diversity from *Y. pestis* reservoirs in the extended Tian Shan region support a local emergence of the recovered ancient strain. Through multiple lines of evidence, our data support an early fourteenth-century source of the second plague pandemic in central Eurasia.

The Black Death, caused by the bacterium *Y. pestis*[10], was the initial wave of a nearly 500-year-long pandemic termed the second plague pandemic and is one of the largest infectious disease catastrophes in human history[1,11,12]. Estimated to have claimed the lives of up to 60% of the western Eurasian population over its eight-year course[1,12], the Black Death had a profound demographic and socioeconomic impact in all affected areas, with the European historical record being the most extensively studied resource until now[2,13–15].

Despite intense multidisciplinary research on this topic, the geographical source of the second plague pandemic remains unclear. Hypotheses based on historical records and modern genomic data have put forward a number of putative source locations ranging from western Eurasia to eastern Asia (Supplementary Information 1). In recent years, comparisons between ancient and modern *Y. pestis* genomes have shown the Black Death to be associated with a star-like emergence of four major lineages (branches 1, 2, 3 and 4)[16,17], the descendants of which are dispersed among rodent foci in Eurasia, Africa and the Americas. Although extant lineages that diverged before this event have been identified in central and eastern Eurasia[16,18,19], complementary ancient DNA (aDNA) data from such regions are lacking. Until now, analyses of

the historical record and ancient *Y. pestis* data have largely focused on the pandemic's progression in western Eurasia[12,17,20,21]. Although efforts to expand historical investigations and provide a wider spatiotemporal perspective are under way[9,11,22–26], the prevailing Eurocentric focus has hampered an identification of the origins of the Second Pandemic.

## A fourteenth-century epidemic in central Eurasia

To explore possible evidence associated with the early history of the second plague pandemic, we investigated the cemeteries of Kara-Djigach and Burana, located in the Chüy Valley near Lake Issyk-Kul of modern-day Kyrgyzstan. Excavations of these cemeteries between 1885 and 1892 revealed a unique archaeological assemblage potentially associated with an epidemic that affected the region during the fourteenth century (Fig. 1 and Supplementary Information 2). On the basis of tombstone inscriptions, these cemeteries showed a disproportionally high number of burials dating between 1338 and 1339, with some inscriptions stating that the cause of death was due to an unspecified pestilence[9,27] (Fig. 1, Extended Data Fig. 1, Supplementary Fig. 1, Supplementary Table 1 and Supplementary Information 2). Given the location,

[1]Institute for Archaeological Sciences, Eberhard Karls University of Tübingen, Tübingen, Germany. [2]Department of Archaeogenetics, Max Planck Institute for Evolutionary Anthropology, Leipzig, Germany. [3]Department of Archaeogenetics, Max Planck Institute for the Science of Human History, Jena, Germany. [4]Laboratory of Population Genetics, Institute of Genetics and Physiology, Almaty, Kazakhstan. [5]Kazakh National University by al-Farabi, Almaty, Kazakhstan. [6]Transmission, Infection, Diversification & Evolution Group, Max Planck Institute for the Science of Human History, Jena, Germany. [7]Department of Civilisations and Forms of Knowledge, University of Pisa, Pisa, Italy. [8]Department of Physical Anthropology, Kunstkamera, Peter the Great Museum of Anthropology and Ethnography, Russian Academy of Sciences, St Petersburg, Russian Federation. [9]Research Institute and Museum of Anthropology, Lomonosov Moscow State University, Moscow, Russian Federation. [10]European Virus Bioinformatics Center (EVBC), Jena, Germany. [11]Division of History, Heritage and Politics, University of Stirling, Stirling, UK. ✉e-mail: maria.spyrou@ifu.uni-tuebingen.de; philip.slavin@stir.ac.uk; krause@eva.mpg.de

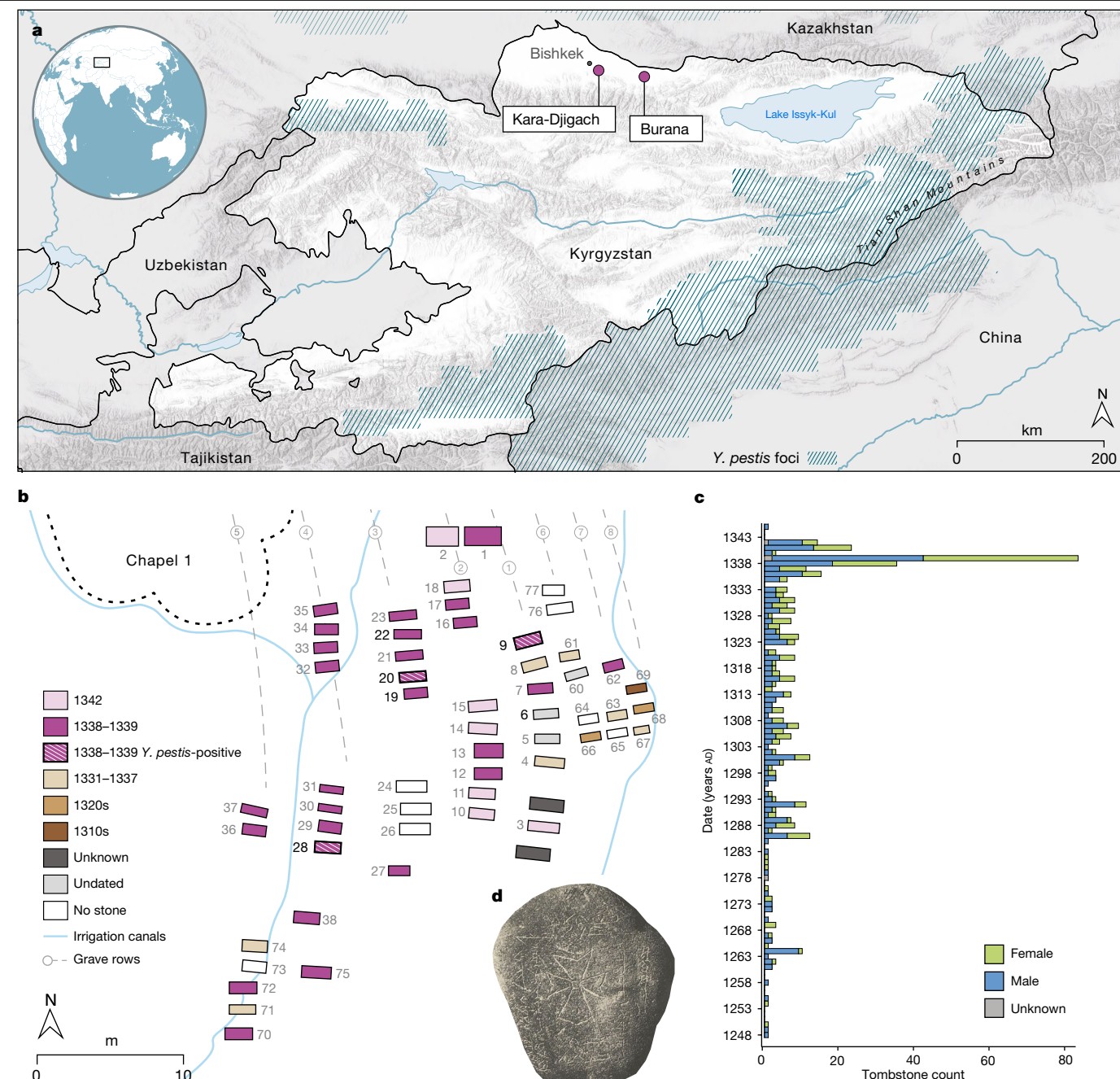

**Fig. 1 | Description of the investigated fourteenth-century Chüy Valley archaeological sites. a**, Locations of the Kara-Djigach and Burana archaeological sites in modern-day Kyrgyzstan. Regions encompassing *Y. pestis* foci at present are highlighted in blue (as in refs. [18,19]). The map was created using QGIS v.3.22.1 (ref. [51]) and uses Natural Earth vector map data from https://www. naturalearthdata.com/. **b**, Area within the Kara-Djigach cemetery, referred to as 'Chapel 1' with the highest concentration of excavated burials dating between 1338 and 1339. Burial dates were determined on the basis of their associated tombstones (Supplementary Information 2). The site map has been redrawn based on the original created by N. Pantusov in 1885. Individuals from graves 6, 9, 20, 22 and 28 (the numbers in bold) were investigated using aDNA in this study. Burials shown with stripe patterns were associated with individuals BSK001, BSK003 and BSK007, which showed evidence of *Y. pestis* infections. **c**, Annual numbers of tombstones from Kara-Djigach (*n* = 456) and Burana (*n* = 11) (Supplementary Table 1). Dataset updated from ref. [9] (see Supplementary Information 2 for details). **d**, Tombstone from the Kara-Djigach cemetery with legible pestilence-associated inscription. The inscription is translated as 'In the Year 1649 [=AD 1338], and it was the Year of the Tiger, in Turkic Bars. This is the tomb of the believer Sanmaq. [He] died of pestilence [=mawtānā]'. For a tracing of the inscription, see Extended Data Fig. 1.

timing and associated demographic pattern, early interpretations considered these characteristics as indicative of a plague epidemic[3,27] and have since triggered a long-lasting debate about the epidemic's association with the onset of the second plague pandemic[1,3–9,26] (Supplementary Information 2).

To better understand the contexts of Kara-Djigach and Burana, we translated and analysed surviving archival information from their excavations (Supplementary Information 2 and Supplementary Figs. 1–4). Furthermore, we generated human genomic data from 7 individuals (5 from Kara-Djigach and 2 from Burana) through a hybridization capture

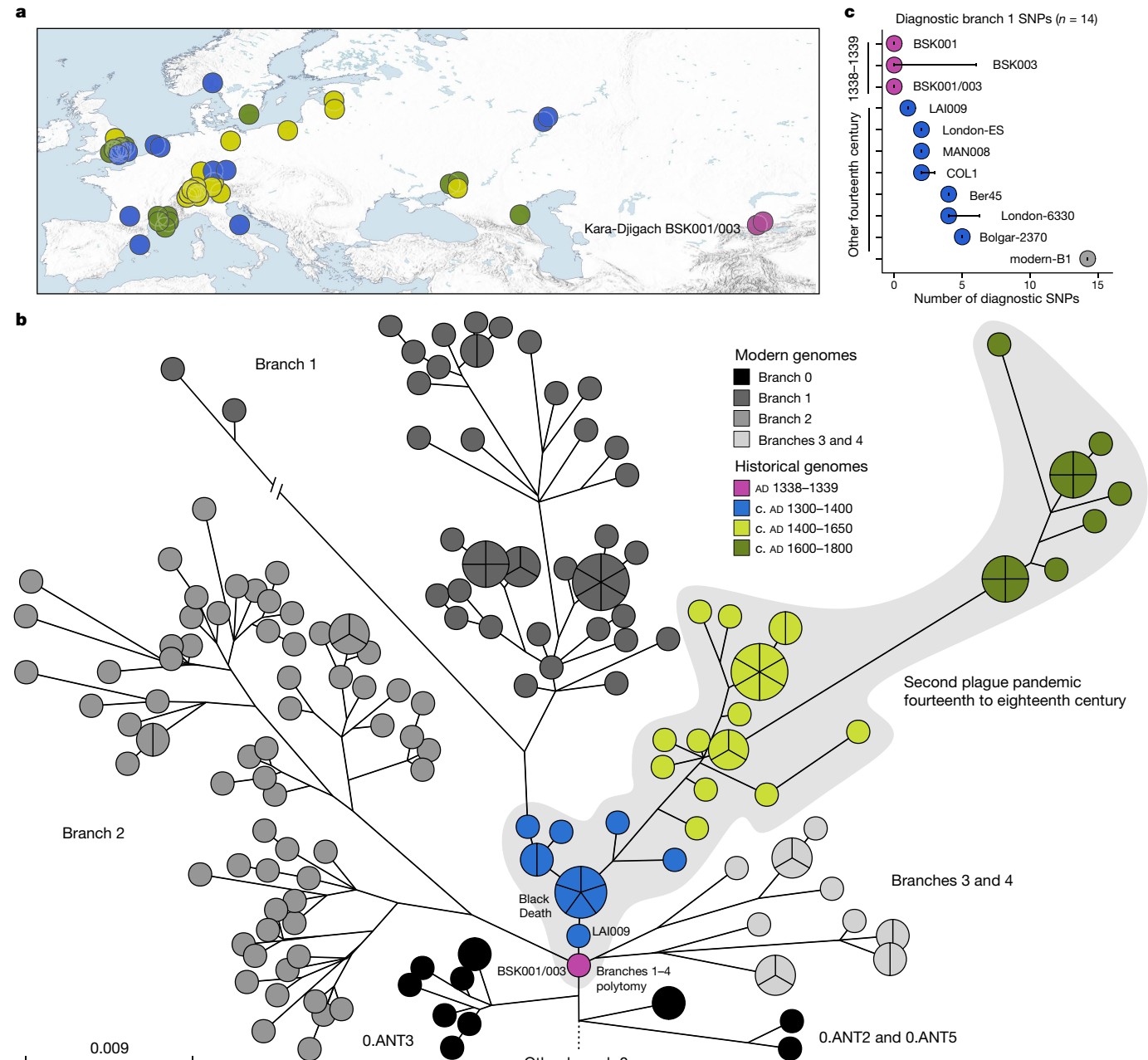

**Fig. 2 | Comparisons between BSK001/003 and published *Y. pestis* genomic diversity. a**, Map of all historical *Y. pestis* genomes used in the present study (*n* = 48). The colours represent different genome ages on a scale between 1300 and 1800, as depicted in **b**. The colour scale is maintained across all panels of this figure. To aid visibility in overlapping symbols, a jitter option was implemented for plotting genomes on the map. The map was created with QGIS v.3.22.1 (ref. [51]) and uses Natural Earth vector map data from https://www.naturalearthdata. com/. **b**, *Y. pestis* maximum likelihood phylogeny based on 2,960 SNPs, visualized using GrapeTree[50]. The depicted portion of the phylogeny contains the closest related lineages to BSK001/003. (For a fully labelled tree, see Extended Data Fig. 5). The colours of published historical strains are consistent with **a**. The scale denotes the number of substitutions per genomic site. **c**, Abundance of diagnostic SNP sharing in fourteenth-century *Y. pestis* genomes. The number of diagnostic SNPs (*n*) shared between all modern genomes on branch 1, and therefore defining this branch, were retrieved from a comparative SNP table of 203 modern *Y. pestis* genomes. SNP sharing was assessed by determining the allele status of each diagnostic position according to a threefold SNP calling threshold. The error bars denote the degree of missing data (*n*) in the respective ancient genome. Refer to Extended Data Fig. 6 and Supplementary Table 18 for an overview of diagnostic SNP sharing on different phylogenetic branches.

of approximately 1.24 million ancestry-informative single-nucleotide polymorphisms (SNPs)[28], which resulted in 4 individuals with sufficient genomic coverage for population genetic analyses (>30,000 SNPs). Using principal component analysis and ancestry modelling, we found these individuals to be falling broadly within the variability of ancient and present-day populations from central Eurasia. However, precise connections could not be determined given the scarcity of contemporaneous human genomic data from this region

(Supplementary Information 3, Supplementary Fig. 5 and Supplementary Tables 2–5). On the basis of the available tombstone inscriptions, burial artefacts, coin hoards and historical records, we found that the Chüy Valley housed ethnically diverse communities that relied on trade and maintained connections with several regions across Eurasia (Supplementary Information 2). Such links may have contributed to the spread of infectious diseases to and from this region during the fourteenth century.

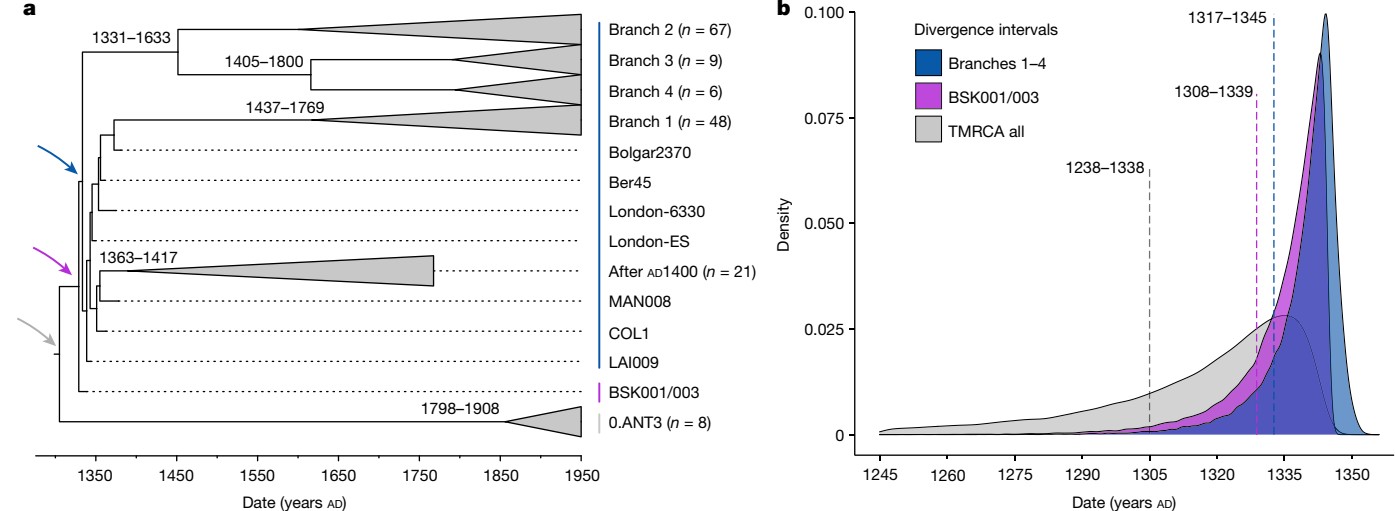

**Fig. 3 | Molecular dating of *Y. pestis* branches 1–4. a**, Maximum clade credibility time-calibrated phylogenetic tree. The tree is based on 167 genomes (historical and modern) and was estimated using the coalescent skyline tree prior and a log-normal relaxed clock. Collapsed branches contain modern and ancient isolates dating after AD 1400 (post-Black Death). The coloured arrows mark the nodes, for which equivalent posterior age distributions are shown in **b**. The estimated divergence dates (95% HPD intervals) of modern branches are shown on each corresponding node. **b**, Estimated posterior distributions based on the coalescent Bayesian skyline tree prior for the divergence of *Y. pestis* branches 1–4 (blue), for the estimated divergence of BSK001/003 (purple) and for the entire dataset used for this analysis (time to the most recent common ancestor of branches 1–4 and 0.ANT3, shown in grey). The dotted lines indicate mean posterior estimates and are annotated with the corresponding 95% HPD intervals.

## Ancient pathogen DNA screening

To investigate traces of ancient pathogen DNA that could explain the cause of the suspected epidemic, shotgun metagenomic data generated from all seven individuals were taxonomically classified using the HOPS pipeline[29] (Supplementary Table 6). Of those, three individuals exhumed from the Kara-Djigach cemetery (BSK001, BSK003 and BSK007) displayed potential evidence of ancient *Y. pestis* DNA (Supplementary Table 7) as well as low edit distances in reads mapping against the CO92 reference genome, and the presence of chemical alterations characteristic of aDNA (Supplementary Fig. 6 and Supplementary Table 8). As such, the respective DNA libraries were subjected to whole-genome *Y. pestis* capture (Methods).

## The ancestor of a fourteenth-century polytomy

Whole-genome *Y. pestis* capture yielded 6.7-fold and 2.8-fold average coverage for BSK001 and BSK003, respectively. Coverage across all three *Y. pestis* plasmids ranged from 24.7-fold to 4.7-fold (Supplementary Tables 9 and 10). For BSK007, genomic coverage was lower, approximately 0.13-fold, resulting from poorer aDNA preservation that was also reflected in the shotgun screening and human DNA enrichment data (Supplementary Tables 2, 3 and 8). Nevertheless, this sample was considered a true *Y. pestis*-positive because of the even distribution of mapping reads against the CO92 reference chromosome and the presence of aDNA-associated damage (Extended Data Figs. 2 and 3 and Supplementary Tables 9–11). Furthermore, a metagenomic classification of BSK007 reads aligning to the pCD1, pMT1 and pPCP1 plasmids identified >99% as *Y. pestis*-specific (Extended Data Fig. 3).

To evaluate whether the higher-coverage *Y. pestis* genomes BSK001 and BSK003 represented distinct bacterial strains, we compared their SNP profiles. To limit variant calls deriving from environmental contamination, particularly given the high amounts of multi-allelic sites identified in both genomes (Supplementary Fig. 7), we performed a taxonomy-informed metagenomic filtering using MALT (Methods and Supplementary Table 11). We identified 20 sites differing between BSK001 and BSK003, all of which are unique variants in the lower-coverage BSK003 (Supplementary Table 12). On the basis of previously defined authenticity criteria[30,31] (Methods), all such variants were consistent with residual exogenous contamination, suggesting that the two genomes were probably identical. Recovery of identical strains from both individuals is consistent with published evidence showing low diversity in *Y. pestis* genomes isolated from single epidemic contexts[10,17,20,21,32]. On the basis of their associated tombstones, BSK001, BSK003 and BSK007 were buried during the epidemic year 1338–1339 (Fig. 1 and Supplementary Information 2) and our data further support a *Y. pestis* involvement in this event.

We performed a comparative SNP analysis between the Kara-Djigach genomes and previously published historical and currently circulating *Y. pestis* diversity (Fig. 2a, Supplementary Tables 13–15). For this, BSK001 and BSK003 were combined (BSK001/003) to achieve an increased genomic resolution (combined coverage of 9.5-fold; Supplementary Table 9). Our analysis revealed one SNP unique to BSK001/003 when compared against 203 modern and 46 historical *Y. pestis* chromosomal genomes (Extended Data Fig. 4 and Supplementary Tables 16 and 17). This SNP was found in a region with persistent multi-allelic sites; therefore, it is considered artefactual[31] (Supplementary Fig. 8). Consistent with previous research on the evolutionary history of *Y. pestis*[16], our inferred maximum likelihood phylogeny exhibited five major branches, designated 0, 1, 2, 3 and 4, with published Second Pandemic genomes being associated with branch 1 (Fig. 2b). The placement of BSK001/003 is ancestral to all published fourteenth-century genomes from western Eurasia (Fig. 2b and Extended Data Fig. 5), separated by one SNP from LAI009, an isolate from the Volga region in eastern Europe[17], and by two SNPs from five genetically identical Black-Death-associated genomes from southern, central and northern Europe[17,21]. Specifically, BSK001/003 is positioned on a node previously designated N07 (ref. [16]), which preceded the multifurcation of branches 1–4. To evaluate whether missing data affected the accuracy of our phylogenetic placements, we investigated all BSK001 and BSK003 variant calls for shared positions with lineages deriving from the N07 node and those directly preceding it. BSK001/003 carries the ancestral state in all covered diagnostic SNPs defining branches 1–4 and 0.ANT3, which is the closest related branch 0 lineage to BSK001/003, as well as the derived state in all positions leading from 0.ANT3 to N07

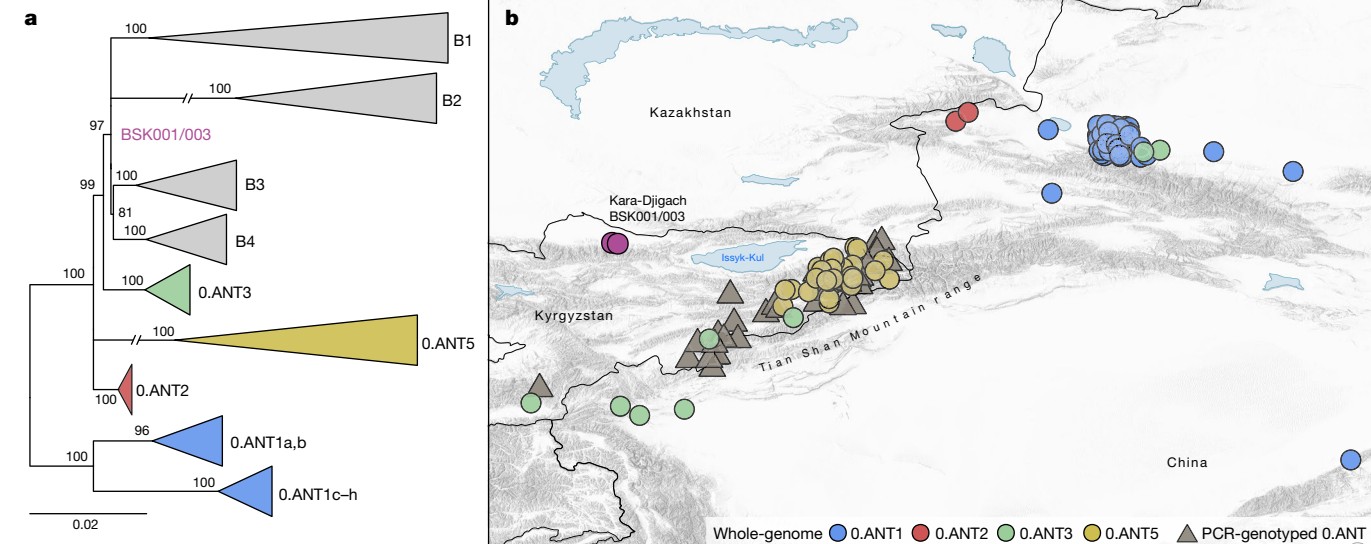

**Fig. 4 | Geographical isolation locations of modern 0.ANT lineages. a**, Maximum likelihood phylogenetic tree, based on 2,441 genome-wide variant positions. The tree was constructed to indicate the genetic relationships between available 0.ANT genomes depicted on the map and BSK001/003. Modern branches were collapsed to enhance tree clarity (see Extended Data Fig. 8 for a full tree). **b**, Map depicting the geographical isolation locations of 0.ANT strains (Supplementary Table 21), which belong to the closest ancestral branching lineages to the Kara-Djigach strain. The map includes both whole-genome data (further specified as 0.ANT lineages 1, 2, 3 and 5) and PCR-genotyped isolates that are broadly defined as 0.ANT, belonging to any of the 4 lineages. For strains in which exact geographical coordinates were unavailable, locations were approximated according to their associated plague reservoirs. To aid visibility in overlapping symbols, a jitter option was implemented for plotting objects on the map. The map was created with QGIS v.3.22.1 (ref. [51]) and uses Natural Earth vector map data from https://www.naturalearthdata.com/.

(Fig. 2c, Extended Data Fig. 6 and Supplementary Table 18). At our current resolution, we conclude that BSK001/003 represents the direct progenitor of the branch 1–4 polytomy.

## Divergence time for the branch 1–4 polytomy

The polytomy of branches 1–4 is a major event in the evolution of *Y. pestis* given its association with the Black Death[9,26,33] and the rich genetic diversity that emerged from it[16] (Fig. 2b). Estimates on the timing of this diversification have so far yielded wide ranges spanning from the tenth to the fourteenth centuries[16,34]. Recently, a narrower time frame was proposed that placed this emergence in the early thirteenth century, more than 100 years before the Black Death[22,26]. As BSK001/003 represents the common ancestor of branches 1–4, we used this genome from 1338 to 1339 to construct a time-calibrated phylogeny and re-estimate an age range for this diversification with BEAST2 (Supplementary Figs. 9 and 10 and Supplementary Table 19). After evaluating a number of demographic models (Supplementary Table 20), our resulting estimates based on the coalescent Bayesian skyline model revealed overlapping ages for the divergence of BSK001/003 (95% highest posterior density (HPD): 1308–1339), as well as for that of branch 1 from branches 2–4 (95% HPD: 1317–1345) (Fig. 3). As BEAST2 only infers bifurcating trees, we also used TreeTime[35] to infer a time-calibrated phylogeny that can retain polytomies. Consistent with our estimates above, we inferred a 1316–1340 date for the split time of branches 1–4 (Supplementary Fig. 11), although we caution that this method does not account for age uncertainties in ancient genomes. Taken together, the present results support an age range spanning the first half of the fourteenth century for the timing of the branch 1–4 polytomy.

Furthermore, to quantify the proportion of present-day *Y. pestis* genetic diversity that emerged from this polytomy, we computed mean pairwise distances (MPDs) and Faith's phylogenetic diversity (FPD) indices in 203 genomes comprising our entire modern dataset, as well as 130 genomes comprising branches 1–4 (Methods). In our dataset, 64% (130 out of 203) of modern *Y. pestis* strains belonged to branches 1–4, reflecting the high worldwide frequency known for these lineages[16,36,37]. We estimate that branches 1–4 represent approximately 40% of the overall phylogenetic diversity within present-day *Y. pestis* based on our full dataset (MPD ratio: 41%; 95% percentile interval (PI): 35.3–46.4; FPD ratio: 35.9%; 95% PI: 31.6–39.5). This value is marginally reduced after equalizing the number of genomes in branches 1–4 and branch 0 (MPD ratio: 36.8%; 95% PI: 32.0–41.9; FPD ratio: 33.9%; 95% PI: 29.4–37.7) (Extended Data Fig. 7). Given that the known history of *Y. pestis* reaches back at least 5,000 years[38], it is notable that a substantial fraction of its surviving genetic diversity accumulated since the fourteenth century.

## Plague reservoirs in the Tian Shan area

To address existing hypotheses on the Black Death's geographical origins (Supplementary Information 1), we investigated the possibility of a local emergence versus an introduction of the BSK001/003 strain into the Chüy Valley from a different area. For this, we assessed the geographical distribution of the most closely related ancestral branching lineages to BSK001/003 and identified 164 present-day 0.ANT strains with record of their isolation locations (Supplementary Table 21). Consistent with previous interpretations[9,18,26], we found that all such strains were retrieved from foci in eastern Kazakhstan, eastern Kyrgyzstan and the Xinjiang Uygur Autonomous Region of northwestern China (Fig. 4 and Extended Data Fig. 8). Although we cannot exclude a different geographical range for these lineages in the past, the current data are consistent with a local emergence of BSK001/003 within the extended Tian Shan region. Intriguingly, the oldest recovered genome associated with 0.ANT was also identified in the Tian Shan region (third century AD)[39] and forms part of an extinct clade that caused the first plague pandemic (sixth to eighth centuries AD)[30]. As noted previously[18,26,33,40], most extant 0.ANT strains have been isolated from marmots and their ectoparasites known to be the primary *Y. pestis* reservoirs in these areas (Supplementary Table 21). Therefore, such species could represent possible candidates for the spillover that led to the second plague pandemic.

## Discussion

The power of ancient metagenomics lies in its potential to provide direct evidence for testing long-standing historical hypotheses and reveal phylogeographical patterns of microbial diversity through time[41]. One such debate concerns the events that triggered the second plague pandemic, as well as the time and place of its emergence. Recently, an analysis of historical, genetic and ecological data led to the suggestion that the emergence of *Y. pestis* branches 1–4 occurred more than a century before the beginning of the Black Death. According to the proposed model, this initial diversification was mediated by people and was linked with territorial expansions of the Mongol Empire across Eurasia during the early thirteenth century[22,26,42]. By contrast, we present ancient *Y. pestis* data from central Eurasia that support a fourteenth-century emergence; therefore, earlier outbreak attributions remain to be explored. At present, the narrow-focused sampling chosen for this study does not allow for an assessment of the spread of the BSK001/003 strain. Previous studies have shown that *Y. pestis* can disseminate rapidly without accumulation of genetic diversity[17,21], thus potentiating a contemporaneous presence of the same strain across a large geographical range. Nevertheless, the known range of extant plague foci associated with lineages ancestral to BSK001/003 provide support for its emergence in central Eurasia and possibly in the extended Tian Shan region. Although the dynamics that triggered the bacterium's emergence in this region are unknown, previous studies showed that environmental factors, such as natural disasters and sudden changes in temperature and precipitation can have an impact on *Y. pestis* host ecologies and, as a result, can trigger outbreaks in human populations[43–46]. Although we have no evidence to infer such connections with the Kara-Djigach epidemic, we envision that our precise 1338–1339 date will serve as a reference point for future environmental, archaeological and historical research focusing on the events that caused a *Y. pestis* introduction into human populations and precipitated the second plague pandemic.

The onset of the Black Death has been conventionally associated with outbreaks that occurred around the Black Sea region in 1346 (refs. [1,47]), eight years after the Kara-Djigach epidemic. At present, the exact means through which *Y. pestis* reached western Eurasia are unknown, primarily due to large pre-existing uncertainties around the historical and ecological contexts of this process. Previous research suggested that both warfare and/or trade networks were some of the main contributors in the spread of *Y. pestis*[21,22,26,47,48]. Yet, related studies have so far either focused on military expeditions that were arguably unrelated to initial outbreaks[47] or others that occurred long before the mid-fourteenth century[22,26]. Moreover, even though preliminary analyses exist to support an involvement of Eurasian-wide trade routes in the spread of the disease[48], their systematic exploration has so far been conducted only for restricted areas of western Eurasia[21,47]. The placement of the Kara-Djigach settlement in proximity to trans-Asian networks[9,49], as well as the diverse toponymic evidence and artefacts identified at the site (Supplementary Information 2) lend support to scenarios implicating trade in *Y. pestis* dissemination. Therefore, an investigation of early-to-mid-fourteenth-century connections across Asia, interpreted alongside genomic evidence, will be important for disentangling the bacterium's westward dispersals.

Past and present experiences have demonstrated that reconciling the source of a pandemic is a complex task that cannot be accomplished by a single research discipline. Although the ancient *Y. pestis* genomes reported in this Article offer biological evidence to settle an old debate, it is the unique historical and archaeological contexts that define our study's scope and importance. As such, we envision that future synergies will continue to reveal important insights for a detailed reconstruction of the processes that triggered the second plague pandemic.

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

## Methods

### Sampling, DNA extraction, partial uracil DNA glycosylase library preparation and sequencing

We obtained permission from the Kunstkamera, Peter the Great Museum of Anthropology and Ethnography in St Petersburg for the sampling and ancient DNA analysis of 7 tooth specimens, excavated between 1885 and 1892 from the medieval cemeteries of Kara-Djigach and Burana (Supplementary Information 2). No statistical methods were used to predetermine the number of samples used in this study. All laboratory procedures were carried out in the dedicated aDNA facilities of the Max Planck Institute for the Science of Human History and the Max Planck Institute for Evolutionary Anthropology. The detailed procedures used for tooth sampling can be found in ref. [52]. In brief, teeth were sectioned in the dentin–enamel junction using an electric saw with a diamond blade. After tooth sectioning, approximately 50 mg of powder was removed from the surface of the pulp chamber of each tooth using rounded dental drill bits.

The recovered tooth powder was used for DNA extractions using a previously established protocol optimized for the recovery of short fragments of DNA[53]. The exact steps and modifications of the procedure used have been made available in ref. [54]. In brief, the tooth powder was incubated overnight (12–16 h) at 37 °C in 1 ml of DNA lysis buffer containing EDTA (0.45 M, pH 8.0) and proteinase K (0.25 mg ml$^{-1}$). After incubation, DNA binding and isolation was performed using a custom GuHCl-based binding buffer and purification using High Pure Viral Nucleic Acid Large Volume Kit (Roche). Finally, DNA was eluted in 100 µl of Tris-EDTA-Tween containing Tris-HCl (10 mM), EDTA (1 mM, pH 8.0) and Tween-20 (0.05%). For procedure monitoring, extraction blanks and positive extraction controls were included throughout the laboratory processing steps.

All DNA extracts were converted into one-to-two double-stranded DNA libraries for Illumina sequencing, using 25 µl of input extract per library with an initial partial uracil DNA glycosylase (UDG) and endonuclease VIII treatment (USER enzyme; New England Biolabs) according to established protocols[55,56]. The detailed library preparation procedure, including the blunt-end repair, adaptor ligation and adaptor fill-in reaction steps can be found in ref. [57]. After library preparation, each library was quantified using a quantitative PCR system (LightCycler 96 Instrument) using the IS7 and IS8 primers[55]. For multiplex sequencing, we performed double indexing of all libraries using previously published procedures[58], outlined in detail in ref. [59]. A combination of unique index primers containing 8 base pair (bp) identifiers were assigned to each library. To aid amplification efficiency, libraries were then split into multiple PCR reactions for the indexing step based on their initial IS7/IS8 quantification. The number of indexing PCR reactions performed for each library was determined so that every reaction was assigned an input of no more than $1.5 \times 10^{10}$ DNA copies. Each reaction was set up using the Pfu Turbo Cx Hotstart DNA Polymerase (Agilent Technologies) and was run for 10 cycles using the following conditions: initial denaturation at 95 °C for 2 min followed by a cycling of 95 °C for 30 s, 58 °C for 30 s and 72 °C for 1 min, as well as a final elongation step at 72 °C for 10 min. All PCR products were purified using the MinElute DNA Purification Kit (QIAGEN), with some modifications to the manufacturer's protocol[59]. Finally, all indexing PCR products were qPCR-quantified (LightCycler 96 Instrument) using the IS5 and IS6 primer combination[58,59]. To avoid heteroduplex formation, indexed libraries were amplified to $10^{13}$ DNA copies per reaction with the Herculase II Fusion DNA Polymerase (Agilent Technologies) and quantified using a 4200 Agilent TapeStation Instrument using a D1000 ScreenTape system (Agilent Technologies). Libraries were diluted to 10 nM and pooled equimolarly for sequencing. We performed shotgun DNA sequencing on an Illumina HiSeq 4000 platform using a 76-cycle kit ($1 \times 76 + 8 + 8$ cycles).

### Shotgun next-generation sequencing read processing and metagenomic screening

After demultiplexing, raw shotgun sequenced reads were preprocessed in the EAGER pipeline v.1.92.58 using AdapterRemoval v.2.2.0 (ref. [60]), which was used to remove Illumina adaptors (minimum overlap of 1 bp), as well as for read filtering according to sequencing quality (minimum base quality of 20) and length (retaining reads ≥30 bp). Subsequently, all datasets were screened for the presence of pathogen DNA traces using the metagenomic pipeline HOPS[29]. First, preprocessed reads were aligned against a custom RefSeq database[61] (November 2017) containing all complete bacterial and viral genome assemblies, a subset of eukaryotic pathogen assemblies and the *GRCh38* human reference genome. Genome assemblies that contained the word 'unknown' were removed from the database, retaining a total of 15,361 entries. The database retained a number of *Yersinia* species entries: *Yersinia aldovae* (*n* = 1), *Yersinia aleksiciae* (*n* = 1), *Yersinia enterocolitica* (*n* = 16), *Yersinia entomophaga* (*n* = 1), *Yersinia frederiksenii* (*n* = 3), *Yersinia intermedia* (*n* = 1), *Yersinia kristensenii* (*n* = 2), *Y. pestis* (*n* = 39), *Yersinia phage* (*n* = 17), *Yersinia pseudotuberculosis* (*n* = 13), *Yersinia rohdei* (*n* = 1), *Yersinia ruckeri* (*n* = 4), *Yersinia similis* (*n* = 1) and *Yersinia* sp. FDA-ARGOS (*n* = 1). MALT v0.4[62] was run using the following parameters: -id 90 -lcaID 90 -m BlastN -at SemiGlobal -topMalt 1 -sup 1 -mq 100 -verboseMalt 1 -memoryMode load -additionalMaltParameters. The resulting alignment files were post-processed with MALTExtract for a qualitative assessment against a predefined list of 356 target taxonomic entries (https://github.com/rhuebler/HOPS/blob/external/Resources/default_list.txt). Specifically, reads were assessed according to their edit distance against a specific pathogen sequence in the database and the potential occurrence of mismatches that could signify the presence of aDNA damage[29]. In cases in which both parameters were met, the corresponding pathogen alignment was considered a strong candidate. Preprocessed reads were mapped against the *Y. pestis* CO92 (NC_003143.1) and human (*hg19*) reference genomes with the Burrows–Wheeler Aligner (BWA). Mapping parameters were set to 0.01 for the edit distance (-n) and seed length was disabled (-l 9999). Subsequently, we used SAMtools v.1.3 (ref. [63]) to remove reads with mapping quality lower than 37 (for CO92) or 30 (for *hg19*); PCR duplicates were removed with MarkDuplicates v1.140 (http://broadinstitute.github.io/picard/). Finally, patterns of aDNA damage were assessed with mapDamage v.2.0 (ref. [64]).

### Single-stranded DNA library preparation and hybridization capture

For specimens BSK001 and BSK003, extra single-stranded DNA libraries were constructed from an input DNA extract of 30 µl. We performed library preparation at the Max Planck Institute for Evolutionary Anthropology using an automated protocol that is publicly available[65]. Single-stranded and double-stranded libraries from individuals BSK001, BSK003 and BSK007 were enriched using DNA probes covering the whole *Y. pestis* genome, as well as 1.24 million genome-wide SNP sites of the human genome[66,67]. For capture preparation, all libraries were amplified for the necessary number of PCR cycles to achieve 1–2 µg of input DNA. PCR reactions were carried out using the Herculase II Fusion DNA Polymerase. They were then purified using the MinElute DNA Purification Kit and eluted in EB elution buffer containing 0.05% Tween 20. Finally, library concentrations (ng µl$^{-1}$) were quantified using a NanoDrop spectrophotometer (Thermo Fisher Scientific). For the in-solution *Y. pestis* captures, the probe set design was based on a set of publicly available modern genomes, specifically the *Y. pestis* CO92 chromosome (NC_003143.1), CO92 plasmid pMT1 (NC_003134.1), CO92 plasmid pCD1 (NC_003131.1), KIM10 chromosome (NC_004088.1), Pestoides F chromosome (NC_009381.1) and the *Y. pseudotuberculosis* IP32953 chromosome (NC_006155.1). For the in-solution human DNA captures, the probe set design was created to target 1,237,207 variants

across the genome that are informative for studying the genetic history of worldwide human populations[28,67]. Both human DNA and *Y. pestis* hybridization captures were carried out for two rounds as described previously[28,69,68,67,66], in which partially UDG-treated libraries from the same individual were pooled in equimolar ratios for capture and single-stranded libraries were captured separately.

## Post-capture *Y. pestis* data processing

After *Y. pestis* whole-genome capture, libraries were sequenced on a HiSeq 4000 platform (1 × 76 + 8 + 8 cycles or 2 × 76 + 8 + 8 cycles) at a depth of approximately 11–27 million raw reads. The preprocessing of raw demultiplexed reads was carried out as described in the 'Shotgun next-generation sequencing read processing and metagenomic screening' section. At this stage, the datasets produced from partially UDG-treated libraries from the same individual were pooled and terminal bases were trimmed using fastx_trimmer (FASTX Toolkit 0.0.14, http://hannonlab.cshl.edu/fastx_toolkit/) to avoid damaged site interference with SNP calling during further processing. The following steps for read mapping, PCR duplicate removal and aDNA damage calculation were carried out in the EAGER pipeline[70]. We performed read mapping with BWA v.0.7.12 against the *Y. pestis* CO92 reference genome (NC_003143.1). For the pooled and trimmed partial UDG-treated libraries, BWA parameters were set to 0.1 for the edit distance (-n) and seed length was disabled (-l 9999). Given that the single-stranded libraries constructed for this study retained aDNA-associated damage, the BWA parameters were set to 0.01 for the edit distance (-n) to allow for an increased number of mismatches that could derive from deamination; seed length was disabled (-l 9999). We performed read mapping against the plasmids using the same parameters against a concatenated reference of all three *Y. pestis* plasmids (pMT1: NC_003134.1; pPCP1: NC_003132.1; and pCD1: NC_003131.1), masking the problematic pPCP1 region between nucleotides 3000 and 4200 that was shown to have high similarity to expression vectors used in laboratory reagents[71]. SAMtools v.1.3 (ref. [63]) was used to remove all reads with mapping quality lower than 37 (-q), whereas MarkDuplicates was used to remove PCR duplicates. Deamination patterns associated with aDNA damage were retrieved with mapDamage v.2.0 (ref. [64]). We used MALT[62] for a taxonomic classification of mapped reads, to attempt a retention of reads that are more likely to be endogenous *Y. pestis*. MALT was run against the same database as described in the section 'Shotgun next-generation sequencing read processing and metagenomic screening', using the following parameters: -m BlastN -at SemiGlobal -top 1 -sup 1 -mq 100 -memoryMode load -ssc -sps. The minimum percentage identity parameter was set to default (-id 0.0), as opposed to a 90% identity filter used for running HOPS[29], to avoid any reference bias that might arise from the removal of endogenous reads with a higher number of mismatches. After run completion, to retain the maximum number of reads accounting for the naive lowest common ancestor algorithm, we extracted reads that were assigned to the *Yersinia* genus node or summarized under the *Y. pseudotuberculosis* complex node. Reads were extracted in FASTA format from MEGAN v.6.4.12 (ref. [72]). Subsequently, FASTA files were converted into FASTQ format with the reformat.sh script in BBMap from the BBtools suite (version 38.86, https://sourceforge.net/projects/bbmap/). FASTQ files were then remapped against the CO92 reference genome using the same parameters as described previously in this section. For single-stranded libraries, mapDamage v.2.0 (ref. [64]) was used to rescale quality scores in read positions at which potential deamination-associated mismatches to the reference were identified. Subsequently, BAM files corresponding to the same individual were concatenated after mapping quality filtering and PCR duplicate removal. We performed concatenation using the SAMtools 'merge' command and with the AddOrReplaceReadGroups tool in Picard (http://broadinstitute.github.io/picard/) for assigning a single read group to all reads in each new file.

## SNP calling, heterozygosity estimates and SNP filtering

Variant calling was carried out for BSK001 and BSK003, both before and after MALT[62] filtering using the UnifiedGenotyper in the Genome Analysis Toolkit (GATK) v.3.5 (ref. [73]). GATK was run using the EMIT_ALL_SITES option, which produced a call for every position on the chromosomal CO92 reference genome. The resulting genomic profiles of BSK001 and BSK003 were compared against a set of 233 modern and 46 historical *Y. pestis* genomes, as well as against the *Y. pseudotuberculosis* reference genome IP32953 (NC_006155.1), using the Java tool MultiVCFAnalyzer v.0.85 (https://github.com/alexherbig/MultiVCFAnalyzer). MultiVCFAnalyzer v.0.85 was run with the following parameters. SNPs were called at a minimum coverage of threefold and in cases of heterozygous positions, calls were made at a 90% minimum support threshold. In addition, SNPs were called at a minimum genotyping quality of 30. Furthermore, previously defined non-core and repetitive regions, as well as regions containing homoplasies, ribosomal RNAs, transfer-messenger RNAs and transfer RNAs were excluded from comparative SNP calling[16,32]. A set of 6,567 total variant sites were identified in the present dataset.

To investigate the extent of possible exogenous contamination within the BSK001 and BSK003 datasets, we estimated the number of ambiguous heterozygous variants beyond the SNP calling threshold. For this, MultiVCFAnalyzer v.0.85 (ref. [74]) was used to generate an SNP table of alternative allele frequencies ranging between 10 and 90%. The results were then used to create 'heterozygosity' histogram plots of the estimated frequencies in R v.3.6.1 (ref. [75]). Heterozygosity plots were created both before and after MALT filtering (see 'Post-capture *Y. pestis* data processing') to investigate whether taxonomy-informed filtering could aid the elimination of contaminant sequences in the investigated datasets (Supplementary Fig. 7).

An SNP table created with MultiVCFAnalyzer v.0.85, containing all variant positions across the present dataset, was filtered to identify SNP differences between the BSK001 and BSK003 genomes. The identified differences (*n* = 20) were then evaluated with the Java tool SNP_Evaluation[30] (build date 13 August 2018; https://github.com/andreasKroepelin/SNP_Evaluation). The variant table and the VCF files of each genome were used as input for SNP_Evaluation. Furthermore, each identified private variant was evaluated within a 50 bp window and was considered 'true' when fulfilling the following criteria established in studies published previously[17,21,30,76]: (1) no multi-allelic sites were permitted within the evaluated window unless they were consistent with aDNA deamination (signified as spurious C-to-T or G-to-A substitutions); (2) the evaluated SNP position itself was not consistent with aDNA damage (no bases overlapping the SNP were downscaled by mapDamage v.2.0 (ref. [64])); (3) no gaps in genomic coverage were identified in the evaluated window; (4) reads overlapping the SNP sites showed specificity to the *Y. pseudotuberculosis* complex when screened with BLASTn (https://blast.ncbi.nlm.nih.gov/Blast.cgi).

Finally, to gain phylogenetic resolution, the BSK001 and BSK003 *Y. pestis* datasets were concatenated. We performed concatenation of BAM files, MALT[62] filtering and aDNA damage rescaling (with mapDamage v.2.0 (ref. [64])) as described in the section 'Post-capture *Y. pestis* data processing'. Moreover, the dataset was included in the comparative SNP analysis using MultiVCFAnalyzer v.0.85 (ref. [74]) as described above. Finally, unique SNPs were evaluated with SNP_Evaluation[30] according to the four criteria listed above.

## Phylogenetic reconstruction and diversity estimations

Phylogenetic analysis was used to explore 233 *Y. pestis* genomes as part of the modern comparative dataset. An SNP alignment produced by MultiVCFAnalyzer v.0.85 (ref. [74]) was used to construct a phylogenetic tree in MEGA7, using the maximum parsimony approach with 95% partial deletion (6,032 SNPs). Of the 233 modern *Y. pestis* genomes in the current dataset, 30 displayed extensive private branch lengths (Supplementary Fig. 12). Such an effect in bacterial

phylogenies could result either from true biological diversity or from technical artefacts associated with false SNP incorporation during computational genome reconstruction. Although we cannot exclude the presence of several strains with exceedingly higher mutation rates in the current dataset, previous studies showed that modern *Y. pestis* strains with 'mutator' profiles are uncommon[16,36]. In this study, 27 out of 30 genomes that showed disparities in their private SNP counts compared to the rest of the dataset, were derived from assemblies for which the quality of SNP calls could not be evaluated (raw data unavailable). Because potential mis-assemblies or false-positive SNP calls can affect evolutionary inferences and diversity estimations, these genomes were excluded from further analyses. Therefore, we performed phylogenetic analysis using a subset of 203 modern *Y. pestis* genomes (Supplementary Table 13). The list of excluded genomes is as follows: 2.MED1_139 (ref. [19]), 2.MED1_A-1809 (ref. [18]), 2.MED1_A-1825 (ref. [19]), 2.MED1_A-1920 (ref. [19]), 2.MED0_C-627 (ref. [19]), 2.MED1_M-1484 (ref. [19]), 2.MED1_M-519 (ref. [19]), 0.ANT5_A-1691 (ref. [18]), 0.ANT5_A-1836 (ref. [18]), 0.PE2_C-678 (ref. [77]), 0.PE2_C-370 (ref. [77]), 0.PE2_C-700 (ref. [77]), 0.PE2_C-746 (ref. [77]), 0.PE2_C-535 (ref. [77]), 0.PE2_C-824 (ref. [77]), 0.PE2_C-712 (ref. [77]), 0.PE2b_G8786 (ref. [16]), 0.PE4_I-3446 (ref. [78]), 0.PE4_I-3517 (ref. [78]), 0.PE4t_A-1815 (ref. [18]), 0.PE4_I-3447 (ref. [78]), 0.PE4_I-3518 (ref. [78]), 0.PE4_I-3443 (ref. [78]), 0.PE4_I-3442 (ref. [78]), 0.PE4_I-3519 (ref. [78]), 0.PE4_I-3516 (ref. [78]), 0.PE4_I-3515 (ref. [78]), 0.PE4_Microtus91001 (ref. [79]), 0.PE5_I-2238 (ref. [80]) and 0.PE7b_620024 (ref. [16]).

A genome-wide SNP alignment consisting of 203 modern-day and 48 historical *Y. pestis* genomes (Supplementary Table 14), as well as the *Y. pseudotuberculosis* IP32953 genome, was used as input to construct a maximum likelihood phylogeny including 2,960 SNPs and up to 4% missing data. We performed phylogenetic analysis with RAxML[81] v.8.2.9 using the generalized time-reversible (GTR) substitution model with 4 gamma rate categories. Finally, 1,000 bootstrap replicates were used to estimate node support for the resulting tree topology. After run completion, the maximum likelihood phylogenies were visualized with FigTree v.1.4.4 (http://tree.bio.ed.ac.uk/software/figtree/) and GrapeTree (v1.5.0)[50].

To estimate the proportion of modern *Y. pestis* diversity descending from BSK001/003, we used the R package picante v1.8.2[82] to compute the MPD and FPD[83] from the reconstructed maximum likelihood substitution tree. Measures made on a subset of the tree corresponding to the subclade descending from BSK001/003 (branches 1–4) were compared to that of the complete *Y. pestis* phylogeny. In both cases, only modern strains were included in the calculation. We used a bootstrapping approach to assess the sensitivity of our results with regard to sampling and phylogenetic uncertainty[84]. For each of the 1,000 RAxML bootstrap trees, we randomly resampled modern strains with replacement and only kept branches of the tree corresponding to the sampled strains. Diversity measures were performed for each of the obtained resampled bootstrap trees, from which median estimates and 95% percentile intervals were derived.

To assess the potential impact of uneven sampling among branches (branches 1–4 contained 130 modern strains whereas branch 0 contained only 73), we repeated the same analysis but adding an initial step intended to equalize the number of genomes in both parts of the tree. We subsampled branches 1–4 to the same number of strains as in branch 0 using sequence clustering in branches 1–4 to obtain representative subsamples. We performed hierarchical clustering based on pairwise phylogenetic distances (derived from the maximum likelihood phylogenetic tree) and the resulting tree was cut to define 73 clusters (functions hclust[85] and cutree in R v.4.0.3). For each bootstrap tree, clusters were randomly downsampled to one strain, resulting in an equal number of strains between branch 1–4 and branch 0. Resampling with replacement was then applied as previously to each of the downsampled trees before computing diversity measures.

## Plasmid SNP analysis

To investigate possible genetic variation among the plasmids of historical genomes, we performed read mapping of BSK001, BSK003 and BSK001/003 with BWA as well as SNP calling with GATK v.3.5 as described in the above section 'SNP calling, heterozygosity estimates and SNP filtering' against each of the three *Y. pestis* plasmids (pMT1: NC_003134.1; pPCP1: NC_003132.1; and pCD1: NC_003131.1). We then performed comparative SNP calling using MultiVCFAnalyzer v0.85 (ref. [74]) against a set of 46 historical *Y. pestis* genomes as well as the modern reference strains CO92, KIM5 and 0.PE4-Microtus91001. Variants were filtered in individual genomes using SNP_Evaluation according to previously defined criteria (see the 'SNP calling, heterozygosity estimates and SNP filtering' section). In the present dataset, we identified ten variants in pCD1, eight in pMT1 and two in pPCP1 (Supplementary Table 15).

## Time-calibrated phylogenetic analysis

To estimate the timing for the divergence of *Y. pestis* branches 1–4 using the BSK001/003 genomes as a new calibration point, we used a dataset comprising all modern genomes from branches 1–4 used for phylogenetic analysis (*n* = 130), genomes of the ancestral branching lineage 0.ANT3 (*n* = 8) and all 29 historical (fourteenth–eighteenth century) genomes in our dataset representing unique genotypes. In cases of identical genomes, the highest coverage genome was chosen for this analysis. We applied a molecular clock test using a maximum likelihood method in MEGA7 (ref. [86]), using a GTR substitution model in which differences in evolutionary rates among sites were estimated using a discrete gamma distribution with four rate categories. On the basis of this molecular clock test, the null hypothesis of equal evolutionary rates across tested phylogenetic branches was rejected, which is consistent with previous studies showing substitution rate variation across *Y. pestis* lineages[16,17]. Therefore, a log-normal relaxed clock model was used for all subsequent molecular dating analyses.

For the molecular dating analysis, we used the Bayesian statistical framework BEAST2 v.6.6 (ref. [87]). The ages of all ancient isolates were used as calibration points to construct a time-calibrated phylogeny with their radiocarbon or archaeological context age ranges set as uniform priors (see Supplementary Table 19 for all used age ranges). The ages of all modern isolates were set to 0 years before the present. We tested a number of coalescent tree priors such as the coalescent constant size, Bayesian skyline[88] and exponential population models, all of which have been used or tested in previous ancient pathogen genomic studies[17,89,90,91]. We also tested the birth–death skyline tree prior, which has gained traction in recent years[91,92,93] because it can account for epidemiological variables and can model sampling disparities through time[94]. Moreover, we used jModelTest v.2.1.10 (ref. [95]) to identify the substitution model of best fit for our dataset. The indicated transversion model was implemented in BEAUti by using a GTR model (4 gamma rate categories) and the AG substitution rate parameter fixed to 1.0 (as indicated previously[93]). All tree priors were used in combination with a log-normal relaxed clock rate with a uniform prior distribution ranging between $1 \times 10^{-3}$ and $1 \times 10^{-6}$ substituions per site per year for the SNP alignment (1,405 sites after a 95% partial deletion), corresponding to a range of $3 \times 10^{-7}$ to $3 \times 10^{-10}$ across the entire genome, which is within the range of previous estimates[17]. As part of the phylogenetic topology set-up, all branch 1–4 genomes (ancient plus modern) as well as the 0.ANT3 lineage were constrained to be independent monophyletic clades. For the constant population size and exponential population tree priors, all other parameters were set to default. For the coalescent skyline tree prior, a Jeffreys prior distribution ($1/x$) was used for the population sizes and a dimension of 5 was used to permit variations in the group and population sizes through time, with an upper bound of 380,000 for the effective population size (default). Moreover, for the birth–death skyline tree prior, we used a uniform prior for the rate to become non-infectious that ranged between 0.03 and 70, to account

for possible infectious periods ranging from 30 years (lifelong infections in rodent reservoirs[96,97]) to 5 days (average infectious period for bubonic plague[98]). We used a prior beta distribution with mean = 0.1 (alpha = 10.0, beta = 90.0) for the sampling probability $\rho$ at time 0 and a uniform distribution ranging between 0 and 0.1 for the sampling proportion $s$. For the latter, two shifts were allowed through time. Finally, the reproductive number $R$ was allowed to vary between 0 and 4.0 using a long normal prior distribution of median = 1.0 and s.d. = 0.7, which is within the range of previous estimates for bubonic and pneumonic plague during medieval epidemics[98].

The suitability of all tree priors was evaluated using path sampling as implemented in the model selection package of BEAST2 v.6.6. Path sampling was run in 50 steps, with 20 million states as the chain length for each step. The resulting log-marginal likelihoods favoured with 'strong support'[99] the coalescent skyline model for the present analysis (log Bayes factor= 8.35 when compared against the second best model) (Supplementary Table 20). Therefore, the coalescent skyline model was chosen for further analysis. To evaluate the temporal signal in the present dataset, we used TempEst v.1.5.3 to estimate the root-to-tip distance against specimen ages in a linear regression analysis[100]. For TempEst, we used a maximum parsimony tree computed in MEGA7 (ref. [86]) in NEXUS format. Moreover, we used the midpoint of the archaeological or radiocarbon date ranges for all ancient genomes as tip dates. All modern genome ages were set to 0 years before the present. The resulting correlation coefficient $r$ (0.39) and $R^2$ (0.16) values supported the existence of a temporal signal in the present dataset. Furthermore, we used the BETS approach[101] for a temporal signal assessment that takes into account all analysis parameters. BETS compares the (log)-marginal likelihood estimations produced from an isochronous model (all sampling dates set to 0 years before the present) against a heterochronous model (including real sampling times). As previously, path sampling was run in 50 steps with 20 million states as the chain length for each step. The estimated (log)-Bayes factor of 129.33 was in strong support of the heterochronous model; therefore, it indicated the presence of a temporal signal in the present dataset.

For the molecular dating analysis using a coalescent skyline model set-up, we performed Markov chain Monte Carlo sampling using 2 independent chains of 300–400 million states each. After completion, runs were combined using LogCombiner v.2.6.7 and convergence was evaluated using Tracer v.1.6 (http://tree.bio.ed.ac.uk/software/tracer/) ensuring that the effective sample sizes were greater than 200 for each estimated posterior distribution after a 10% burn-in. Maximum clade credibility trees were constructed using TreeAnnotator in the BEAST2 v.6.6 package[87] with a 10% burn-in and were then visualized in FigTree v.1.4.4. In parallel with the molecular dating analysis, we performed a sampling from the prior analysis to test for possible overfitting of the prior to the data. We performed Markov chain Monte Carlo sampling for 2 independent chains of 600 million states each. After run completion, runs were combined and convergence was evaluated after a 30% burn-in. The results indicate that the posterior distributions of the uncorrelated log-normal relaxed clock and the time to the most recent common ancestor estimates are not concordant with those obtained when using a data-informed analysis (Supplementary Fig. 13).

Because most Bayesian phylogenetic frameworks (such as BEAST2) are based on bifurcating trees and hence are poor at resolving multifurcating nodes, we complemented our approach by using TreeTime v.0.8.4 (ref. [35]) to infer a time-calibrated phylogeny using a maximum likelihood approach. TreeTime has been shown to resolve polytomies in a way that is consistent with specimen tip dates. We generated a rooted maximum likelihood phylogeny using RAxML (Supplementary Fig. 10) from the same SNP alignment as the one used for BEAST2 (95% partial deletion). The maximum likelihood tree was then used as input for TreeTime, which was run using all known sampling dates for modern genomes and the midpoint of the age range for the ancient genomes (Supplementary Table 22). TreeTime was run using the Kingman coalescent tree prior with the skyline setting. An appropriate substitution model was chosen for the data using the -gtr infer option. The time-scaled phylogeny was inferred using an uncorrelated relaxed clock and with the branch length optimization, keep-root and keep-polytomies options. Moreover, the divergence time intervals were estimated from the highest likelihood tree using the -confidence option. Analyses were run using a maximum number of 500 and 1,000 iterations (maximum number of iterations option) and produced consistent outputs. The resulting time tree can be found in Supplementary Fig. 11.

### Reporting summary

Further information on research design is available in the Nature Research Reporting Summary linked to this paper.

## Data availability

The raw sequence data produced in this study, the *Y. pestis* aligned reads after metagenomic filtering and the human aligned reads are available through the European Nucleotide Archive under accession no. PRJEB46734. More data are available in the Supplementary Information.

## Code availability

No specialized custom code was used for this study. All software used for the data analyses in this study is publicly available.

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

**Acknowledgements** We thank A. Herbig, G. Neumann and A. Andrades Valtueña and all other members of the Molecular Palaeopathology and Computational Pathogenomics working groups of the Max Planck Institute for Evolutionary Anthropology for helpful discussions throughout the course of this study; C. Posth and R. Barquera for comments on early versions of the manuscript; T. Hermes for helpful discussions during the review of this study; M. O'Reilly for graphical support; R. I. Tukhbatova, N. Martins, F. Aron, A. Wissgott, R. Radzeviciute and G. Brandt at the Max Planck Institute for the Science of Human History in Jena as well as S. Nagel at the Max Planck Institute for Evolutionary Anthropology in Leipzig for laboratory support; K. Prüfer and S. Clayton for computational assistance. Radiocarbon dating took place at the Curt-Engelhorn-Zentrum Archäometrie in Mannheim, Germany; we thank R. Friedrich for assisting with the interpretation of the radiocarbon dating results. Moreover, we thank V. I. Selezneva from the Peter the Great Museum of Anthropology and Ethnography for initial aid with specimen sampling and N. Smelova, currently at the University of Oslo, for providing essential contextual information during the initial phase of this project. This project received funding from the European Research Council under the European Union's Horizon 2020 research and innovation programme under grant no. 771234 (PALEoRIDER to W.H.). L.M. and L.D. were supported by grant no. AP08856654 from the Ministry of Education and Science of the Republic of Kazakhstan. M.A.S., G.A.G.R., A.K., K.I.B., D.K. and J.K. were also supported by the Max Planck Society.

**Author contributions** M.A.S., P.S. and J.K. conceived and led the investigation. M.A.S., K.I.B., P.S. and J.K. designed the study. M.A.S. and L.M. performed the laboratory work. M.A.S. and A.K. performed the bacterial genomic data analysis. M.A.S., A.K. and D.K. performed the molecular dating analysis. G.A.G.R. performed the human population genetic analysis. P.-G.B. and P.S. assembled, analysed and translated the historical, archaeological and epigraphic context information. A.B. and V.I.K. provided access to the archaeological material and contextual information. M.A.S., A.K., L.D., K.I.B., D.K., W.H., P.S. and J.K. aided in interpreting the results. M.A.S. and P.S. wrote the paper with contributions from all co-authors.

**Funding** Open access funding provided by Max Planck Society.

**Competing interests** The authors declare no competing interests.

**Additional information**
**Correspondence and requests for materials** should be addressed to Maria A. Spyrou, Philip Slavin or Johannes Krause.

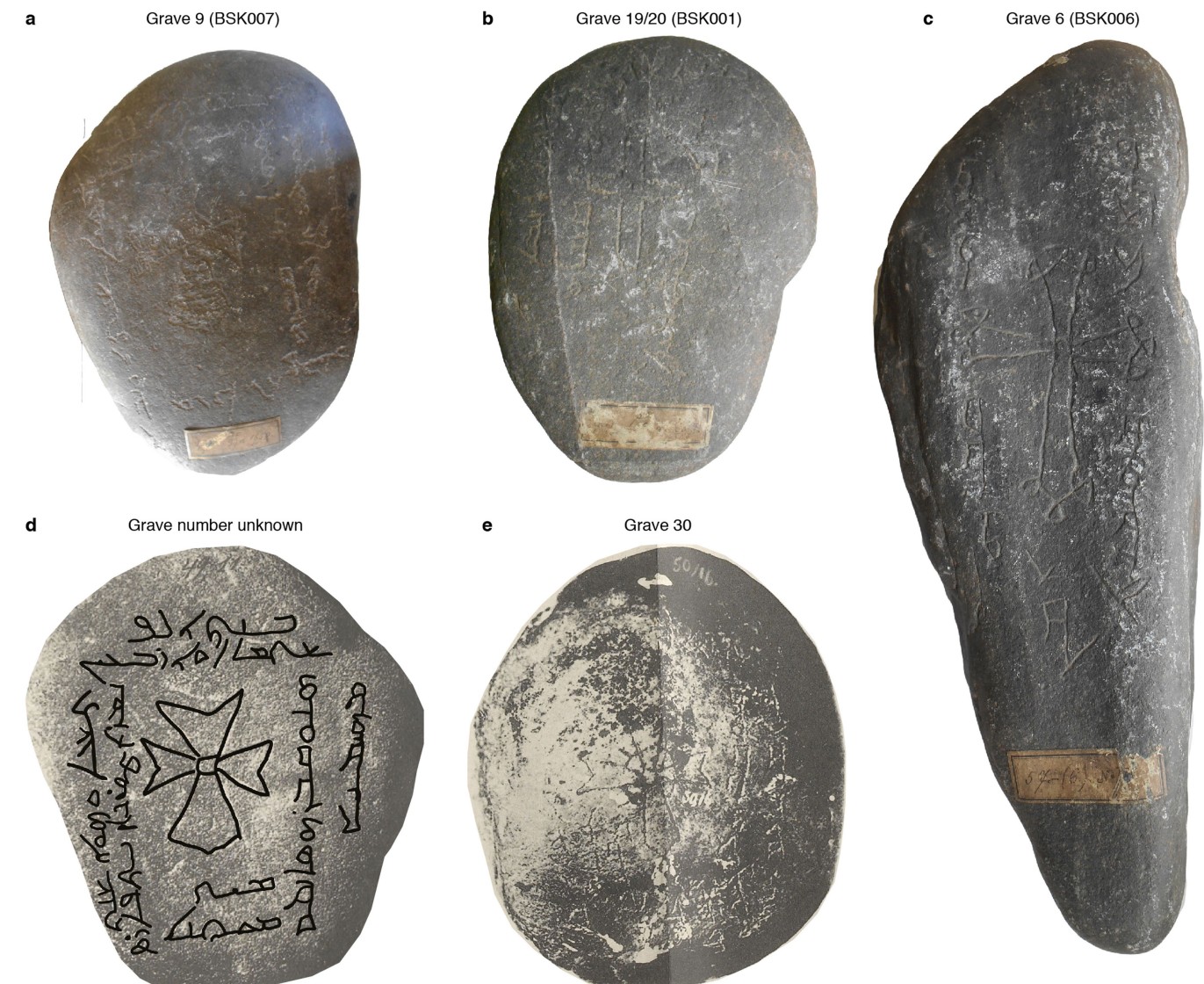

**a** Grave 9 (BSK007)  **b** Grave 19/20 (BSK001)  **c** Grave 6 (BSK006)

**d** Grave number unknown  **e** Grave 30

**Extended Data Fig. 1 | Available tombstone pictures from Kara-Djigach. a–c,** Available tombstone pictures from individuals investigated as part of this study. For a translation of the tombstone inscriptions, see individual descriptions within Supplementary Information 2. Tombstone dates are as follows: Grave 9 (1338-9 CE), Grave 19/20 (1338-9 CE), Grave 6 (year not inscribed). Picture credits to P.-G. Borbone. **d+e,** Tombstones identified in Kara-Djigach containing pestilence-stating inscriptions, dating to the years 1338 and 1339 CE. These tombstones do not correspond to individuals analysed within our aDNA dataset. The original tombstone on panel **d** (without traced inscription) is shown in Fig. 1. Complete translations are available within Supplementary Information 2.

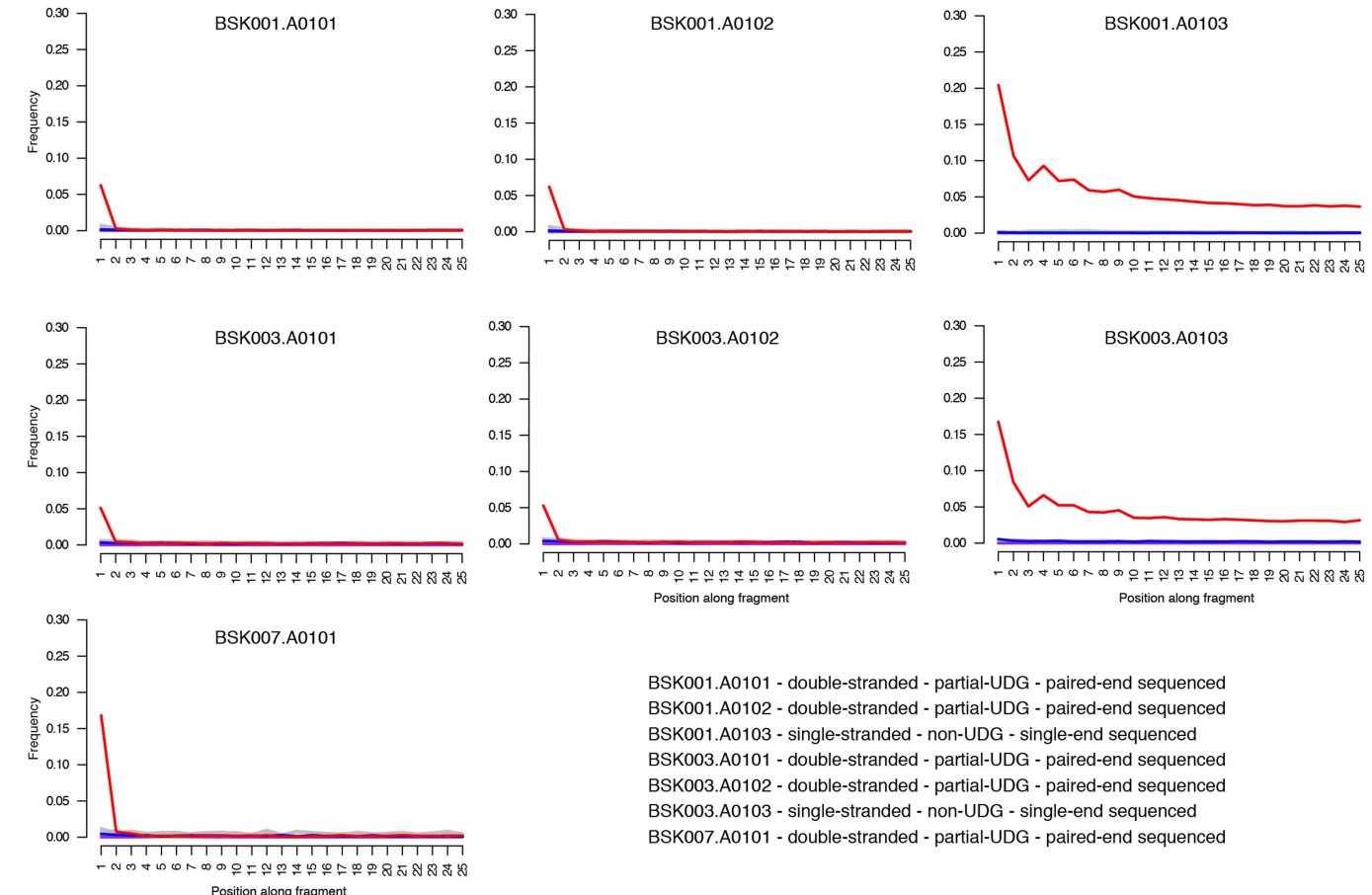

BSK001.A0101 - double-stranded - partial-UDG - paired-end sequenced
BSK001.A0102 - double-stranded - partial-UDG - paired-end sequenced
BSK001.A0103 - single-stranded - non-UDG - single-end sequenced
BSK003.A0101 - double-stranded - partial-UDG - paired-end sequenced
BSK003.A0102 - double-stranded - partial-UDG - paired-end sequenced
BSK003.A0103 - single-stranded - non-UDG - single-end sequenced
BSK007.A0101 - double-stranded - partial-UDG - paired-end sequenced

**Extended Data Fig. 2 | Ancient DNA damage substitution frequencies for all *Y. pestis* captured libraries.** C-to-T substitution frequencies characteristic of post-mortem deamination of ancient DNA are shown for the 5′ ends of sequenced reads aligned against the CO92 *Y. pestis* reference genome (NC_003143.1).

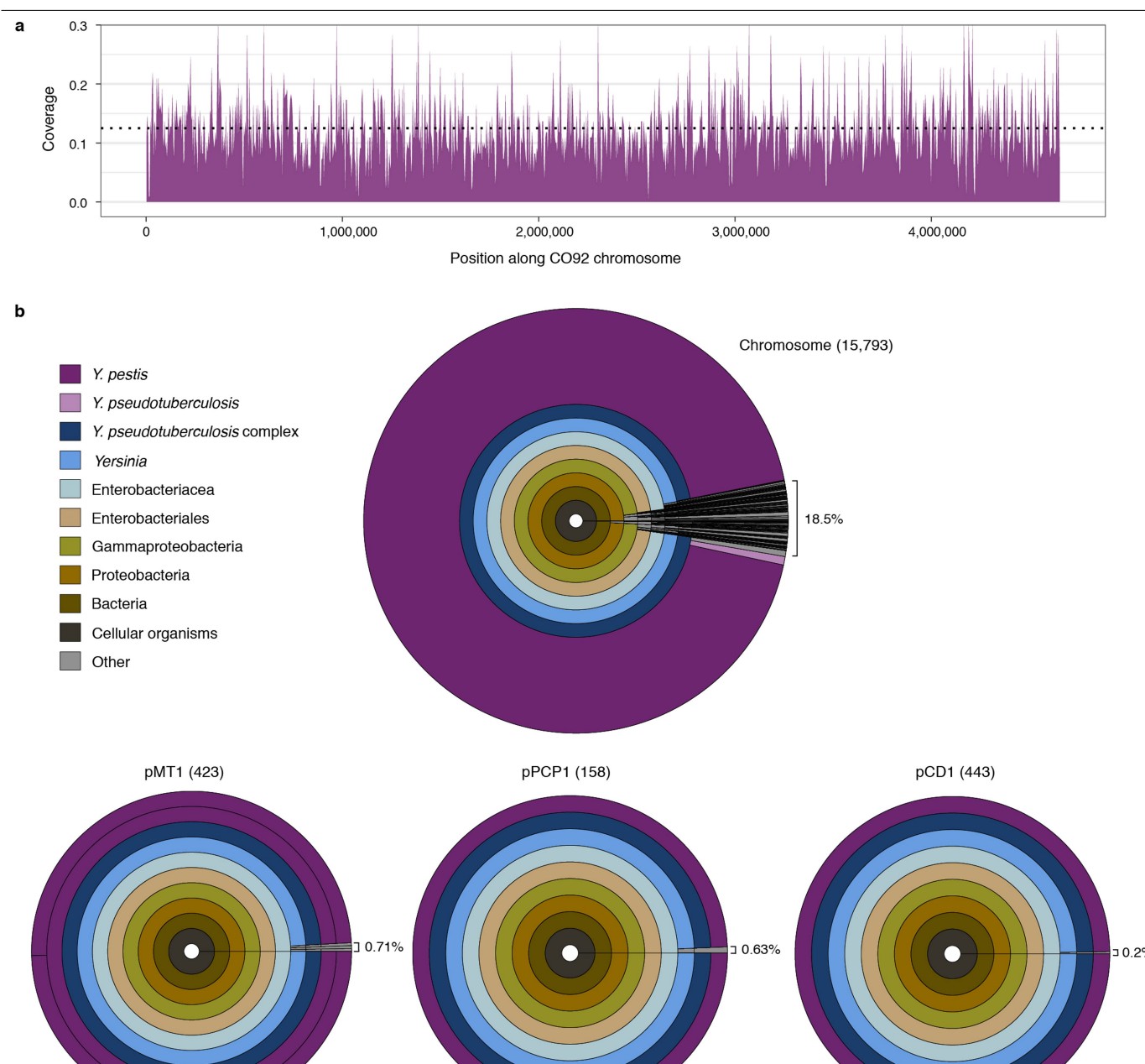

**Extended Data Fig. 3 | Evaluation of BSK007 after whole-genome *Y. pestis* capture. a**, Post-capture coverage distribution of BSK007 across the *Y. pestis* CO92 chromosome. Mean coverage was estimated across the genome in 4,000 bp windows. The dotted gray line indicates the mean coverage across the entire genome (0.125-fold). **b**, Krona plots showing the taxonomic classification of BSK007 reads mapping against all *Y. pestis* CO92 elements (chromosome NC_003143.1, pMT1 NC_003134.1, pPCP1 NC_003132.1 and pCD1 NC_003131.1).

Numbers in brackets next to element designations correspond to the number of assigned reads in MALT. The colours of Krona sectors represent different taxonomic levels and their completeness is proportional to the relative abundance of summarised reads at each corresponding taxonomic node. The shown percentages indicate the species-level (outermost circle) proportion of reads aligned to taxa other than *Y. pestis* and *Y. pseudotuberculosis*.

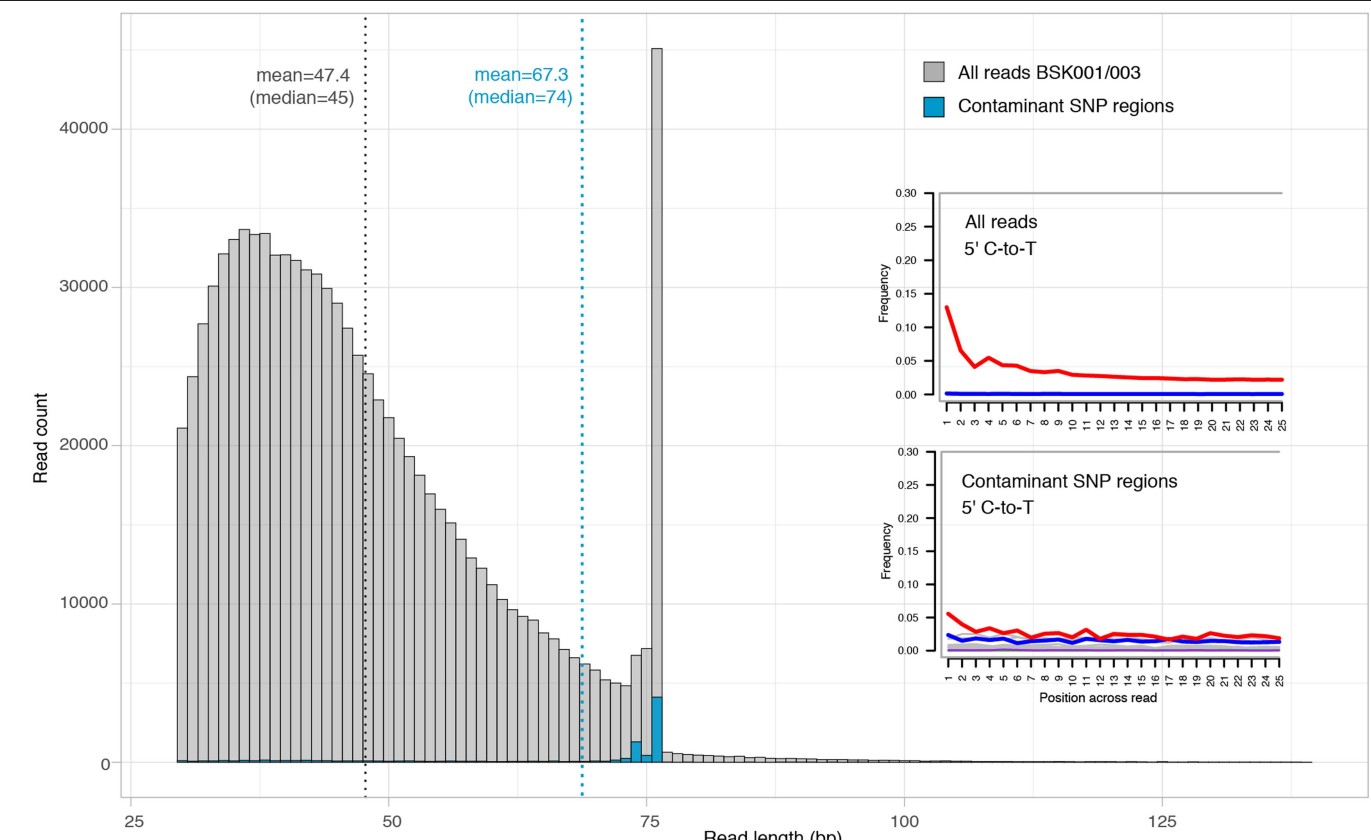

**Extended Data Fig. 4 | Read length and ancient DNA damage distribution of contaminant SNP regions in the combined BSK001/003 dataset prior to metagenomic filtering.** Overlayed length distributions of reads mapping against the CO92 *Y. pestis* reference genome, calculated for the entire dataset (gray) as well as for 117 regions surrounding putatively contaminant SNPs (blue). Regions were extracted within a 150 bp window surrounding each putatively contaminant SNP. Dotted lines represent average fragment lengths for the entire dataset and for the 117 putatively contaminant SNP regions in gray and blue, respectively. Reads comprising contaminant SNP regions show a distinct length distribution compared to the one observed across the entire BSK001/003 genome, with a marked shift towards longer read lengths. The 76 bp fragment length peak represents the uppermost possible read length of single-end sequenced reads, which comprised the majority of data within the present dataset. Ancient DNA damage patterns were compared between the entire dataset (upper panel) and the putatively contaminant SNP regions (lower panel), showing a near 3-fold reduction in the latter as estimated for the terminal 5' base.

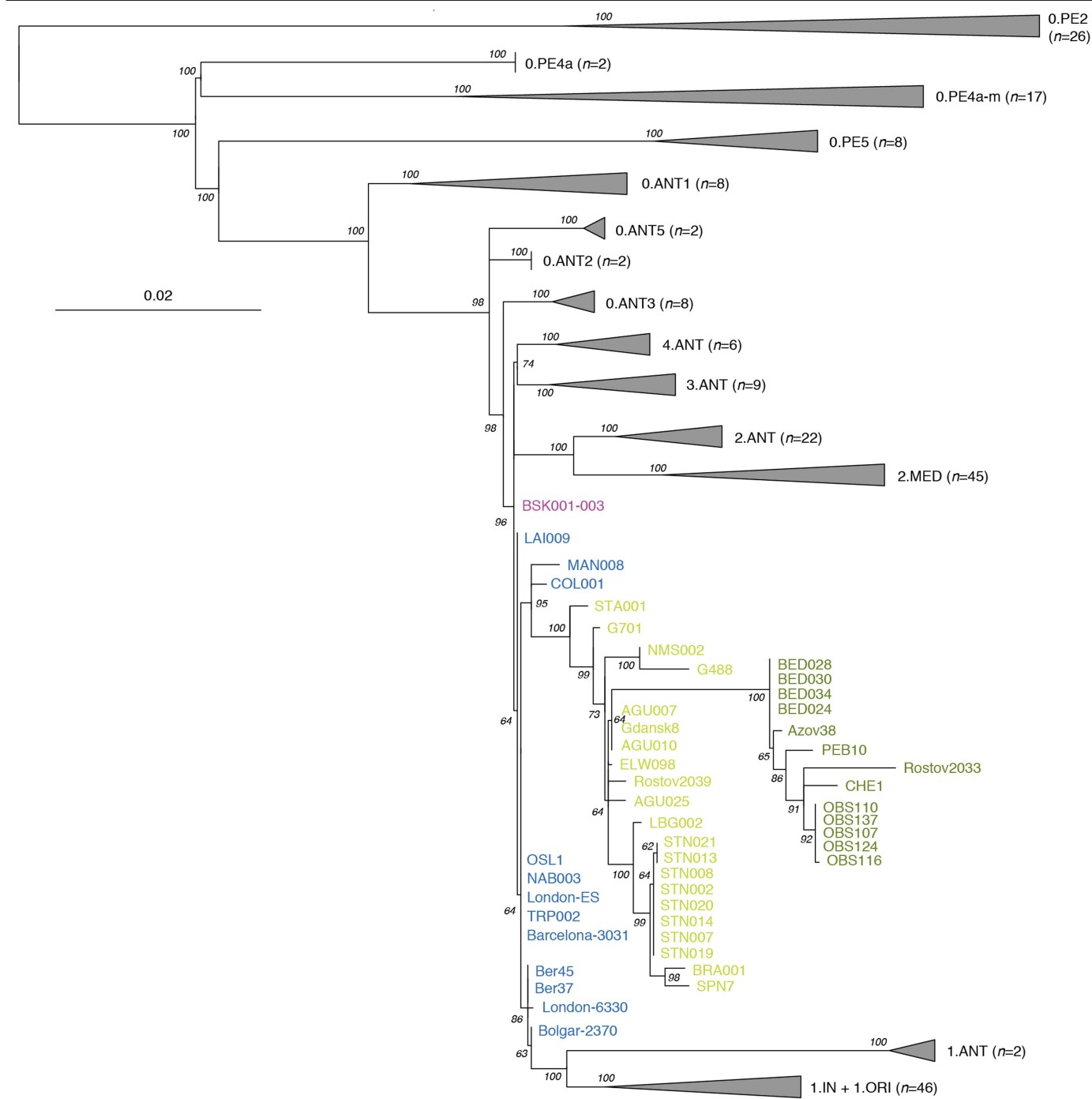

**Extended Data Fig. 5 | Phylogenetic comparisons between BSK001/003 against ancient and modern *Y. pestis* diversity.** Full length maximum likelihood phylogenetic tree using 1,000 bootstrap iterations for estimating node support and visualised using FigTree v1.4.4. The tree is constructed with 203 modern and 48 historical *Y. pestis* genomes, and is based on 2,960 SNPs (96% partial deletion). Scale denotes the number of substitutions per genomic site.

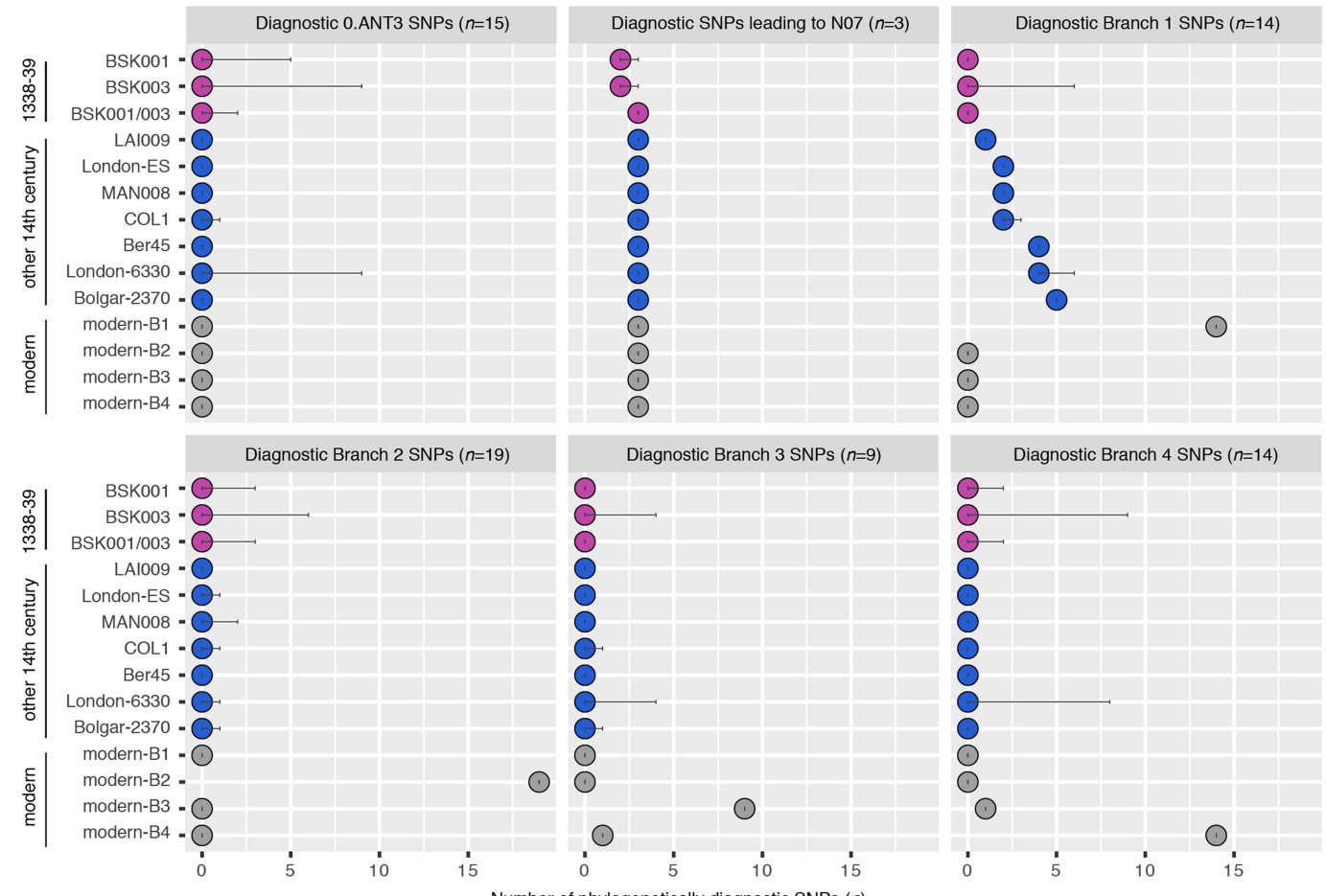

**Extended Data Fig. 6 | Evaluation of phylogenetically diagnostic SNPs across 14th century *Y. pestis* genomes.** The estimated variant calls were retrieved from a SNP table comprising 203 modern and 48 historical *Y. pestis* genomes (full dataset contains 3,533 SNPs). Error bars indicate uncertainty due to the presence of missing data (Ns) within the variant calls of the respective genomes.

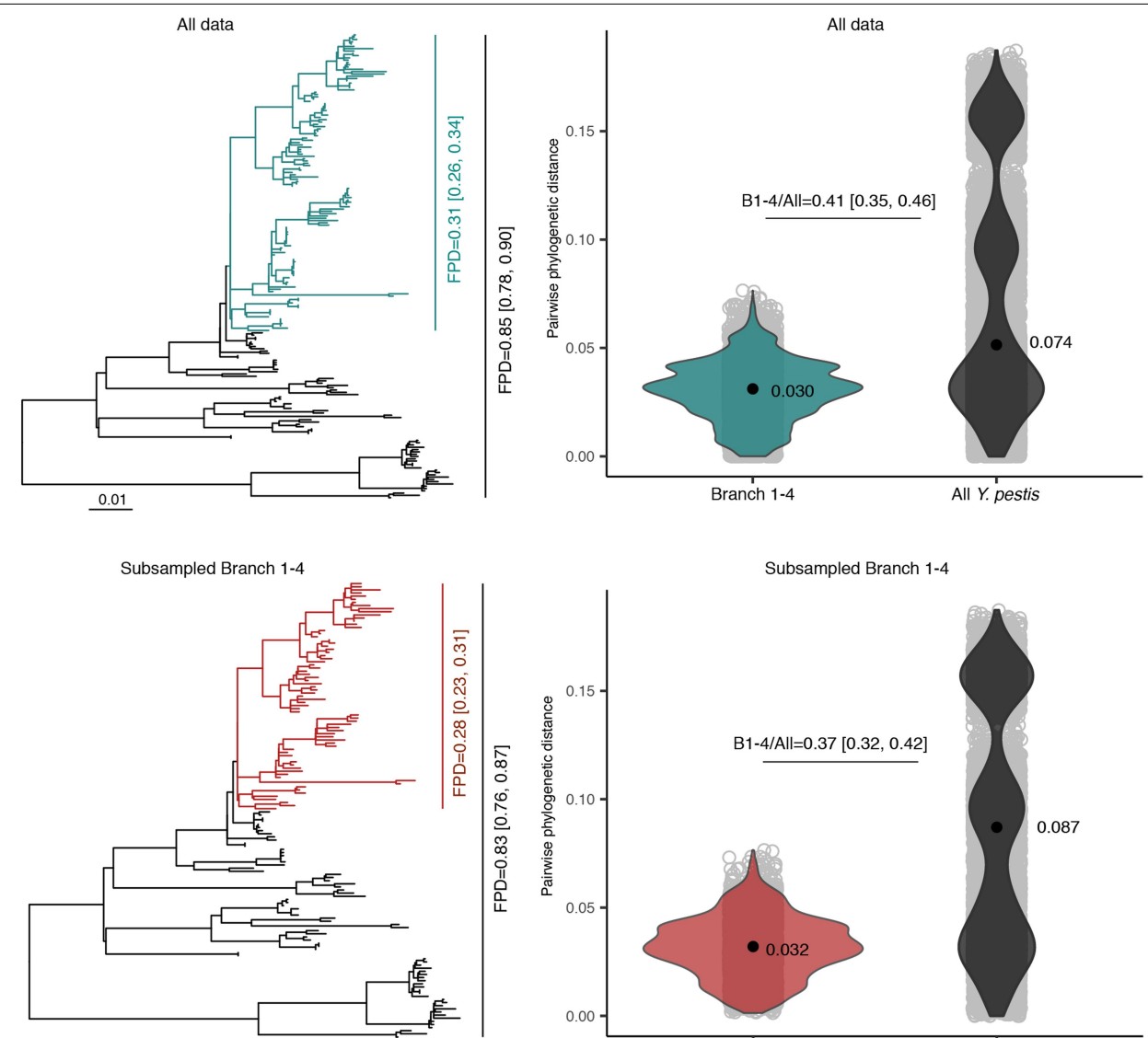

**Extended Data Fig. 7 | Comparison of phylogenetic diversity measures computed for the complete modern *Y. pestis* phylogeny against the Branch 1-4 subclade.** The left panels show maximum likelihood substitution trees considering only extant *Y. pestis* genomes, annotated based on the compared clades and their corresponding Faith's phylogenetic diversity index (FPD). Phylogenetic branches considered for the Branch 1-4 FPD computation are shown in green (full dataset) and red (subsampled dataset). The right panels show violin plots indicating the distribution and mean of pairwise phylogenetic distances based on maximum likelihood trees (MPD). Median estimates and 95% percentile intervals were derived from the resampled bootstrap trees (1,000 bootstrap iterations). Points within violin plots indicate the mean estimated phylogenetic distance for all datasets.

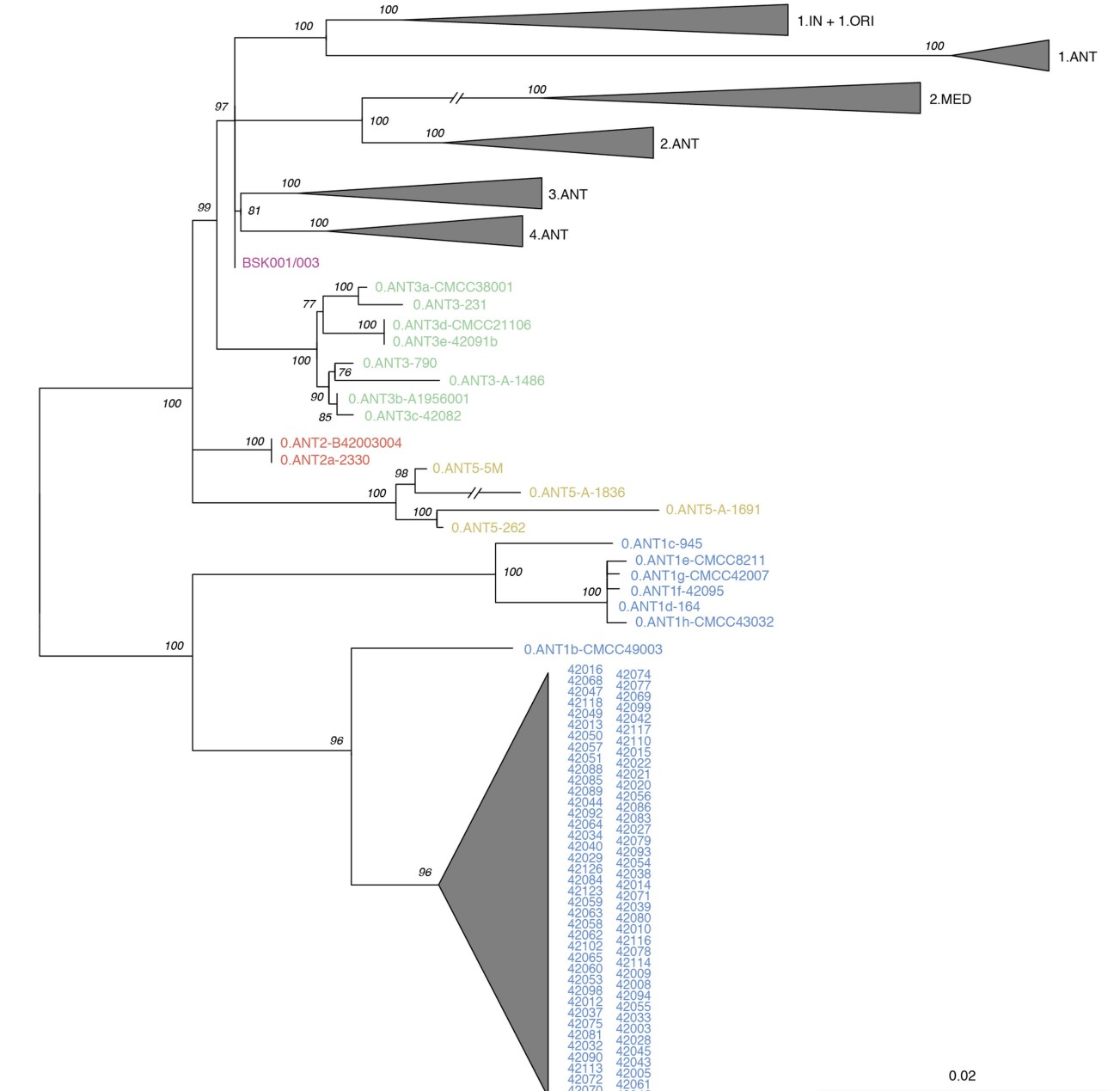

**Extended Data Fig. 8 | Phylogenetic relationships of 0.ANT lineages.**
Maximum likelihood phylogenetic tree based on 2,441 genome-wide variant positions (all SNPs). The tree was constructed to indicate the genetic relationships between all previously published extant 0.ANT genomes and BSK001/003. Scale denotes the number of substitutions per genomic site. Node support was determined by 1,000 bootstrap iterations.

# Reporting Summary

## Statistics

For all statistical analyses, confirm that the following items are present in the figure legend, table legend, main text, or Methods section.

| n/a | Confirmed | |
|---|---|---|
| ☐ | ☒ | The exact sample size (*n*) for each experimental group/condition, given as a discrete number and unit of measurement |
| ☐ | ☒ | A statement on whether measurements were taken from distinct samples or whether the same sample was measured repeatedly |
| ☐ | ☒ | The statistical test(s) used AND whether they are one- or two-sided<br>*Only common tests should be described solely by name; describe more complex techniques in the Methods section.* |
| ☒ | ☐ | A description of all covariates tested |
| ☒ | ☐ | A description of any assumptions or corrections, such as tests of normality and adjustment for multiple comparisons |
| ☐ | ☒ | A full description of the statistical parameters including central tendency (e.g. means) or other basic estimates (e.g. regression coefficient) AND variation (e.g. standard deviation) or associated estimates of uncertainty (e.g. confidence intervals) |
| ☐ | ☒ | For null hypothesis testing, the test statistic (e.g. *F*, *t*, *r*) with confidence intervals, effect sizes, degrees of freedom and *P* value noted<br>*Give P values as exact values whenever suitable.* |
| ☐ | ☒ | For Bayesian analysis, information on the choice of priors and Markov chain Monte Carlo settings |
| ☒ | ☐ | For hierarchical and complex designs, identification of the appropriate level for tests and full reporting of outcomes |
| ☒ | ☐ | Estimates of effect sizes (e.g. Cohen's *d*, Pearson's *r*), indicating how they were calculated |

*Our web collection on statistics for biologists contains articles on many of the points above.*

## Software and code

Policy information about availability of computer code

| Data collection | No software was used for data collection |
|---|---|
| Data analysis | The following software was used for the data analysis portion of this study:<br>bcl2fastq v2.20, EAGER pipeline (version 1.92.58), FastQC version 0.11.4, AdapterRemoval v2.2.0, HOPS pipeline, MALT (v040), MEGAN (v6.4.12), BLASTn (web version), BWA (version 0.7.12), mapDamagev2.0, SAMtools v1.3, Picard version 1.140 MarkDuplicates, DeDup v0.12.2, QualiMap v.2.2.1, BBtools suite (https://sourceforge.net/projects/bbmap/), GATK v3.5, MultiVCFAnalyzer v0.85 (https://github.com/alexherbig/MultiVCFAnalyzer), R version 3.6.1, SNPEvaluation (build date 2018-08-13, https://github.com/andreasKroepelin/SNP_Evaluation), MEGA7, RAxML (version 8.2.9), TempEst v1.5.3, BEAST2 v6.6, Tracer v1.6, TreeTime v0.8.4, FigTree v1.4.4, GrapeTree version 1.5.0, Schmutzi (https://github.com/grenaud/schmutzi), HaploGrep2, pileupCaller v1.4.0 (https://github.com/stschiff/sequenceTools), bamUtil v.1.0.13, READ (https://bitbucket.org/tguenther/read/src/master/), qpWave/qpAdm (v1520), pMMR (https://github.com/TCLamnidis/pMMRCalculator), smartpca v16000, ANGSD v0.910, EIGENSOFT v6.0.1 and QGIS 3.22.1.<br>All listed software used for the data analysis portion of this study is publicly available. |

For manuscripts utilizing custom algorithms or software that are central to the research but not yet described in published literature, software must be made available to editors and reviewers. We strongly encourage code deposition in a community repository (e.g. GitHub). See the Nature Portfolio guidelines for submitting code & software for further information.

## Data

Policy information about availability of data

All manuscripts must include a data availability statement. This statement should provide the following information, where applicable:
- Accession codes, unique identifiers, or web links for publicly available datasets
- A description of any restrictions on data availability
- For clinical datasets or third party data, please ensure that the statement adheres to our policy

The raw sequence data produced in this study, the Y. pestis aligned reads after metagenomic filtering and the human aligned reads are available through the European Nucleotide Archive under accession number PRJEB46734. Additional data is available within the Supplementary Information section of this study. Comparative data including Y. pestis genome accessions can be found in Supplementary table 13 . Comparative human genomic data was retrieved from version v50.0 of the Allen Ancient DNA resource (https://reich.hms.harvard.edu/allen-ancient-dna-resource-aadr-downloadable-genotypes-present-day-and-ancient-dna-data).

# Field-specific reporting

Please select the one below that is the best fit for your research. If you are not sure, read the appropriate sections before making your selection.

☒ Life sciences          ☐ Behavioural & social sciences          ☐ Ecological, evolutionary & environmental sciences

For a reference copy of the document with all sections, see nature.com/documents/nr-reporting-summary-flat.pdf

# Life sciences study design

All studies must disclose on these points even when the disclosure is negative.

| | |
|---|---|
| Sample size | Specimen sample size was determined on the basis of available skeletal material stored within the Kunstkamera, Peter the Great Museum of Anthropology and Ethnography, Russian Academy of Sciences, in St. Petersburg, Russia. The seven tooth specimens analysed in this study were evaluated on the basis of ancient DNA (aDNA) preservation using previously defined cirteria. Specimens with sufficient levels of aDNA preservation were used for whole-genome or genome-wide variant analysis. A detailed description of specimen analysis is provided within the Methods section of this study. |
| Data exclusions | Data from specimens that showed insufficient levels of ancient DNA preservation were excluded from further genomic analyses. Genomic analyses of specimens that showed sufficient ancient DNA preservation were carried out after read filtering according to previously defined ancient DNA criteria. In brief, raw sequenced reads that displayed poor sequencing quality and those shorter than 30 base pairs in length were excluded. In addition, reads of with low mapping quality (<30 for human mapping reads and <37 for Y. pestis mapping reads) were also filtered. Moreover, for the analysis of the newly generated Y. pestis genomes, a taxonomy-informed read filtering was implemented in this study in order to exclude DNA fragments that potentially stem from environmental microbial contamination. For the comparative dataset, present-day genomes showing possible presence of contaminant SNPs were excluded on the basis of their terminal branch lengths, whereby genome assemblies with excessively long branches were not considered for evolutionary analysis as their associated raw data could not be evaluated. A detailed description of data filtering is provided within the Methods section of this study. |
| Replication | For individuals where ancient Y. pestis DNA was detected, three genetic libraries were produced from each of the specimens BSK001 and BSK003, and one genetic library was generated from BSK007, all confirming the pathogen's presence. Evolutionary inferences were performed using different methods, including the maximum likelihood and Bayesian phylogenetic methods. Phylogenetic analyses were furthermore repeated using different comparative datasets. All methods used support the conclusions reported in this study. A detailed description is provided within the Methods section of this study. |
| Randomization | No experiment or analysis requiring allocation of samples/organisms or participants in random groups was carried out for this study. |
| Blinding | Blinding is not relevant to this study. No experiment or analysis requiring group allocations was carried out for this study. |

# Reporting for specific materials, systems and methods

We require information from authors about some types of materials, experimental systems and methods used in many studies. Here, indicate whether each material, system or method listed is relevant to your study. If you are not sure if a list item applies to your research, read the appropriate section before selecting a response.

## Materials & experimental systems

| n/a | Involved in the study |
|-----|----------------------|
| ☒ | ☐ Antibodies |
| ☒ | ☐ Eukaryotic cell lines |
| ☐ | ☒ Palaeontology and archaeology |
| ☒ | ☐ Animals and other organisms |
| ☒ | ☐ Human research participants |
| ☒ | ☐ Clinical data |
| ☒ | ☐ Dual use research of concern |

## Methods

| n/a | Involved in the study |
|-----|----------------------|
| ☒ | ☐ ChIP-seq |
| ☒ | ☐ Flow cytometry |
| ☒ | ☐ MRI-based neuroimaging |

# Palaeontology and Archaeology

| | |
|---|---|
| Specimen provenance | Excavations of the Kara Djigach and Burana cemeteries took place during the years 1885 and 1886 by N. Pantusov and A. Fetisov. Human skeletal remains have since the year 1937 been stored at Kunstkamera, Peter the Great Museum of Anthropology and Ethnography, Russian Academy of Sciences, in St. Petersburg, Russia. A detailed description of the excavations and sample provenance is included within the Supplementary Information section of this study. Samples from Kara Djigach (archaeological IDs: 176/4; 176/5; 176/7; 5559/1; 5559/2 , aDNA Jena lab IDs: BSK001,  BSK002, BSK003, BSK006, BSK007) and from Burana (archaeological IDs: 188/1; 188/2, aDNA Jena lab IDs: BSK004,  BSK005) were analysed within the ancient DNA clean room facilities of the Max Planck Institute for the Science of Human History,in Jena, Germany, with permission from Kunstkamera, Peter the Great Museum of Anthropology and Ethnography, Russian Academy of Sciences. |
| Specimen deposition | The skeletal assemblages associated with the Kara Djigach and Burana archaeological sites are kept within the collection of the Kunstkamera, Peter the Great Museum of Anthropology and Ethnography. |
| Dating methods | Human skeletal remains have been precisely dated on the basis of associated burial tombstones. Detailed translations of all tombstone inscriptions associated with the analysed burials are provided within the Supplementary Information section of this study. Moreover, radiocarbon dating was performed for individuals BSK003 and BSK007 that were positive for Y. pestis. Radiocarbon dating was performed in the Curt-Engelhorn-Zentrum Archäometrie gGmbH in Mannheim, Germany. Collagen was extracted from the tooth roots (modified Longin method) and purified by ultrafiltration (fraction >30kD). Resulting dates were calibrated using the dataset IntCal20 and the software SwissCal (L.Wacker, ETH-Zürich). The corresponding laboratory IDs, uncalibrated radiocarbon dates and 2-sigma (95.45%) probability intervals are provided in Supplementary Information 2 of this study. |

☒ Tick this box to confirm that the raw and calibrated dates are available in the paper or in Supplementary Information.

| | |
|---|---|
| Ethics oversight | Seven tooth specimens from the archaeological sites of Kara-Djigach (n=5) and Burana (n=2) have been analysed in the present study. Approvals for ancient DNA analysis have been obtained from the relevant custodians within the Kunstkamera, Peter the Great Museum of Anthropology and Ethnography, Russian Academy of Sciences, in St. Petersburg, who are co-authors in this paper and have approved the study protocol. |

Note that full information on the approval of the study protocol must also be provided in the manuscript.

