## [Peer Review File · Nature]

Manuscript Title: The source of the Black Death in 14th-century central Eurasia

Reviewer Comments & Author Rebuttals

Reviewer Reports on the Initial Version:

Referees' comments:

Referee #1 (Remarks to the Author):

A/Summary and B/Originality and Significance: In this manuscript, Spyrou and colleagues characterized the complete genome sequence of a plague strain from 1338-1339 CE, leveraging archaeological material uncovered by the late 19th century in Central Asia. This represents a key achievement both due to the date and to the location of the material investigated. It is indeed the first time that plague genomes from the 2nd pandemic are characterized from far outside Europe. The material also dates to one decade before the pandemics entered Europe. The material analyzed is, thus, essential to understand the early development of the 2nd pandemic, before it reached Europe and killed 30-60% of the population there (incidentally, the authors should cite the full range rather than just insisting on 60%, as the range is already dramatic enough; page 2, line 51). The genome data supports a rather late emergence of the underlying strains, during the 14th century, and not during the 13th century as most commonly posited. This represents a major finding and will settle long-standing debates in historical science. It will also be of great interest for a broad audience, and is thus fully appropriate for publication in Nature.

C. Data and methodology, and F Suggested improvements: The authors make use of state-of-the-art methods in ancient DNA research, including a combination of shotgun metagenomics for screening, and target-enrichment techniques for genome production. They also apply another set of target-enrichment techniques to characterize the genomic makeup of the human individuals infected. I generally found no issues with the methodology as implemented, both regarding wet-lab and dry-lab techniques. I was, however, surprised to see that the authors applied lenient mapping parameters to the whole set of sequences generated (and not to just a subset). Supp Table 5 and Extended Fig 3 yet show that only two libraries, representing about 29% of the total number of preprocessed reads, have been obtained from DNA extracts that were not USER-treated. The vast majority of the sequences is thus almost devoid of DNA damage and may be well qualified for stringent mapping conditions (while the remaining part could be mapped with lenient parameters). I am also surprised that the authors used seeding for mapping, while it is recommended to remove seeding if aimed at increasing the mapping sensitivity for those reads containing damage signatures (eg see Schubert et al. 2012). Since there are only about 69M of reads to be mapped, the extra running time incurred may be manageable, especially as it may help increase genome coverage, hence, quality. The analysis of the human DNA data is well done (although the mq filter used in schmutzi/haplogrep is rather permissive, and I am curious to see the posterior probability of the mtDNA assignments, and the support for alternative assignments). Unfortunately this section is not much informative as it remains unknown whether the people buried at Kara-Djigach were local in origins. Isotopic signatures may offer a range of complementary analyses to address this and I would suggest to add

such information, if available (to be clear, I am not conditioning acceptance on this, just suggesting to develop this aspect if at all possible). If no further data are available about the human genomic makeup, I would suggest to reduce this section to minimal, as it currently breaks the flow. Out of curiosity, have the authors run any kinship analysis?

D Statistics and Uncertainties: With the exception of the mapping conditions and/or filters discussed above, I found the statistical methodologies appropriate. I particularly appreciated the historical/linguistic effort made to track all possible information about the material investigated. This was critical since it was originally excavated by 1885-1892 and transferred to different institutions then and since. Even though the authors have done a fantastic job at tracking all possible information, I could not help but thinking that, assuming no issues with calibration curve in the expected time range, radiocarbon dating of some of the material could further confirm that the material investigated is indeed that originally excavated. On this issue, the authors point to some information swapping between graves 21 and 22 (SI, page 10) but they don't tell whether the age of both graves is known. Similarly, I was not able to track grave 19 on pages 6 and 7 of the SI material (this was possible for the other 4 graves analyzed from Kara-Djigach). Additionally, I noticed that the BSK001/003 branch is the first to split on the MMC tree shown in Fig3b, and remains separated from all descending lineages with some decent internal branch. This, and the differentiation between branches 1-4, does not look exactly the same in Fig4a though. Can the authors clarify whether the same data set (site-wise) was used in both analyses and elaborate why do we see such differences?

E Conclusions. The conclusions are sound and not overtoned. I wish the authors address the points listed above, as some may even strengthen their current evidence.

G References are appropriate.

H Clarity and context. The manuscript is clear and well written, but at times un-necessarily lengthy. This may particularly apply to the sections on the archaeological context and human DNA analysis. I have noticed a number of typos but nothing serious (eg Black Deah on line 179, page 5).

Referee #2 (Remarks to the Author):

A. The manuscript describes two genomes with a summed coverage of about 9fold plus a third with low coverage from graves dated by the gravestones from 1338. According to core genome SNPs, the two genomes are located at node N07 defined by Cui et al as the founding node of major splits in the *Y. pestis* genealogy that resulted in five modern major branches. This dates the split into those branches to 1838 or earlier, and locates predecessors of the Black Death of 1848 eastwards to Kyrgyzstan.

B. One of the primary authors, P. Slavin, published citation 11 in 2019 which pointed out all the anthropological conclusions summarised here and predicted that the two cemeteries investigated here might contain the predecessors of *Y. pestis* that are now described here. That paper also included a histogram of the dates on the grave inscriptions which seems to have been reproduced with minor modifications in the figures in this manuscript, but is not attributed to the earlier

publication. Historians tend to publish the same data more than once but this smacks of lack of originality.

C. The data and methodology are fine. However, the methods do not differ dramatically from those in multiple publications by Spyrou in the last few years. And the presentation is so methodical that the manuscript is not exciting even though the data and conclusions are. Basically they have found genomes which map to the founding node of a major split in phylogeny, and are 1-2 SNPs earlier in that phylogeny than any previous genomes. The rest are details, many of which are better placed into supplementary material or methods. The section on the PCA plots of human genomics is unimportant and uninformative and belongs in supplemental text. The figures are superb but the text and message would benefit by being dramatically tightened and shortened.

I am unenthusiastic about the skyline plots used for Beast2 analyses. A few years ago there was an enlightening paper by Eduardo Rocha showing that bacterial coalescent pipelines calculated with Beast all looked the same, and were likely artefactual (<https://doi.org/10.1093/molbev/msw048>) I am unenthusiastic about the calculations on the numbers of genomes per phylogenetic branch. This is a function of sample bias by the author, who used an arbitrary collection of genomes rather than the complete databases available within Enterobase (<https://genome.cshlp.org/content/30/1/138.short>), which is not even cited.

D. The statistics look fine. But what I was missing was a very simple overview of the coverage and numbers of reads for each of the nucleotides at the crucial SNPs distinguishing the two new genomes from other ancient and modern genomes.

E. The conclusions are simple. Ancient genomes have now been found in Kyrgyzstan which predate other ancient genomes that caused the Black Death in Europe, and this supports further investigations elsewhere in central and Eastern Asia. These conclusions are robust, valid, novel, interesting but not surprising. They were basically predicted by Morelli et al and Cui et al prior to ancient DNA work.

F. Tighten the manuscript and lighten the text and approach. It is too heavy at the moment.

G. The authors need to give even more credit to Slavin's 2019 paper.

H. Good abstract. Nice synthesis in the text. Less defensiveness would be better.

Review: Mark Achtman

Review 2021-07-11627A

The source of the Black Death uncovered from a 14th-century plague epidemic in central Eurasia

A) Summary of the key results

The geographic origins of the second plague pandemic (the Black Death) have been subject to endless speculation, with some of the earliest evidence found at the shores of Issyk Kul in what is now Kyrgyzstan. This work presents data generated and screened from seven individuals excavated at two cemeteries in the region. Following classification, this data recovers evidence of the presence of *Y. pestis* in three cases (two of analytical quality following capture enrichment) as well as human genome SNP capture data generated at sufficient quality for two individuals. Despite the low number of genomes, the generation of *Y. pestis* genomic data from these sites in their associated epidemic context immediately preceding the medieval Black Death contributes to the inference that the agent (strain) of the subsequent Black Death likely emerged in a local context confirming previous suggestions based on historical accounts and archaeology of an entrance of the Black Death from East to West Eurasia. The authors suggest their work provides strong evidence for a central Eurasian origin of the Second Plague Pandemic. This is a valuable study, moving away from the “Eurocentric” view of plague the authors claim has existed until now.

B) Originality and significance: if not novel, please include reference

Prior work, based on historical accounts has suggested an origin for the second plague pandemic in the South-West of Russia eg. Barker et al, *Speculum* 96, 97-126 (2021) and previous work by some of these authors hypothesised an Eastern European origin for the second plague pandemic using ancient DNA and phylogenetic methods. The archaeological sites under consideration have also been linked to epidemic ‘pestilence’ events (eg. Supp text 1 and Figure 1d). The work here can therefore be seen as confirmatory for these hypotheses rather than a *de novo* insight. However, the observation of the lineage of *Y. pestis* present at these sites, and the link to the major lineage giving rise to the medieval Black Death provides valuable new information to enrich our understanding of *Y. pestis* phylogeographic patterning and genomic diversity.

I do however have queries on the approach used as outlined below:

C) Data & methodology: validity of approach, quality of data, quality of presentation

Lines 173-188 and methods: For the phylogenetic tip-dating it is stated that two models are tested (coalescent skyline and birth-death). These are models with quite different assumptions about demographic history. Given we are estimating a polytomous event the choice of coalescence model prior is important to justify. Were other models tested and did they also provide similar estimates? I note that coalescent constant approaches were rejected based on previous work (line 630). How were these particular models therefore selected to take forward and why was a GTR substitution model prior used? In addition, given the results presented are highly concordant one explanation could be over fitting of the prior to the data. Again, this is of particular concern given likely uncertainties in the tree building step due to uncertainty at the polytomy basal to the tree and the possibility for a few erroneous SNPs in ancient samples to strongly impact posterior rates. The authors should

provide the posterior tree heights and clock rates obtained when running the analysis without the data (sampling from the prior) to confirm their inference is robust to the choice of prior model specified and be more explicit about why these models were chosen.

Figure 2: Given the identical position of BSK001/BSK003 and claim of conserved genomic diversity (line 143) and joint analysis suggested by line 151 I do not follow what gives rise to the long tail in the diagnostic SNPs presented for BSK001/BSK003 in Figure 2C relative to BSK001. Do the diagnostic SNPs include some that were previously claimed as artefactual (lines 139-143). Were BSK001/003 and BSK003 considered both jointly and separately in this analysis? I'd appreciate additional clarification in the methods.

Figure 3: What % of the posterior tree distribution support the basal placement of BSK001/003? Can this polytomy be considered as resolved using a time tree approach?

D) Appropriate use of statistics and treatment of uncertainties

Lines 181/612: A prerequisite of phylogenetic tip-dating is the presence of temporal signal in the alignment. The robustness of the signal should be assessed with provided p -value following date randomization. The temporal regression plot should also be provided.

Supplementary table 2: many more reads are assigned to the *Y. pseudotuberculosis* complex than *Y. pestis* stricto. Were other *Yersinia* species present at reasonable abundance in the shotgun data? How may this have impacted the results eg. cross mapping or genus ubiquitous capture? Extended Data Fig 4 suggests such an effect is reasonable for BSK007 (~17.2% post capture). It would be useful to see the equivalent figures for BSK001 and BSK003 or the data break down on how reads were assignments post capture. I raise this due to comments on the high number of multi-allelic sites in BSK001 and BSK003 (lines 138) which may arise due to the presence of reads deriving from related *Yersinia* species which can be highly diverse. The authors should provide the break-down of read assignments to other *Yersinia* sp. reported by MALT and excluded by the MALT filtering (extended Data Figure 5).

E) Conclusions: robustness, validity, reliability

Lines 191-198: Given some uncertainty in the phylogeny the point estimates of FPD and MPD feel overly precise. Indeed, with different genomic sampling these values may shift potentially quite a lot. In addition, FPD has been shown to be sensitive to rate variation in the tree eg. see Ritchie *et al. Diversity & Distribution* (2021) **27**,1 which is commented on as a property of *Y. pestis* datasets eg. line 176. Given randomly sampling was performed for the modern subsampling it would seem prudent to assess the distribution of FPD and MPD values obtained for the contribution of Branch 1-4 to overall phylogenetic diversity following a leave one out approach and provide these values as a range.

F) Suggested improvements: experiments, data for possible revision

The authors consider the plasmid coverage of their strains. It would seem that this is also useful information to leverage in terms of understanding the phylogeography and age of divergence of the precursor to the medieval Black Death. I would suggest not using this

information in the downstream analysis is a waste of valuable data. In particular, were any attempts made to conduct phylogenetic inference on the shared plasmids, which presumably have accumulated mutations since the split (perhaps resolving the polytomy?) or indeed to use these to the date the resultant phylogeny? Do BSK001 and BSK003 carry exactly the same complements of SNPs in each plasmid?

Some additional supplementary figures would be most valuable to support the validity of the *Y. pestis* genomes obtained eg. visualised edit distances to other *Yersinia* species and the root-to-tip correlation with accompanying significance.

Line 133: Given the detection rate what microbes were identified in the *Y. pestis* negative individuals? Were any other potentially pathogenic/epidemic agents present which could have been relevant to the epidemiology of the site?

G) References: appropriate credit to previous work?

Prior work has been referenced. This necessarily includes heavy referencing of the authors own work given their clear track record in this area.

H) Clarity and context: lucidity of abstract/summary, appropriateness of abstract, introduction and conclusions

The paper is generally very well written and enjoyable to read. There is a lack of details in places but this may be due to the word count restrictions.

The abstract is appropriate though should be more explicit about the number of genomes which were actually used for analysis (2 human and 2 *Y. pestis*).

Additional minor comments

Supplementary table 1: typo in table legend

Why were these three of seven individuals selected for human aDNA genotyping?

Line 55: “Despite intense multidisciplinary research on this topic”, however the studies commented on are almost exclusively based on aDNA data. While these studies may be interdisciplinary in nature it would be appropriate to acknowledge contributions to this topic from other disciplines.

Line 60: ref 25 is a phylogenetic study rather than based on historical accounts as the preceding sentence suggests.

Line 162: what is the sampling age of LA1009 from Volga (if available please state). Is one SNP difference consistent with expectation given the temporal window and estimates of the *Y. pestis* mutation rate (number of mutations per year assuming a strict clock)? Similarly for the Black Death strains which supposedly post-date by at least seven years? Assuming seven years, two SNPs suggests a very slow mutation rate?

Please add relevant accessions to genomes considered in supplementary Table S9 to support future reanalysis with this dataset.

Line 452: Does November 2017 refer to the date of the custom download? How many genomes are considered, and perhaps most relevantly for this case how many distinct *Yersinia* species?

Figure 4: There was no description of PCR-genotyped samples

Extended Data Fig 2: The manuscript text suggests two human genomes were of sufficient quality for SNP capture and used in the PCA eg. line 133 'both individuals' but four 'BSK' data points are provided in the PCA?

Author Rebuttals to Initial Comments:

Point-by-point response to Referees

Nature Manuscript 2021-07-11627A

We are grateful to all three reviewers for their remarks and detailed suggestions and believe that the quality of our study has improved as a result of their input. Below we address all comments through a point-by-point response. All changes and additions within the main text and supplementary information of our study are highlighted in blue.

Referees' comments:

Referee #1 (Remarks to the Author):

A/Summary and B/Originality and Significance: In this manuscript, Spyrou and colleagues characterized the complete genome sequence of a plague strain from 1338-1339 CE, leveraging archaeological material uncovered by the late 19th century in Centra Asia. This represents a key achievement both due to the date and to the location of the material investigated. It is indeed the first time that plague genomes from the 2nd pandemic are characterized from far outside Europe. The material also dates to one decade before the pandemics entered Europe. The material analyzed is, thus, essential to understand the early development of the 2nd pandemic, before it reached Europe and killed 30-60% of the population there (incidentally, the authors should cite the full range rather than just insisting on 60%, as the range is already dramatic enough; page 2, line 51). The genome data supports a rather late emergence of the underlying strains, during the 14th century, and not during the 13th century as most commonly posited. This represents a major finding and will settle long-standing debates in historical science. It will also be of great interest for a broad audience, and is thus fully appropriate for publication in Nature.

We thank Referee #1 for their constructive comments and the overall positive assessment regarding the results and impact of our study.

We agree that a range would be a more representative metric to reflect the mortality rate of the Black Death across Europe. However, the study by O. Benedictow¹, which so far is the most comprehensive overview on the pandemic, includes explicit estimates of a 60% mortality (see also a review of O. Benedictow's book here²). Given the general nature of our statement, whereby we broadly refer to the collective impact of the Black Death, and in order to account for this unrealistically precise metric without misrepresenting O. Benedictow's work, we opted to use the phrase "up to 60%" in our description.

C. Data and methodology, and F Suggested improvements:

The authors make use of state-of-the-art methods in ancient DNA research, including a combination of shotgun metagenomics for screening, and target-enrichment techniques for genome production. They also apply another set of target-enrichment techniques to characterize the genomic makeup of the human individuals infected. I generally found no issues with the methodology as implemented, both regarding wet-lab and dry-lab techniques. I was, however, surprised to see that the authors applied lenient mapping parameters to the whole set of sequences generated (and not to just a subset). Supp Table 5 and Extended Fig 3 yet show that only two libraries, representing about 29% of the total number of preprocessed reads, have been obtained from DNA extracts that were not USER-treated. The vast majority of the sequences is thus almost devoid of DNA damage and may be well qualified for stringent mapping conditions (while the remaining part could be mapped with lenient parameters). I am also surprised that the authors used seeding for mapping, while it is recommended to remove seeding if aimed at increasing the mapping sensitivity for those reads containing damage signatures (eg see Schubert et al. 2012). Since there are only about 69M of reads to be mapped, the extra running time

incurred may be manageable, especially as it may help increase genome coverage, hence, quality.

We thank Referee #1 for this important comment. Initially, we had chosen to use all recovered data (including damaged sites) for genome reconstruction as we considered that the short fragment length observed in our dataset (Supplementary tables 2, 3, 8-11) in combination with the relatively low coverage yields could affect the robustness of our analysis. More specifically, we considered that trimming off the ends of all reads could lower our recovered genomic coverage and variant yields and, therefore, limit our genomic resolution. Following the suggestion by Referee #1 we have now reconsidered this decision and trimmed the terminal bases (one base from each end) from reads recovered after USER treatment (UDG-half protocol). In addition, we have used stringent mapping parameters for USER-treated data (-n 0.1) and have disabled the seeding from all mapping attempts (-l 9999). We find that, despite a minor decrease in the recovered mean genomic coverage across all three newly described *Yersinia pestis* genomes due to read trimming (see comparisons of before and after terminal base trimming in Supplementary tables 9, 10), our SNP yields remain highly consistent between the trimmed and un-trimmed datasets (2,960 total recovered SNPs in the new analysis as opposed to 3,109 reported in the previous version of our paper). Moreover, we find that the stringent mapping parameters used for our re-analysis have helped decrease the levels of environmental contamination within our genomic reconstruction. Specifically, we find less false-positive variants within the re-analysed combined dataset of BSK001/003, both prior metagenomic filtering with MALT (117 identified false-positive variants reported in Supplementary table 16 as opposed to 219 previously identified) and after filtering (one identified false-positive variant reported in Supplementary table 17 as opposed to three previously identified). Therefore, we have chosen to utilise the suggested changes and apply those to all our downstream analyses. Importantly, our re-analysis of BSK001/003 has resulted in an unchanged phylogenetic topology compared to the one reported in the initial version of our paper, therefore, confirming our reported results, which state that BSK001/003 represents the common ancestor of the polytomy that gave rise to *Y. pestis* Branches 1, 2, 3 and 4.

The analysis of the human DNA data is well done (although the mq filter used in schmutzi/haplogrep is rather permissive, and I am curious to see the posterior probability of the mtDNA assignments, and the support for alternative assignments). Unfortunately this section is not much informative as it remains unknown whether the people buried at Kara-Djigach were local in origins.

We thank Referee #1 for this comment. We have now added the posterior probabilities for all our mitochondrial haplogroup assignments (from HaploGrep) after applying quality filters of 10, 20 and 30 as defined by schmutzi. This analysis shows that our assignments remain constant after filtering (see updated Supplementary Table 3).

Moreover, we have further attempted an ancestry modelling of all individuals using *qpWave/qpAdm*³. Below we enclose the relevant results section, now appearing within Supplementary Information 3.

"We performed individual based modeling of BSK001 (85,996 overlapping SNPs), BSK002 (293,486 overlapping SNPs), BSK003 (94,220 overlapping SNPs) and BSK005 (100,702 overlapping SNPs) and we also grouped together BSK001, BSK003 and BSK005 (Bishkek) given their proximity in PCA space to gain resolution given the higher SNP overlap (302,211 SNPs). Overall, the results of these models show that all individuals can be best modeled as a mixture of temporally preceding central Eurasian populations with a minor (~10%) genetic influx from eastern Eurasia. The major source best matches either 6th-century Alanic period groups (Alan) from the Pontic-Caspian steppe or 3rd-century southern Kazakhstan groups related to the Kangju culture (Konyr_Tobe_300CE or Kazakhstan_Kangju.SG)⁴⁻⁷. The minor source best matches older Iron Age Saka/Scythian groups in the case of BSK002 or eastern steppe 1st - 2nd century Xiongnu or Xianbei period groups⁴⁻⁷ for BSK001, BSK003 and BSK005.

The very limited ancient DNA data available from the 1st millennium CE onwards across central Eurasia makes the testing of finer scale hypothesis unfeasible. Nevertheless, these results are in agreement with the depicted PCA clustering, which shows that a broadly defined early medieval central Eurasian genetic source is to be preferred over other sources that instead provide only largely unfeasible models (Pvalue << 0.05; Supplementary Tables 4, 5)."

Isotopic signatures may offer a range of complementary analyses to address this and I would suggest to add such information, if available (to be clear, I am not conditioning acceptance on this, just suggesting to develop this aspect if at all possible). If no further data are available about the human genomic makeup, I would suggest to reduce this section to minimal, as it currently breaks the flow.

We thank Referee #1 for this comment. An investigation of mobility patterns in past populations is usually best performed through the application of strontium isotope ($^{87}\text{Sr}/^{86}\text{Sr}$) analysis of tooth enamel from permanent teeth. Such an analysis, although indeed could provide complimentary results to our study, should always be combined with a robust and detailed examination of the bioavailable $^{87}\text{Sr}/^{86}\text{Sr}$ levels within the investigated region, as an individual's strontium isotope levels will represent a combination of all their consumed food and water during their lifetime. By directly comparing the levels of obtained $^{87}\text{Sr}/^{86}\text{Sr}$ to the pre-determined regional baseline, it may be possible to determine whether an individual was raised locally or migrated to this region during their lifetime. Importantly, a lack—or gaps in information—on the bioavailable strontium within a wider region of interest could cause the mis-interpretation of results, whereby variation rising from individuals acquiring foods from different sources is erroneously considered as a migration pattern. In the case of the Chu Valley of central Eurasia, to our knowledge, the levels of bioavailable Sr have not been mapped and therefore, such an evaluation would require the sampling of plants and small fauna within a wider geographic range in order to produce a comprehensive map of Sr isotopic variation. Moreover, a characteristic of regions with substantial landscape variation, such as the highly mountainous area of interest, is that there is continuous mixing of geological substrates, so sampling must be performed using region-specific expertise. The required multi-month effort with no guarantee of producing meaningful results, therefore, does not permit an addition of such data to our current study. We acknowledge, however, that such an analysis would be sufficient for a stand-alone paper and would make an important contribution to future archaeological and population genetic studies.

To address the suggestion by Referee #1, we employed an alternative approach to shed light on possible places of origin for the inhabitants the Kara-Djigach and Burana. For this, we investigated all surviving 467 tombstone inscriptions for indications of toponyms. Although the investigated tombstones do not mention the ethno-linguistic background of most interred individuals, they include a number of toponyms designating some individuals as 'Mongol', 'Uyghur', 'Armenian' and 'Chinese'. In addition, several individuals are designated as *Ālmālygyā/Ālmālygytā*, namely 'of Almaliq', and one as *Kaškaryā*, that is 'of Kashgar'. Both Almaliq and Kashgar were important international trade cities within the present-day Uygur Autonomous Province (Xinjiang) and are known to have housed sizeable East Syriac communities during the 13th and 14th centuries. Unfortunately, we could not deduce any toponymic information from the tombstones of individuals BSK001, BSK002, BSK003 and BSK005 whose human genomic profiles were analysed as part this paper (see inscription translations below). Nevertheless, based on our collective analysis of all tombstones, it appears likely that Kara-Djigach and Burana had a diverse profile where mobility from close-by and further-away regions of central and eastern Eurasia played an important role in the communities' make-up. This additional analysis now appears as a separate supplementary section and is named "*Toponymic indications of mobility in Kara-Djigach and Burana*" within Supplementary Information 2.

Inscription translations:

BSK001 (Grave 19/20, Kara-Djigach): *"In the Year 1650 [=1338-9 CE], [it was the year of the] Hare. This is the tomb of Bačaq, a faithful woman"*

BSK002 (Grave 22, Kara-Djigach): *"It was in the Year 1650 [=1338-9 CE], the year of the Hare, in Turkic Taviškan. This is the tomb of Sargis the student"*

BSK003 (Grave 28/29, Kara-Djigach, partially illegible tombstone): *"In the Year 1650 [=1338-9 CE] ... Hare, in Turkic Taviškan"*

BSK005 (Grave X, Burana): No tombstone was found in association with this burial.

We agree with Referee #1 that the human genomic analysis performed here offers complimentary results to our study, but does not directly contribute to its central conclusions. Therefore, we have moved this section to our paper's supplement, now appearing as Supplementary Information 3. Nonetheless, we consider those results as important to be represented within our study. Interpreted collectively, our analysis of human DNA, tombstone inscriptions and coin hoards show that mobility for trade or immigration purposes had an impact on the Chu Valley communities and, although a speculation, such practices could have contributed to the spread of plague to and from this region during the 14th century.

Out of curiosity, have the authors run any kinship analysis?

We estimated genetic relatedness between the individuals calculating a commonly used statistic (<https://github.com/TCLamnidis/pMMRCalculator>^{8,9}) called pairwise mismatch rate (pMMR) which is the rate of mismatching alleles between each pair of individuals. This statistic has resolution to identify genetic relatedness down to second degree relatives. We also calculated a similar statistic implemented in READ¹⁰ that estimates pMMR in windows of 100 kb along the genome which allows to calculate standard errors for the estimated degrees of relatedness (identical, first and second degree). Based on these analyses, we have no evidence that any of the individuals analysed are close genetic relatives (analysis now appearing as Supplementary Table 3).

D Statistics and Uncertainties: With the exception of the mapping conditions and/or filters discussed above, I found the statistical methodologies appropriate. I particularly appreciated the historical/linguistic effort made to track all possible information about the material investigated. This was critical since it was originally excavated by 1885-1892 and transferred to different institutions then and since. Even though the authors have done a fantastic job at tracking all possible information, I could not help but thinking that, assuming no issues with calibration curve in the expected time range, radiocarbon dating of some of the material could further confirm that the material investigated is indeed that originally excavated.

We thank Referee #1 for acknowledging our efforts in reconstructing the archaeological context information for the cemeteries of Kara-Djigach and Burana. This has been a challenging effort since these excavations were carried out ~150 years ago and the surviving information is partial and in not digital format. Following the reviewer's suggestion, we have conducted radiocarbon dating on two of three *Y. pestis*-positive individuals, BSK003 and BSK007. The dates were retrieved from the tooth roots of the same teeth that were used for DNA analysis.

Our 14C dating results revealed a wide 2-sigma (95.4%) interval that is consistent between both individuals and ranges between the end of the 13th and the end of the 14th century (see figure below). Such a ~100-year range is typical of late medieval specimens, reported in previous studies (for example^{11,12}). Of note is that radiocarbon dates on human bone collagen may not represent the most accurate way for retrieving absolute dates for archaeological or historical events. Human bone is a complex tissue that reflects an individual's life history, including their diet and health and, therefore, a number of aspects can affect the reliability of its resulting age estimates. Moreover, it has been suggested that

¹⁴C dates from human bone collagen often represent the termination of puberty rather than the absolute age-at-death of an individual, although the assumption that bone is a short-lived tissue has also been considered as an oversimplification. Here, we utilised a method called human bone collagen offset (HBCO) correction¹³ to account for the age-at-death of both BSK003 and BSK007, which has been suggested to offer improved estimates on collagen chronologies. We tested a 20-year offset (± 15 years), as both specimens were likely retrieved from adult individuals. While the resulting dates with respect to the 68.3% (1-sigma), 95.4% (2-sigma) and 99.7% (3-sigma) probabilities remain largely unchanged, we observe an overall increased likelihood across the entire range. Therefore, although our expected dates based on archaeological context information are overlapping with the resulting calibrated radiocarbon interval, we conclude that a narrower age range cannot be retrieved based on radiocarbon dating of these specimens.

In addition, it should be noted that we did not attempt an estimation of the marine or freshwater reservoir effect on these specimens as we do not have access to animal remains for isotopic comparisons. To our knowledge and according to the diary of N. Pantusov (principal archaeologist of the site), no animal remains have been excavated from Kara-Djigach or Burana. We acknowledge, however, that these communities likely explored water sources for dietary purposes (associated with the Chu River and the Lake Issyk Kul). Therefore, an impact of such practices on the retrieved radiocarbon dates could be expected.

Contrary to the wide radiocarbon age range, we believe to be making a strong case for a firm 1338/1339 CE date based on the studied archaeological context, to a scale that is unprecedented for ancient DNA studies. Such context has not only been appreciated by our own team, but has been subject to research by the plague community for more than a century. Here, we provide new details on the site's description with particular attention in its accurate assessment, through close collaboration with a number of experts in the field

including anthropologists from the Kunstkamera Museum (Valeri I. Khartanovich), historians with proficiency in Semitic languages and expertise in Central Asian medieval history (Philip Slavin and Pier-Giorgio Borbone), as well as experts on the palaeography and epigraphy of the Syriac language (Pier-Giorgio Borbone). We are highly confident regarding the information provided since, as stated within Supplementary Information 2, all original documentation on excavated human remains has been retained in the form of archaeological IDs since initial excavations in 1886, and all such information was cross-checked by a corresponding author of our paper, Philip Slavin, during a visit to the (non-digitised) archive of the Institute for the History of Material Culture of the Russian Academy of Sciences in St. Petersburg. Within the relevant section of Supplementary Information 2 we state:

“Importantly, the association between grave numbers, tombstones and the identity of buried individuals can be established as, in the course of the summer 1886 excavations at Kara-Djigach, all tombstones were marked with the same number as their associated graves¹⁴. Likewise, excavated human remains were assigned the same number as their associated graves and tombstones. Moreover, in his subsequent work, D. Chwolson maintained all original grave/tombstone numbers assigned by N. Pantusov during the August 1886 excavations next to his own editorial numbers¹⁵.”

On this issue, the authors point to some information swapping between graves 21 and 22 (SI, page 10) but they don't tell whether the age of both graves is known.

We are confident about the described information swap between Graves 21 and 22 as it appears to be one of few inconsistencies between the initial documentation of the archaeological findings by N. Pantusov (principal archaeologist excavating Kara-Djigach) and their later analysis by G. Debets. In addition, our genetic sex determination of individual BSK002 (presumably interred in Grave 22) revealed that the individual was a male, which matches the published morphological analysis by G. Debets. Instead, Grave 21 was reported to contain the remains of a female. The dates of both Graves 21 and 22 are known and shown on Figure 1 of the main text, which contains a re-adapted map of the “Chapel 1” designated area of the Kara-Djigach archaeological site. This map was created by N. Pantusov during initial excavation of the site (original map shown below) and dates were assigned to all graves according to inscriptions on their associated tombstones. Both Graves 21 and 22 date to the epidemic year 1338/1339 and, therefore, discrepancies between the two could not be addressed with radiocarbon dating. The tombstone inscriptions for both graves were published in the analysis by D. Chwolson in 1890¹⁵, and their translations are included below. To our knowledge, the locations of the original tombstones are unknown and, therefore, their photographs are unavailable.

Grave 21: *“It was in the Year 1650 [=1338-9 CE], the year of the Hare, in Turkic Taviškan. This is the tomb of Qamt'ā, a faithful woman”*

Grave 22: *“It was in the Year 1650 [=1338-9 CE], the year of the Hare, in Turkic Taviškan. This is the tomb of Sargis the student”*

Figure: Picture of “Chapel 1” designated area in Kara-Djigach. The picture was taken from the diary of N. Pantusov, the principal archaeologist that excavated the site between 1885 and 1892 CE (picture credits Philip Slavin).

Furthermore, to aid our identifications, we have now requested photographs of all sampled skull specimens from the *Kunstkamera* museum, which were not available to us during initial sampling. As a result, we have noticed an additional inconsistency with regarding to specimen BSK001. BSK001 was retrieved from a conjoined burial of a man and a woman, likely a family tomb, designated as Graves 19 and 20. Based on our genetic analysis and the associated tombstone inscription, we had initially considered that our sampled individual comes from Grave 19. However, the picture of the sampled specimen is indicated as being from Grave 20 (see specimen picture below). We are currently unsure whether the tombstone swap between the two burials occurred at the time of excavation or at the time of the burial. Both Graves 19 and 20 are associated with the epidemic year 1338-9.

Figure: Skull from individual buried in Grave 20, which belonged to a female individual named Baçaq. The Grave number is indicated on the individual's mandible, in addition to the *Kunstkamera* museum ID (176/4)

Similarly, I was not able to track grave 19 on pages 6 and 7 of the SI material (this was possible for the other 4 graves analyzed from Kara-Djigach).

We thank Referee #1 for this important comment. As mentioned above, we have come to realise that the individual analysed in this study was actually buried in Grave 20, which formed a family burial with Grave 19. Our realisation does not change the individual's designation, which was already described in the previous version of our paper in a supplementary section named "Information on individuals from Kara-Djigach analysed in this study". The section begins on page 16 of our supplementary information document. The updated part relating to Grave 20 is included below for convenience.

"BSK001

The analysed tooth specimen belonged to an individual buried in Grave 20, which together with the adjacent Grave 19, represent two conjoined burials, described by N. Pantusov as a 'family tomb' (AIIIMKRAN, Fund 1, Inventory 1, File 40b, fols.12r). The skeleton height of both individuals was measured at 146.7cm. No artefact was identified within the burials. Both individuals were interred into a bricked pit, rather than individual coffins (full description in AIIIMKRAN, Fund 1, Inventory 1, File 40b, fols.12r-v). Based on cranial suture closure, BSK001 was classified as matusus (40-50 years) in age.

It should be noted that the inscription initially associated with Grave 20 states "In the Year 1650 [=1338-9 CE], [it was the year of the] Hare. This is the tomb of Yoḥannan, a faithful man". However, the human genetic analysis associated with specimen BSK001 revealed the individual to be of female genetic sex (see Supplementary Table 3), which raises uncertainty regarding the initial tombstone identification or documentation. Moreover, a 1948 study of 17 skulls specimens from Kara-Djigach by G. Debets, identified the individual from Grave 20 as female¹⁶. Most likely, this discrepancy derives from the fact that the above-mentioned inscription is to be associated with Grave 19 and not 20. Conversely, the correct inscription associated with Grave 20 is that initially associated with Grave 19. Indeed, the tombstone inscription associated with Grave 19 states "In the Year 1650 [=1338-9 CE], [it was the year of the] Hare. This is the tomb of Bačaq, a faithful woman". Importantly, N. Pantusov described the skull from Grave 19 as 'decayed and disintegrated', which does not reflect the currently preserved state of the skull from Grave 20 (Supplementary Fig. 3), implying that the skulls, unlike headstones, were not mixed up. The fact that Graves 19-20 are a family burial may explain the erroneous inversion of the graves and their associated tombstones during initial in situ documentation during the August 1886 excavations. Alternatively, it is possible that the tombstones may have been inversely misplaced during the burial process at the time of the 1338-9 epidemic.

If this interpretation of the erroneous inversion of the graves and their associated tombstones is correct, then it appears that the analysed tooth specimen associated with the skull from Grave 20 belongs to a woman named Bačaq. The skull associated with this individual is currently held at the Kunstkamera and its accession reference number is 176/4 according to the hand-written catalogue of the museum's craniological collections (by Ye. Zhiron), file 176 (previous accession number 970 according to ref.¹⁷). The transition of the skull from the Kara-Djigach cemetery into the IAC in the year 1888, and its subsequent deposition at the Kunstkamera are documented in AIIIMKRAN, Fund 1, Inventory 1, File 40b, fols.12r-v, 84v-85r, 119r. The original Syriac texts of the inscriptions of both burials within Grave 19-20 are printed in ref.¹⁵. The original stone associated with Bačaq's grave is at the Hermitage (Hermitage SA no. 14432). The current whereabouts of the stone associated with the male individual, Yoḥannan, are unknown."

Additionally, I noticed that the BSK001/003 branch is the first to split on the MMC tree shown in Fig3b, and remains separated from all descending lineages with some decent internal branch. This, and the differentiation between branches 1-4, does not look exactly the same in Fig4a though. Can the authors clarify whether the same data set (site-wise) was used in both analyses and elaborate why do we see such differences?

We thank the Referee #1 for raising this aspect, whose clarification is of importance. The classic binary rooted time-trees computed in BEAST2 do not allow for an accurate resolution of trifurcations or polytomies and, therefore, tend to produce branching patterns that do not signify substitution differences between lineages. Rather, these patterns signify differences in their computed mean (or median) divergence times. On the contrary, substitution trees, for example those using the maximum likelihood approach, provide more accurate representations of branch topologies that reflect evolutionary events in non-recombining species phylogenies such as that of *Yersinia pestis* (see Fig. 2 and Extended data fig. 5). Here, we have used BEAST2 to estimate a date range for the divergence of Branches 1-4, using the BSK001/003 genome as a proxy, as we have demonstrated that it represents the node that gave rise to this polytomous event. We find that the divergence time interval of BSK001/003 as well as that of Branches 1, 2, 3 and 4 overlap despite their seeming branching differences on the displayed MCC tree, hence supporting a common split time for these lineages. Moreover, we find that the posterior divergence estimate of Branch 1 is much narrower compared to those of Branches 2, 3 and 4, which is not unexpected as these lineages do not contain any ancient calibration points (see updated version of Figure 3). The dataset used for the molecular dating analysis comprises a subset of genomes (n=167) and consequently variants (1405 SNPs) compared to the complete comparative dataset used for the maximum likelihood tree shown in Figure 2 and Extended data fig. 5 (n= 251 and 2,960 total SNPs). Specifically, our molecular dating analysis contains Branch 1-4 genomes (modern + ancient) and those comprising the 0.ANT3 lineage (see Methods lines 616-620). For clarification, we have now also computed a substitution tree with the reduced dataset using RaxML and demonstrate that the polytomy is retained (see below, now appears as Supplementary figure 10).

Moreover, we have explored the software TreeTime¹⁸ as a potential alternative method to BEAST2, as it employs an approach that infers maximum likelihood time-trees that can retain polytomies in a manner consistent with the tip dates of sampled isolates. Based on this new analysis with TreeTime, our results further support an early-14th-century date for the Branch 1-4 polytomy (tree inserted below, now appears as Supplementary figure 11). We caution, however, that TreeTime is a software designed primarily for studying present-day rapidly evolving populations, such as viruses, and does not allow for extensive model parametrisation and for the inclusion of prior uncertainties, such as tip date intervals, as is possible using more complex Bayesian approaches. Given the wide expected readership of our paper we have now added a clarification of these interpretation aspects in L161-166.

E Conclusions. The conclusions are sound and not overtone. I wish the authors address the points listed above, as some may even strengthen their current evidence.

We thank Reviewer #1 for their overall positive assessment of our conclusions. We agree that their comments have helped improve the quality and clarity of our paper.

G References are appropriate.

H Clarity and context. The manuscript is clear and well written, but at times un-necessarily lengthy. This may particularly apply to the sections on the archaeological context and human DNA analysis. I have noticed a number of typos but nothing serious (eg Black Deah on line 179, page 5).

We thank Referee #1 for their positive assessment of our manuscript's conclusions. Following their suggestion, we have now combined the archaeological context and human genomic sections in the main text and present those in a more concise manner. The new section now appears with title "*A mortality crisis in 14th-century central Eurasia*" and can be found in L.68-92. Moreover, we have added a new supplementary section with an expanded version of the human genomic analysis, including a number of the reviewer's suggested analyses (Supplementary Information 3). Finally, we have also integrated a new line of evidence, where we used toponyms identified on burial tombstones to investigate the diversity of the Kara-Djigach and Burana communities.

Additionally, the above indicated typos have been corrected.

Referee #2: Mark Achtman (Remarks to the Author):

A. The manuscript describes two genomes with a summed coverage of about 9fold plus a third with low coverage from graves dated by the gravestones from 1338. According to core genome SNPs, the two genomes are located at node N07 defined by Cui et al as the founding node of major splits in the *Y. pestis* genealogy that resulted in five modern major branches. This dates the split into those branches to 1838 or earlier, and locates predecessors of the Black Death of 1848 eastwards to Kyrgyzstan.

We thank Prof. Achtman for this overall assessment of our study.

Below we provide an outline of our study's main conclusions, which we have deduced from three main lines of evidence:

(1) The aDNA identification of *Y. pestis* infections in individuals from Kara-Djigach, whose graves were dated based on tombstone inscriptions, indicates an involvement of plague in the mortality crisis previously shown to have affected the communities of the Chu Valley region in the years 1338-1339 CE.

(2) Our analysis of two ancient *Y. pestis* genomes from Kara-Djigach shows their placement on the N07 node of Branches 1, 2, 3 and 4, commonly associated with the onset of the Second Plague Pandemic by both geneticists^{11,12,19,20} and historians²¹⁻²³. Such finding shows that the Black Death's founding strain was epidemically active in northern Kyrgyzstan by the years 1338-1339 CE.

(3) To infer a tentative source for the Kara-Djigach strain, we analysed modern-day lineages that are branching closely ancestral to this genome. We show that these lineages are today geographically restricted to the Tian Shan region, located in close proximity to the studied archaeological sites. This result suggests a proximal source for the Kara-Djigach epidemic, likely within plague foci located in the bordering regions of modern-day eastern Kazakhstan, eastern Kyrgyzstan and the Uygur Autonomous Province (Xinjiang, China).

B. One of the primary authors, P. Slavin, published citation 11 in 2019 which pointed out all the anthropological conclusions summarised here and predicted that the two cemeteries investigated here might contain the predecessors of *Y. pestis* that are now described here. That paper also included a histogram of the dates on the grave inscriptions which seems to have been reproduced with minor modifications in the figures in this manuscript, but is not attributed to the earlier publication. Historians tend to publish the same data more than once but this smacks of lack of originality.

We thank Prof. Achtman for this comment, which gives us the opportunity to clarify differences between the two datasets. While both studies used tombstone inscriptions to explore the archaeological context of Kara-Djigach and Burana, there are a number of differences between the two datasets. First, the dataset of Figure 1 in this study is based on 467 dated tombstones (including 118 from the plague years), in contrast to 439 tombstones

(including 114 from the plague years) used in in the earlier paper. Second, in the present study we further explored the inscriptions to estimate a sex ratio of deaths on an annual basis, a level of detail that was not reflected in the earlier study. Third, we have now further investigated all surviving tombstones for indications of toponyms, to shed light on possible places of origin for the inhabitants Kara-Djigach and Burana, an analysis that has revealed possible mobility patterns in these communities (see Supplementary section 2). Fourth, in his earlier paper, P. Slavin did not analyse the tombstones alongside other archaeological materials (such as grave goods and coins), which provide independent lines of evidence for the studied sites. Lastly, all the information retrieved from excavated artefacts was not previously compared with the chronology of excavated graves (which differs from that of retrieved tombstones) – something that has been solely examined in the present study. To make these distinctions clearer, we have now revised the archaeological context section in the main text of our study. This new section is entitled “*A mortality crisis in 14th-century central Eurasia*”.

C. The data and methodology are fine. However, the methods do not differ dramatically from those in multiple publications by Spyrou in the last few years. And the presentation is so methodical that the manuscript is not exciting even though the data and conclusions are. Basically, they have found genomes which map to the founding node of a major split in phylogeny, and are 1-2 SNPs earlier in that phylogeny than any previous genomes. The rest are details, many of which are better placed into supplementary material or methods.

Given the challenges associated with the present ancient DNA dataset and to aid an easier assessment by the readers, we had initially opted to transparently present our analysis pipeline within the main text. We agree, however, that part of these details may be better placed in other sections of the manuscript. We have now moved the human genomic analysis to Supplementary Information section 3 and placed a number of analytical details of the bacterial genomic portion to the Methods section and Supplementary Information section 4.

The section on the PCA plots of human genomics is unimportant and uninformative and belongs in supplemental text.

We agree that this section was interrupting the manuscript's flow and, therefore, we have placed most of its contents with some additional analyses to Supplementary Information 3. We do, however, consider that our human genomic analysis offers important information regarding the genetic make-up of the Kara-Djigach and Burana communities. Interpreted alongside the newly examined archaeological context, these data have revealed a diverse profile for the communities of the Chu Valley, where mobility for trade and immigration purposes played an important role to their composition and subsistence. Such findings may have important future implications when building models of plague dispersals across Eurasia using additional historical, archaeological and genomic data.

The figures are superb but the text and message would benefit by being dramatically tightened and shortened.

We thank Prof. Achtman for the positive evaluation of our figures. We have now made efforts to condense the text by moving the human genomic analysis to our paper's Supplementary Information and by placing a number of analytical details of the bacterial genomic portion to the Methods and Supplementary Information sections.

I am unenthusiastic about the skyline plots used for Beast2 analyses. A few years ago there was an enlightening paper by Eduardo Rocha showing that bacterial coalescent pipelines calculated with Beast all looked the same, and were likely artefactual (<https://doi.org/10.1093/molbev/msw048>).

In our study, we have used the Bayesian skyline demographic model in BEAST2 merely for estimating divergence time intervals for the N07 polytomy. We have refrained from

presenting estimates of the bacterium's effective population size through time, particularly since we only use a subset of the known diversity for this analysis (0.ANT3 and Branches 1-4). We thank Prof. Achtman for recommending the paper by Lapierre et al., which we were not aware of, as it will be a useful guide for a number of our ongoing studies. We agree that several aspects affect the results presented in skyline plots, including sampling biases and the assumption of a single panmictic population, which is an unrealistic view of the global population history of *Yersinia pestis*.

I am unenthusiastic about the calculations on the numbers of genomes per phylogenetic branch. This is a function of sample bias by the author, who used an arbitrary collection of genomes rather than the complete databases available within EnteroBase (<https://genome.cshlp.org/content/30/1/138.short>), which is not even cited.

We thank Prof. Achtman for this comment. Our description of the number of genomes within each phylogenetic branch has not been used to formulate any of the central conclusion in our study. Instead, we only use this calculation to make a statement of increased worldwide frequency for strains belonging to Branches 1-4, an observation that is reflected both in our comparative dataset and within EnteroBase. We acknowledge that EnteroBase is the most complete *Y. pestis* genome database to date, and, thus, we have now cited it within this section (L172). Of note is that EnteroBase contains a large number of genomes used to examine the microdiversity of *Y. pestis* within single regions, such as Madagascar, Brazil and the USA, and sampling in these locations has been performed within narrow temporal intervals. In addition, the majority of genomes within EnteroBase belong to the lineage 1.ORI (Orientalis serovar) that arose during the last 150 years. Therefore, while a seemingly large dataset, inclusion of these genomes would be of limited benefit to our comparative genomics analysis, which focuses on events that occurred >600 years ago.

In addition, in our analysis of phylogenetic diversity (Extended data figure 7) we had initially accounted for a higher number of genomes within Branches 1-4 by down-sampling that clade for computing our estimates. To compliment this approach, we have now used a bootstrapping method to assess the sensitivity of our results with respect to sampling and phylogenetic uncertainty. We have further updated this analysis by using all 1,000 bootstrap trees as well as a sampling with replacement approach to retrieve confidence intervals for our diversity estimates (see Methods section L585-603). Our results appear largely consistent with those presented in the initial version of our paper.

D. The statistics look fine. But what I was missing was a very simple overview of the coverage and numbers of reads for each of the nucleotides at the crucial SNPs distinguishing the two new genomes from other ancient and modern genomes.

We thank Prof. Achtman for this suggestion. We have now added a SNP table showing the allelic status of all historical *Y. pestis* genomes (including BSK001 and BSK003) with regard to diagnostic SNPs defining Branches 1, 2, 3 and 4, as well as those leading to the N07 polytomy. This table appears as Supplementary table 18.

E. The conclusions are simple. Ancient genomes have now been found in Kyrgyzstan which predate other ancient genomes that caused the Black Death in Europe, and this supports further investigations elsewhere in central and Eastern Asia. These conclusions are robust, valid, novel, interesting but not surprising. They were basically predicted by Morelli et al and Cui et al prior to ancient DNA work.

We thank Prof. Achtman for the positive assessment of our study's conclusions and impact. We are familiar with the seminal papers by Morelli et al., 2010²⁴, and Cui et al., 2013¹⁹, which are cited multiple times in our paper. Both studies, using modern genomic data, seem to put forward a hypothesis that *Y. pestis* originated in China or its vicinity and that its pandemic eruptions were associated with independent dispersals from this wider region. In particular, the most recent paper¹⁹ discusses the Qinghai-Tibet Plateau as the possible focal

point, given its vicinity to ancient trade routes and the rich modern *Y. pestis* diversity existing in this region. Through the use of ancient genomes as well as previously published modern genomic data, in our current paper we extend these hypotheses and rather suggest the central Eurasian Tian Shan region as a possible birthplace for the Second Plague Pandemic. We further suggest that the presented aDNA and archaeological data provide key evidence to elucidate a clear timing for the proposed origin. We have now created a separate Supplementary section (Supplementary Information 1) to describe in more detail all existing theories on the Second Pandemic's origins.

F. Tighten the manuscript and lighten the text and approach. It is too heavy at the moment.

We agree with this point. We have now made efforts to condense the text by shortening the introductory section, by moving the human genomic analysis to the Supplementary Information and by placing a number of analytical details bacterial genomic portion to our paper's Methods section.

G. The authors need to give even more credit to Slavin's 2019 paper.

We thank Prof. Achtman for this suggestion. We have now added a number of additional citations of P. Slavin's paper across the main text of our study.

H. Good abstract. Nice synthesis in the text. Less defensiveness would be better.

We thank Prof. Achtman for the overall positive assessment of our paper's structure. In our study we have paid particular attention to credit all previous works that dealt with the origins of the Second Plague Pandemic, especially with regard to both genetic and historical data. We agree that we have shown a number of early and recently presented hypotheses that are contrasted by our results. Our results unambiguously show an association of the 1338/1339CE Issyk Kul outbreak with *Y. pestis* and with the initiation of the Second Plague Pandemic, a notion that has been denied in a number previous works. To clarify this further in a balanced way, we have now expanded this section to present a more detailed overview of previous studies (Supplementary Information 1) and in order to emphasize the aspects that we confirm and contrast with our presented dataset.

Reviewer #3

A) Summary of the key results

The geographic origins of the second plague pandemic (the Black Death) have been subject to endless speculation, with some of the earliest evidence found at the shores of Issyk Kul in what is now Kyrgyzstan. This work presents data generated and screened from seven individuals excavated at two cemeteries in the region. Following classification, this data recovers evidence of the presence of *Y. pestis* in three cases (two of analytical quality following capture enrichment) as well as human genome SNP capture data generated at sufficient quality for two individuals. Despite the low number of genomes, the generation of *Y. pestis* genomic data from these sites in their associated epidemic context immediately preceding the medieval Black Death contributes to the inference that the agent (strain) of the subsequent Black Death likely emerged in a local context confirming previous suggestions based on historical accounts and archaeology of an entrance of the Black Death from East to West Eurasia. The authors suggest their work provides strong evidence for a central Eurasian origin of the Second Plague Pandemic. This is a valuable study, moving away from the "Eurocentric" view of plague the authors claim has existed until now.

We thank Referee #3 for this overall positive assessment of our study. Below we enclose a point-by-point response to all their suggestions.

B) Originality and significance: if not novel, please include reference

Prior work, based on historical accounts has suggested an origin for the second plague pandemic in the South-West of Russia eg. Barker et al, *Speculum* 96, 97-126 (2021) and previous work by some of these authors hypothesised an Eastern European origin for the second plague pandemic using ancient DNA and phylogenetic methods. The archaeological sites under consideration have also been linked to epidemic 'pestilence' events (eg. Supp text 1 and Figure 1d). The work here can therefore be seen as confirmatory for these hypotheses rather than a de novo insight. However, the observation of the lineage of *Y. pestis* present at these sites, and the link to the major lineage giving rise to the medieval Black Death provides valuable new information to enrich our understanding of *Y. pestis* phylogeographic patterning and genomic diversity.

We thank Referee #3 for this comment, which gives us the opportunity to clarify how our work informs previous hypotheses on the Second Pandemic's proximal and more distal origins. Here, we indeed conducted a hypothesis-based study given the rich corpus of pre-existing work focusing on the origins of the Second Plague Pandemic that has been published over the last 200 years (on archaeology, historiography, plague ecology and genetics). To date, the hypotheses that had been put forward were largely inconsistent with regard to the geographic source of the pandemic, ranging from locations in western Eurasia^{25,12}, central Eurasia^{21,22,26,27} and eastern Asia^{19,23,24,28,29} (see a more detailed overview now in Supplementary Information 1). In addition, recent studies have challenged the idea of the onset of Second Plague Pandemic being a 14th-century phenomenon by proposing an early-13th century origin and spread, supposedly facilitated by the Mongol expansions across Eurasia^{8,23,30}. While our genetic findings lend support to pre-existing hypotheses suggesting central Eurasia^{12,13} or even specifically the Tian Shan region^{21,22} as a place of origin, we clearly contrast others with regard to alternative geographic sources^{25,12,19,23,24,28,29} and the timing^{8,23,30} of the pandemic's onset. We are aware of the study by H. Barker mentioned by Referee #3; however, this study does not claim to be dealing with the origins of the Second Plague Pandemic. It rather focuses on events that unfolded between 1343 and 1347 CE in the Black Sea region to elucidate the spread of plague from the Tatar lands into southern Europe and the Middle East.

I do however have queries on the approach used as outlined below:

C) Data & methodology: validity of approach, quality of data, quality of presentation

Lines 173-188 and methods: For the phylogenetic tip-dating it is stated that two models are tested (coalescent skyline and birth-death). These are models with quite different assumptions about demographic history. Given we are estimating a polytomous event the choice of coalescence model prior is important to justify. Were other models tested and did they also provide similar estimates? I note that coalescent constant approaches were rejected based on previous work (line 630). How were these particular models therefore selected to take forward and why was a GTR substitution model prior used?

We agree with Referee #3 that our model choices were not sufficiently justified within our methods section. We have now revised this portion of our analysis, to include a thorough testing of all applied models. Overall, the central conclusions of our study do not change as a result of these new analyses.

We now shifted all our analysis to the most recent version of BEAST2 (v6.6), which includes a number of bug fixes and updates compared to previous versions (for version history see <https://github.com/CompEvol/beast2/releases>). Prior to molecular dating, we used the *jmodeltest*³¹ (2.1.10), and identified that a transversion model (TVM) would be the substitution model of best fit for our dataset. TVM was, therefore, implemented in BEAST2 by using a GTR model with four gamma rate categories and the AG substitution rate parameter fixed to 1.0 (as implemented previously³²). Within BEAST2, we compared a number of coalescent demographic models (coalescent constant size, exponential population and skyline), which have often been used in previous studies for time-calibrated

phylogenies and for divergence estimations using ancient tip dates (see for example³³⁻³⁶). In addition, we also used the birth-death skyline model that has started to gain traction in recent years for the reconstruction of time-calibrated ancient pathogen phylogenies^{32,37-39}. We used PathSampling implemented in the model-selection package of BEAST2 to assess the best suited demographic model for our dataset. We found “strong support” (see definition in⁴⁰) for the choice of coalescent skyline model compared to other tested models (with a log Bayes factor of 8.4 when compared to the second-best model). Therefore, we restricted all our subsequent molecular dating analyses to this model. Our new results are now shown in an updated version of Figure 3, as well as in Supplementary tables 19, 20 and continue to support an early-14th-century divergence of *Y. pestis* Branches 1-4.

In addition, given the results presented are highly concordant one explanation could be over fitting of the prior to the data. Again, this is of particular concern given likely uncertainties in the tree building step due uncertainty at the polytomy basal to the tree and the possibility for a few erroneous SNPs in ancient samples to strongly impact posterior rates. The authors should provide the posterior tree heights and clock rates obtained when running the analysis without the data (sampling from the prior) to confirm their inference is robust to the choice of prior model specified and be more explicit about why these models were chosen.

We agree with Referee #3 that we should have shown a molecular dating analysis without data (sampling from the prior) to confirm the robusticity of our results. We have now implemented this analysis using the model of strongest support (coalescent skyline). For this we ran two chains of 600 million states each and combined the resulting log files using a 30% burn-in. Our results indicate convergence of all parameters (ESS>200) and show a mean divergence estimate of ~19,000 years before the present without the data and the substitution rate to be forming a uniform distribution that is fully overlapping with the specified uniform prior (1.E-3-1.E-6 subs/site/year). Both of these estimates differ considerably from those obtained using a data-informed analysis (see figure below). Therefore, we are confident that our data appears most informative for the presented molecular dating analysis and that the priors alone are not strong enough to be driving the obtained posterior estimates.

“Supplementary Fig. 13 – Sampling from the prior analysis to assess robusticity of posterior estimates. The analysis was performed using BEAST2 v6.6. Posterior intervals are shown in brown for the data-informed analysis, whereas estimates after sampling from the prior are shown in purple.”

Figure 2: Given the identical position of BSK001/BSK003 and claim of conserved genomic diversity (line 143) and joint analysis suggested by line 151 I do not follow what gives rise to the long tail in the diagnostic SNPs presented for BSK001/BSK003 in Figure 2C relative to BSK001. Do the diagnostic SNPs include some that were previously claimed as artefactual (lines 139-143). Were BSK001/003 and BSK003 considered both jointly and separately in this analysis? I'd appreciate additional clarification in the methods.

The combined BSK001/003 genome does not appear to have a long tail of diagnostic SNPs in Figure 2C. Instead, this genome supports 0 shared variants on Branch 1, which further confirms its ancestral placement. As shown in Figure 2C and Extended Data Fig. 6, BSK001 and BSK003 were also treated separately for the diagnostic SNP analysis. In several of the comparisons, BSK003 appears to have a long tail of diagnostic SNPs indicating missing data in the respective positions, given the lower mean coverage of this genome (2.8-fold). As indicated in the legend of Extended Data Fig. 6: "*Confidence intervals indicate uncertainty due to the presence of missing data (Ns) within the variant calls of the respective genomes.*" To clarify this further, we have now compiled a table (shown as Supplementary table 18) indicating the genomic positions of all diagnostic SNPs and their respective alleles in our entire *Y. pestis* dataset (including BSK001, BSK003 and the combined BSK001/003).

Figure 3: What % of the posterior tree distribution support the basal placement of BSK001/003? Can this polytomy be considered as resolved using a time tree approach?

We thank Referee #3 for this comment, which allows us to clarify important motivational aspect for our molecular dating analysis. We have not attempted to resolve the presented Branch 1-4 polytomy using a time-tree approach. We consider that the classic bifurcating time-calibrated phylogenies produced by BEAST2 are not appropriate for resolving polytomies. We view our maximum-likelihood substitution tree to be a more robust approach for resolving the presented polytomy, a phylogenetic topology and has been previously demonstrated for *Y. pestis* by a number of papers (first here¹⁹ and in a number of subsequent studies, for example^{11,12,20,35}). As indicated in our Introduction:

"In recent years, comparisons between ancient and modern Y. pestis genomes have shown the Black Death to be associated with a star-like emergence of four major lineages (Branches 1, 2, 3 and 4)^{12,19}, whose descendants are currently dispersed among rodent foci in Eurasia, Africa and the Americas."

Here, we have used BEAST2 to date the divergence of *Y. pestis* Branches 1-4 using the split-time of BSK001/003 as a proxy for their direct ancestor. While we realise the shortcomings of this approach, we still consider BEAST2 as the most appropriate tool for performing molecular dating analyses using ancient DNA datasets, given that it can account for tip-date uncertainties (through the use of radiocarbon date intervals) and permits extensive model parametrisation (for example, allows for testing a range of demographic and substitution models).

To complement our approach, we have now tested the program TreeTime¹⁸, which infers maximum likelihood time-trees that can retain polytomies in a manner consistent with the tip dates of sampled isolates. We caution that this approach was not developed to fully accommodate ancient DNA data and, therefore, an implementation of tip-date intervals (such as radiocarbon dates) is not possible. As a result, all ancient genomes were incorporated in this new analysis using their midpoint ages as calibrations. This is of particular concern for specimens with wide temporal intervals, which in our dataset appear to be several genomes dating between the 15th and 17th centuries (see Supplementary tables 19, 22). Nevertheless, based on this new analysis with TreeTime, our results further support an early-14th-century date for the Branch 1-4 polytomy (tree inserted below, now appears as Supplementary figure 11).

D) Appropriate use of statistics and treatment of uncertainties

Lines 181/612: A prerequisite of phylogenetic tip-dating is the presence of temporal signal in the alignment. The robustness of the signal should be assessed with provided p-value following date randomization. The temporal regression plot should also be provided.

We agree with Referee #3 that we should have shown additional evidence of temporal signal in our dataset beyond a correlation coefficient metric. We have now incorporated the full regression of root-to-tip distance against specimen age in our study appearing as Supplementary Fig. 9. Moreover, we have chosen to use the most recently developed method BETS⁴¹ for temporal signal analysis. The BETS approach uses PathSampling to compare marginal likelihood estimates between two model set-ups, one assuming an

isochronous population and one including the true sampling dates (heterochronous model). For our set-up we used our best supported demographic model (coalescent skyline) and PathSampling was run using 50 steps of 20 million iteration each. Subsequently, the resulting likelihoods were compared in the format $BF = M_{heterochronous} - M_{isochronous}$ to retrieve the log Bayes factor (BF). The BF was then interpreted as indicated before^{40,41}, whereby a value > 3 was considered as “strong support” for temporal signal in the dataset. Our resulting BF was 129.33, supporting the presence of temporal signal in our dataset (Supplementary table 20).

Supplementary table 2: many more reads are assigned to the *Y. pseudotuberculosis* complex than *Y. pestis stricto*. Were other *Yersinia* species present at reasonable abundance in the shotgun data? How may this have impacted the results e.g. cross mapping or genus ubiquitous capture? Extended Data Fig 4 suggests such an effect is reasonable for BSK007 (~17.2% post capture). It would be useful to see the equivalent figures for BSK001 and BSK003 or the data break down on how reads were assignments post capture. I raise this due to comments on the high number of multi-allelic sites in BSK001 and BSK003 (lines 138) which may arise due to the presence of reads deriving from related *Yersinia* species which can be highly diverse. The authors should provide the break-down of read assignments to other *Yersinia* sp. reported by MALT and excluded by the MALT filtering (extended Data Figure 5).

We thank Referee #3 for this comment. We indeed observe an impact of exogenous reads mapping against the *Y. pestis* genomes of both BSK001 and BSK003, as demonstrated by the increased numbers of multi-allelic sites (also called ‘heterozygosity’) identified in both genomes. Such an effect is unlikely to be determined with our capture reagent and is more likely associated with the preservation environment of these specimens, which were excavated 150 years ago and have been kept in several storage facilities ever since. In a previous study, we have shown variations with respect to ‘heterozygosity’ estimates in *Y. pestis* genomes isolated from the same archaeological site, indicating specimen-specific patterns of background contamination rather than an overall reduced specificity of our capture reagent¹². Here, we used metagenomic filtering of the higher-coverage genomes BSK001 and BSK003 in an attempt to retain *Y. pestis*-specific reads for our genomic reconstruction. For this, we used MALT, which employs the naïve lowest common ancestor (LCA) algorithm, first introduced in MEGAN4⁴². Given this approach, reads are assigned to the lowest possible taxonomic rank based on their specificity. Previous tests for the development of the metagenomic pipeline HOPS (which uses MALT as the classifier) showed that simulated *Y. pestis* reads tended to be preferentially assigned to their corresponding complex node rather than the species node⁴³. Moreover, *Y. pestis* has been demonstrated to be a clone within the diversity of *Y. pseudotuberculosis* and that the two taxa have an identity of >97% with respect to protein coding genes⁴⁴⁻⁴⁸. Given the above-described aspects, it is unsurprising that our metagenomic analysis resulted in a large number of reads summarised under the *Y. pseudotuberculosis* complex node, the common ancestor to *Y. pestis* and *Y. pseudotuberculosis*, with no superior identity to either of the two species. Nevertheless, we find that the number of reads summarised under the *Y. pestis* species node is vastly higher than those on *Y. pseudotuberculosis* (Supplementary table 11).

We have now provided an overview of reads matching different taxonomic nodes for all captured libraries of specimens BSK001, BSK003, and BSK007. We have restricted this overview to nodes with ≥10 summarised reads (Supplementary table 11).

E) Conclusions: robustness, validity, reliability

Lines 191-198: Given some uncertainty in the phylogeny the point estimates of FPD and MPD feel overly precise. Indeed, with different genomic sampling these values may shift potentially quite a lot. In addition, FPD has been shown to be sensitive to rate variation in the tree eg. see Ritchie et al. Diversity & Distribution (2021) 27,1 which is commented on as a

property of *Y. pestis* datasets eg. line 176. Given randomly sampling was performed for the modern subsampling it would seem prudent to assess the distribution of FPD and MPD values obtained for the contribution of Branch 1-4 to overall phylogenetic diversity following a leave one out approach and provide these values as a range.

We agree with Referee #3 that we should have accounted for phylogenetic uncertainties in our diversity analysis. We have now adapted our approach and used all 1,000 maximum-likelihood bootstrap trees inferred in our phylogenetic reconstruction. In addition, we used a sampling with replacement method to account for sampling biases in our dataset.

As discussed in the previous version of our paper, in our dataset, 64% (130/203) of modern *Y. pestis* strains belonged to Branches 1-4. Based on our adapted approach, we estimate that this lineage represents about 40% of the overall phylogenetic diversity of *Y. pestis* (MPD ratio: 41%, 95%PI: [35.3,46.4]; FPD ratio: 35.9%, 95%PI: [31.6,39.5]). Our subsampling of Branches 1-4 to the same number of strains as in Branch 0 gave comparable results (MPD ratio: 36.8%, 95%PI: [32,41.9]; FPD ratio: 33.9%, 95%PI: [29.4,37.7]).

For convenience, we have inserted below our adapted methods section and a graphic summary of our new approach:

*“In order to estimate the proportion of modern *Y. pestis* diversity descending from BSK, we used the R package picante⁸⁸ to compute the mean pairwise distance (MPD) and Faith’s phylogenetic diversity index (FPD)⁸⁹ from the reconstructed ML substitution tree. Measures made on a subset of the tree corresponding to the subclade descending from BSK (Branches 1, 2, 3 and 4) were compared to that of the complete *Y. pestis* phylogeny. In both cases, only modern strains were included in the calculation. We used a bootstrapping approach to assess the sensitivity of our results with respect to sampling and phylogenetic uncertainty⁹⁰. For each of the 1,000 RAxML bootstrap trees, we randomly resampled modern strains with replacement, and only kept branches of the tree corresponding to sampled strains. Diversity measures were performed for each of the obtained resampled bootstrap trees, from which median estimates and 95% percentile intervals were derived.*

In addition, in order to assess the potential impact of uneven sampling among branches (Branches 1-4 contain 130 modern strains while branch 0 contains only 73), we repeated the same analysis but adding an initial step intended to equalize the number of genomes in both parts of the tree. We subsampled Branches 1-4 to the same number of strains as in Branch 0 using sequence clustering within Branches 1-4 in order to obtain representative subsamples. Hierarchical clustering was performed based on pairwise phylogenetic distances (derived from the ML phylogenetic tree) and the resulting tree was cut in order to define 73 clusters (R functions hclust⁹¹ and cutree). For each bootstrap tree, only one randomly selected strain from each cluster in Branches 1-4 was kept before performing the resampling procedure and computing diversity measures.”

F) Suggested improvements: experiments, data for possible revision Appropriate use of statistics and treatment of uncertainties

The authors consider the plasmid coverage of their strains. It would seem that this is also useful information to leverage in terms of understanding the phylogeography and age of divergence of the precursor to the medieval Black Death. I would suggest not using this information in the downstream analysis is a waste of valuable data. In particular, were any attempts made to conduct phylogenetic inference on the shared plasmids, which presumably have accumulated mutations since the split (perhaps resolving the polytomy?) or indeed to use these to date the resultant phylogeny? Do BSK001 and BSK003 carry exactly the same complements of SNPs in each plasmid?

We agree with Referee #3 that an analysis of plasmid variants could shed light on previously unrecognised phylogenetic patterns. As suggested, we have now leveraged the retrieved coverages in BSK001, BSK003 and BSK001/003 as well as those of previously published datasets^{11,12,20,49-53}, to identify patterns of plasmid variation in historical genomes. For this analysis, each of the *Y. pestis* plasmids, pPCP1, pMT1 and pCD1 was treated separately. For mapping and SNP calling, we followed the same criteria as the ones used for the chromosome (see Methods). Our resulting SNP table was ascertained to variants identified within our ancient dataset (SNPs identified uniquely in modern genomes were excluded). While we do identify plasmid variants in some historical genomes, including few singletons and some shared diversity among 16th-18th century genomes, we do not find sufficient variation to distinguish 14th-century diversity among any of the three plasmids. Specifically, all non-reference alleles identified in 14th-century genomes (including BSK001, BSK003 and BSK001/003) were shared among all historical genomes as well as with the modern genomes KIM5 (Branch 2) and 0.PE4-Microtus91001 (Branch 0) that were used for comparison. Therefore, we conclude that plasmid variation could not inform our phylogenetic reconstruction with regard to the early stages of the Second Plague Pandemic. A table describing all plasmid SNPs identified in historical genomes now appears as Supplementary table 15 of our paper.

Some additional supplementary figures would be most valuable to support the validity of the *Y. pestis* genomes obtained eg. visualised edit distances to other *Yersinia* species and the root-to-tip correlation with accompanying significance.

We thank Referee #3 for this suggestion. Both figures have now been added to our study's supplement, now appearing as Supplementary figures 6 and 9. The figures have also been inserted below for convenience.

“Supplementary Fig. 6 – Edit distance of shotgun sequenced reads mapping against members of the *Y. pseudotuberculosis* complex, *Y. pestis*, *Y. pseudotuberculosis* and *Y. similis*, in three individuals showing metagenomic signatures of ancient *Y. pestis* DNA. Reads mapping against *Y. pestis* and *Y. pseudotuberculosis* are showing overall declining edit distances. The majority of reads of lowest edit distance are found to be mapping against *Y. pestis* and, therefore, individuals BSK001, BSK003 and BSK007 were considered as putatively-positive and were subjected to whole-genome enrichment. All panels were created using the `ggplot2`⁵⁴ package on R version 3.6.1⁵⁵.”

“Supplementary Fig. 9 - Temporal signal regression analysis of root-to-tip distance against specimen age. All panels were created using the `ggplot2`⁵⁴ package on R version 3.6.1⁵⁵”

Line 133: Given the detection rate what microbes were identified in the *Y. pestis* negative individuals? Were any other potentially pathogenic/epidemic agents present which could have been relevant to the epidemiology of the site?

We thank Referee #3 for this suggestion. We now provide a table, appearing in our study as Supplementary table 6, where we indicate our highest confidence metagenomic identifications made through HOPS. These identifications are deduced from a list of targeted taxonomic nodes belonging to human-associated pathogens (see HOPS documentation for the complete list of target nodes:

https://github.com/rhuebler/HOPS/blob/external/Resources/default_list.txt). The pathogen hits shown here (number of assigned reads indicated in each cell) are those that show potential signatures of post-mortem damage (consistent with aDNA) and have passed the edit distance filter (third criterion within hierarchical post-processing step⁴³).

As can be seen in the table below, apart from *Y. pestis*, we have not identified other epidemic pathogens within our dataset. The majority of identifications are of non-obligate pathogens often associated with the oral microbiota^{56,57}. Moreover, a number of the identified species have been shown to be prone to false-positive identification, which is often a result of their (1) close genetic identity to environmental taxa and (2) the over-representation of pathogens genomes in public databases (the issues associated with problematic identification in ancient metagenomic datasets have been reviewed in detail elsewhere⁵⁷). Here, the depicted results were deduced from shotgun sequenced datasets of ~4-17 million reads per sample. Although beyond the scope of this study, whole-genome enrichment a deeper shotgun sequencing would be optimal for a more secure assessment of these identifications.

Supplementary table 6: Taxonomic nodes targeted through HOPS, which show indications of reads with potential aDNA-associated damage and a declining edit distance.

Taxonomic Node / Sample*	BSK001.A0101	BSK002.A0101	BSK003.A0101	BSK006.A0101	BSK007.A0101
Yersinia pseudotuberculosis complex	265		199		59
Yersinia pestis	92				
Treponema denticola	1,617	38			
Tannerella forsythia 92A2	503				
Tannerella forsythia	5,076				
Streptococcus pyogenes			218		
Streptococcus mutans		63			
Streptococcus gordonii str. Challis substr. CH1		940			
Streptococcus			49		
Porphyromonas gingivalis	220				
Neisseria meningitidis		2,487			
Neisseria		2,012			
Fusobacterium	35	149			
Clostridium botulinum				12,945	
Clostridium				3,780	

*Samples BSK004 and BSK005 that are not shown are those where no pathogen-associated identification were made.

G) References: appropriate credit to previous work?

Prior work has been referenced. This necessarily includes heavy referencing of the authors own work given their clear track record in this area.

H) Clarity and context: lucidity of abstract/summary, appropriateness of abstract, introduction and conclusions

The paper is generally very well written and enjoyable to read. There is a lack of details in places but this may be due to the word count restrictions.

The abstract is appropriate though should be more explicit about the number of genomes which were actually used for analysis (2 human and 2 *Y. pestis*).

Given journal length restrictions and a number of comments by the other two reviewers, the human genomic analysis was moved to the supplement of our paper and, therefore, we do not mention these results in the abstract. We have, however, modified the pathogen genomic part to explicitly state the retrieval of two ancient *Y. pestis* genomes.

Additional minor comments

Supplementary table 1: typo in table legend

The typo has been corrected.

Why were these three of seven individuals selected for human aDNA genotyping?

All seven individuals used for this study were also used for human genomic analysis. Unfortunately, only four of those (BSK001, BSK002, BSK003 and BSK005) had sufficient SNP yields for population genetic analysis. We have now moved the vast proportion of the human genomic analysis to our paper's supplement. We have further expanded this section and made a number of analytical details more clear (Supplementary Information 3).

Line 55: "Despite intense multidisciplinary research on this topic", however the studies commented on are almost exclusively based on aDNA data. While these studies may be interdisciplinary in nature it would be appropriate to acknowledge contributions to this topic from other disciplines.

We thank Referee #3 for this important comment. We have now modified this entire introductory paragraph to reflect a more balanced presentation of contributions with regard to plague historiography, ecology and genetics. The modified paragraph can be found in L. 55-67 and is inserted below for convenience.

*"Despite intense multidisciplinary research on this topic, the geographic source of the Second Plague Pandemic remains elusive. Hypotheses based on historical records and modern genomic data have put forward a number of putative source locations ranging from western Eurasia to eastern Asia (Supplementary Information 1). In recent years, comparisons between ancient and modern *Y. pestis* genomes have shown the Black Death to be associated with a star-like emergence of four major lineages (Branches 1, 2, 3 and 4)^{16,17}, whose descendants are currently dispersed among rodent foci in Eurasia, Africa and the Americas. Although extant lineages that diverged prior to this event are presently identified in central and eastern Eurasia^{16,18,19}, complementary ancient DNA (aDNA) data from such regions are lacking. To date, analyses of the historical record and aDNA data have largely focused on the pandemic's progression within western Eurasia^{12,17,20,21}. While efforts to expand historical investigations and to provide a wider spatiotemporal perspective are currently under way^{9,11,22-26}, the prevailing Eurocentric focus has hampered an identification of the Second Pandemic's origins."*

Line 60: ref 25 is a phylogenetic study rather than based on historical accounts as the preceding sentence suggests.

We had two references cited after this statement, one of which is a genomic study showing the west-ward dispersal of plague from the Volga region into other regions of Europe during the Black Death and the second described the same dispersal pattern from the Crimea

region based on historical records. Therefore, we believe that our statement is balanced in this case.

Historical study:

Barker, H. Laying the Corpses to Rest: Grain, Embargoes, and *Yersinia pestis* in the Black Sea, 1346–48. *Speculum* 96, 97-126 (2021).

Genomic study:

Spyrou, M. A. *et al.* Phylogeography of the second plague pandemic revealed through analysis of historical *Yersinia pestis* genomes. *Nature communications* 10, 1-13 (2019).

Line 162: what is the sampling age of LAI009 from Volga (if available please state). Is one SNP difference consistent with expectation given the temporal window and estimates of the *Y. pestis* mutation rate (number of mutations per year assuming a strict clock)? Similarly for the Black Death strains which supposedly post-date by at least seven years? Assuming seven years, two SNPs suggests a very slow mutation rate?

Through our molecular dating analysis, we have estimated a mutation rate of $1.09E-4$ substitution/site/year using a dataset of 1,405 SNPs. When using this estimate to calculate the mutation rate across the entire genome we calculate a rate of $3.29E-8$ substitution/site/year, which is equivalent to 1 mutation every 6.5 years for the *Y. pestis* chromosome. This retrieved rate is in accordance with previous estimates^{12,58} and is consistent with the notion of *Y. pestis* being a slow-evolving monomorphic pathogen⁵⁹, where episodes of mutation rate acceleration have been associated with epidemic eruptions¹⁹.

Unfortunately, we do not have a precise date for the LAI009 genome from the Volga region. The investigated “Laishevo III cemetery” was affected by erosion and, therefore, the majority of burial artefacts could not be accessed and studied¹². Based on limited archaeological context information, specifically some ceramic findings and a bronze earring, the LAI009 specimen is placed within the 14th century, during the Golden Horde period of Volga Bulgaria¹². Based on the genome’s phylogenetic placement in an intermediate position between BSK001/003 and Black Death-related genomes from southern, central and western Europe we expect an associated date between 1338 and 1353 CE. Below we enclose some relevant historical information, which supports a possible 1345-1346 date for the LAI009 genome based on other documented outbreaks across the Volga region during the mid-14th-century.

During the Mongol period, the Volga region was characterised by a multitude of urban and quasi-urban settlements, all tightly integrated into both regional and long-distance international trade routes. Currently, it is possible to estimate that there were over 50 such settlements in that region (from the Volga-Caspian confluence in the south and the Volga-Kama confluence in the north), all easily navigable by river boats. Laishevo was situated at the northern edge of the Volga basin, on the Volga-Kama confluence. By November 1345, the plague has reached the south-eastern limits of the Golden Horde, with an outbreak reported in Ürgench^{60,61}. According to Rafaino Caresini (c.1314-90) and Lorenzo de Monacis (c.1351-1428), both Venetian diplomats and historians, the plague was in the ‘Tartar lands’ in 1345, with the latter designating these as ‘Scythia’, namely the Pontic-Caspian steppe between the Black and Caspian Seas^{62,63}. Given that medieval Venetian year ran from 1 March to 27/28 February, the reference to ‘1345’ implies either late 1345, or early 1346. Several Russian chronicles report plague in Volga cities of *Astrakhan*, *Sarai* (most likely, New Sarai) and *Bezdesch* (=Beldjamen) in the year 6854 (sometime between 1 September 1345 and 31 August 1346, according to the Byzantine and Russian Orthodox reckoning)⁶⁴⁻⁶⁷. The chronicles also mention the city of *Ornach/Arnach*, whose location cannot be certainly established, but some historians identified it with Ürgench, some others with Tana, and some with Saryklych (later, Sarov) in the Nizhny Novgorod region^{68,69}. Saryklych is situated

400 km south-west of Laishevo, while Beljamen and New Sarai are located, respectively, 770 and 800 km to the south. Given the dense nature of Volga urban network and the fast speed of river travel reported for that time, it is likely that the plague came to Laishevo in the same year that it was reported in other Volga towns (1345-6), possibly in the summer of 1346.

Please add relevant accessions to genomes considered in supplementary Table S9 to support future reanalysis with this dataset.

We thank Referee #3 for this important comment. We have now updated this table with all genome accessions (now appearing as Supplementary Table 13).

Line 452: Does November 2017 refer to the date of the custom download? How many genomes are considered, and perhaps most relevantly for this case how many distinct *Yersinia* species?

We thank Referee #3 for this comment. Indeed, November 2017 refers to the date of download of our custom RefSeq full-genome database. As mentioned in the Methods section of our paper the database contains 15,361 entries. We have inserted below all *Yersinia* species included in the database, with the parenthesis indicating the number of genomes included from each species:

Yersinia aldovae (1), *Yersinia aleksiciae* (1), *Yersinia enterocolitica* (16), *Yersinia entomophaga* (1), *Yersinia frederiksenii* (3), *Yersinia intermedia* (1), *Yersinia kristensenii* (2), *Yersinia pestis* (39), *Yersinia phage* (17), *Yersinia pseudotuberculosis* (13), *Yersinia rohdei* (1), *Yersinia ruckeri* (4), *Yersinia similis* (1), *Yersinia* sp. FDAARGOS (1).

We have now indicated this information within the relevant portion of our Methods section.

Figure 4: There was no description of PCR-genotyped samples

Within Supplementary table 21 we have compiled a detailed table describing all data used for Figure 4, including lineage, strain ID, isolation host/vector (with species names), type of data, accession codes, isolation year and geographic coordinates. The table was already available in our study's initial submission (initially Supplementary table 14, now 21), and has now been further updated with >20 additional isolates recently published (Kukleva et al., 2021). Moreover, we have now changed the depiction of PCR-genotyped strains to reflect them as broadly O.ANT (without lineage specification) as we identified a number of inconsistencies between published papers.

Extended Data Fig 2: The manuscript text suggests two human genomes were of sufficient quality for SNP capture and used in the PCA eg. line 133 'both individuals' but four 'BSK' data points are provided in the PCA?

We agree with Referee #3 that this may have not been clearly indicated in our paper. We assessed all seven analysed specimens for human DNA preservation and identified four individuals with sufficient SNP yields that could be used for population genetic analysis. Those were BSK001, BSK002, BSK003 and BSK005, two of which also yielded ancient *Y. pestis* genomes. This has now been further clarified within a dedicated human genomics section (Supplementary Information 3).

References

- 1 Benedictow, O. J. *The Black Death, 1346-1353: The complete history*. (Woodbridge: Boydell & Brewer, 2004).
- 2 Wray, S. K. Reviewed Work: *The Black Death, 1346-1353: The Complete History* by Ole J. Benedictow. *Journal of the History of Medicine and Allied Sciences* **60**, pp. 514-516 (2005).
- 3 Haak, W. *et al.* Massive migration from the steppe was a source for Indo-European languages in Europe. *Nature* **522**, 207-211 (2015).
- 4 Ning, C. *et al.* Ancient genomes from northern China suggest links between subsistence changes and human migration. *Nature communications* **11**, 1-9 (2020).
- 5 de Barros Damgaard, P. *et al.* 137 ancient human genomes from across the Eurasian steppes. *Nature* **557**, 369 (2018).
- 6 Jeong, C. *et al.* A dynamic 6,000-year genetic history of Eurasia's Eastern Steppe. *Cell* **183**, 890-904. e829 (2020).
- 7 Gneecchi-Ruscone, G. A. *et al.* Ancient genomic time transect from the Central Asian Steppe unravels the history of the Scythians. *Science Advances* **7**, eabe4414 (2021).
- 8 Jeong, C. *et al.* Bronze Age population dynamics and the rise of dairy pastoralism on the eastern Eurasian steppe. *Proceedings of the National Academy of Sciences* **115**, E11248-E11255 (2018).
- 9 Kennett, D. J. *et al.* Archaeogenomic evidence reveals prehistoric matrilineal dynasty. *Nature communications* **8**, 1-9 (2017).
- 10 Kuhn, M., Manuel, J., Jakobsson, M. & Günther, T. Estimating genetic kin relationships in prehistoric populations. *PloS one* **13**, e0195491 (2018).
- 11 Namouchi, A. *et al.* Integrative approach using *Yersinia pestis* genomes to revisit the historical landscape of plague during the Medieval Period. *Proc Natl Acad Sci U S A*, 201812865 (2018).
- 12 Spyrou, M. A. *et al.* Phylogeography of the second plague pandemic revealed through analysis of historical *Yersinia pestis* genomes. *Nature communications* **10**, 1-13 (2019).
- 13 Barta, P. & Štolc, S. HBCO correction: its impact on archaeological absolute dating. *Radiocarbon* **49**, 465-472 (2007).
- 14 Pantusov, N. in *Arkheologiya Semirech'ya, 1857-1912 gg.* (ed I.M. Samigulin) Ch. Otnosheniye Starshego Chinovnika Osobykh Poruchenii pri Voyennom Gubernatore Semirechenskoy Oblasti N.N. Pantusova v Imperatorskuyu Arkheologicheskuyu Kommissiyu o Khode Raskopok na Nestorianskom Kladbishche vblizi Pishpeka, pp. 191-192 (Almaty: LEM: 2011).
- 15 Chwolson, D. Syrisch-nesorianische Grabinschriften aus Semirjetschie. *Mémoires de l'Académie Impériale des Sciences de St.-Pétersbourg. VII ser.* **XXXVII:8** (1890).
- 16 Debets, G. F. *Paleoantropologiya SSSR.* (Moscow, 1948).
- 17 Ludevig, Y. *Spisok cherepam kraniologicheskoi kollektzii Muzeya Antropologii i Etnografii imeni Imperatora Petra Velikago.* (St Petersburg, 1904).
- 18 Sagulenko, P., Puller, V. & Neher, R. A. TreeTime: Maximum-likelihood phylodynamic analysis. *Virus evolution* **4**, vex042 (2018).

- 19 Cui, Y. *et al.* Historical variations in mutation rate in an epidemic pathogen, *Yersinia pestis*. *Proceedings of the National Academy of Sciences* **110**, 577-582 (2013).
- 20 Guellil, M. *et al.* A genomic and historical synthesis of plague in 18th century Eurasia. *Proceedings of the National Academy of Sciences* **117**, 28328-28335 (2020).
- 21 Slavin, P. Death by the Lake: Mortality Crisis in Early Fourteenth-Century Central Asia. *Journal of Interdisciplinary History* **50**, 59-90 (2019).
- 22 Green, M. H. The Four Black Deaths. *The American Historical Review* **125**, 1601-1631 (2020).
- 23 Hymes, R. Epilogue: a hypothesis on the East Asian beginnings of the *Yersinia pestis* polytomy. *The Medieval Globe* **1**, 12 (2014).
- 24 Morelli, G. *et al.* *Yersinia pestis* genome sequencing identifies patterns of global phylogenetic diversity. *Nat Genet* **42**, 1140-1143 (2010).
- 25 Norris, J. East or west? The geographic origin of the Black Death. *Bulletin of the History of Medicine* **51**, 1-24 (1977).
- 26 Dols, M. W. *The Black Death in the Middle East*. (Princeton: Princeton University Press, 1979).
- 27 Schmid, B. V. *et al.* Climate-driven introduction of the Black Death and successive plague reintroductions into Europe. *Proc Natl Acad Sci U S A* **112**, 3020-3025 (2015).
- 28 McNeill, W. H. *Plagues and peoples*. (Anchor, New York, 1976).
- 29 Campbell, B. M. *The great transition. Climate, Disease and Society in the Late-Medieval World*. (Cambridge: Cambridge University Press, 2016).
- 30 Fancy, N. & Green, M. H. Plague and the Fall of Baghdad (1258). *Medical history* **65**, 157-177 (2021).
- 31 Darriba, D., Taboada, G. L., Doallo, R. & Posada, D. jModelTest 2: more models, new heuristics and parallel computing. *Nat Methods* **9**, 772-772 (2012).
- 32 Sabin, S. *et al.* A seventeenth-century *Mycobacterium tuberculosis* genome supports a Neolithic emergence of the *Mycobacterium tuberculosis* complex. *Genome biology* **21**, 1-24 (2020).
- 33 Mühlemann, B. *et al.* Ancient hepatitis B viruses from the Bronze Age to the Medieval period. *Nature*, 1 (2018).
- 34 Mühlemann, B. *et al.* Diverse variola virus (smallpox) strains were widespread in northern Europe in the Viking Age. *Science* **369** (2020).
- 35 Rasmussen, S. *et al.* Early Divergent Strains of *Yersinia pestis* in Eurasia 5,000 Years Ago. *Cell* **163**, 571-582 (2015).
- 36 Bos, K. I. *et al.* Pre-Columbian mycobacterial genomes reveal seals as a source of New World human tuberculosis. *Nature* **514**, 494-497 (2014).
- 37 Bos, K. I. *et al.* Paleomicrobiology: diagnosis and evolution of ancient pathogens. *Annu Rev Microbiol* **73**, 639-666 (2019).
- 38 Key, F. M. *et al.* Emergence of human-adapted *Salmonella enterica* is linked to the Neolithization process. *Nature ecology & evolution* **4**, 324-333 (2020).
- 39 Bouckaert, R. *et al.* BEAST 2.5: An advanced software platform for Bayesian evolutionary analysis. *PLoS Comp Biol* **15**, e1006650 (2019).
- 40 Kass, R. E. & Raftery, A. E. Bayes factors. *Journal of the american statistical association* **90**, 773-795 (1995).
- 41 Duchene, S. *et al.* Bayesian evaluation of temporal signal in measurably evolving populations. *Mol Biol Evol* **37**, 3363-3379 (2020).

- 42 Huson, D. H., Mitra, S., Ruscheweyh, H.-J., Weber, N. & Schuster, S. C. Integrative analysis of environmental sequences using MEGAN4. *Genome Res* **21**, 1552-1560 (2011).
- 43 Hübler, R. *et al.* HOPS: Automated detection and authentication of pathogen DNA in archaeological remains. *Genome biology* **20**, 1-13 (2019).
- 44 Chain, P. S. *et al.* Insights into the evolution of *Yersinia pestis* through whole-genome comparison with *Yersinia pseudotuberculosis*. *Proceedings of the National Academy of Sciences* **101**, 13826-13831 (2004).
- 45 Achtman, M. *et al.* *Yersinia pestis*, the cause of plague, is a recently emerged clone of *Yersinia pseudotuberculosis*. *Proceedings of the National Academy of Sciences* **96**, 14043-14048 (1999).
- 46 McNally, A., Thomson, N. R., Reuter, S. & Wren, B. W. 'Add, stir and reduce': *Yersinia* spp. as model bacteria for pathogen evolution. *Nat Rev Microbiol* **14**, 177-190 (2016).
- 47 Reuter, S. *et al.* Parallel independent evolution of pathogenicity within the genus *Yersinia*. *Proceedings of the National Academy of Sciences* **111**, 6768-6773 (2014).
- 48 Willcocks, S. J., Stabler, R. A., Atkins, H. S., Oyston, P. F. & Wren, B. W. High-throughput analysis of *Yersinia pseudotuberculosis* gene essentiality in optimised in vitro conditions, and implications for the speciation of *Yersinia pestis*. *BMC Microbiol* **18**, 1-11 (2018).
- 49 Giffin, K. *et al.* A treponemal genome from an historic plague victim supports a recent emergence of yaws and its presence in 15 th century Europe. *Scientific Reports* **10**, 1-13 (2020).
- 50 Spyrou, M. A. *et al.* Historical *Y. pestis* genomes reveal the European Black Death as the source of ancient and modern plague pandemics. *Cell Host Microbe* **19**, 874-881 (2016).
- 51 Bos, K. I. *et al.* A draft genome of *Yersinia pestis* from victims of the Black Death. *Nature* **478**, 506-510 (2011).
- 52 Schuenemann, V. J. *et al.* Targeted enrichment of ancient pathogens yielding the pPCP1 plasmid of *Yersinia pestis* from victims of the Black Death. *Proceedings of the National Academy of Sciences* **108**, E746-E752 (2011).
- 53 Morozova, I. *et al.* New ancient Eastern European *Yersinia pestis* genomes illuminate the dispersal of plague in Europe. *Philosophical Transactions of the Royal Society B* **375**, 20190569 (2020).
- 54 Wickham, H. *ggplot2: elegant graphics for data analysis*. (Springer, 2016).
- 55 RCoreTeam. R: A language and environment for statistical computing. *R Foundation for Statistical Computing, Vienna, Austria.*, doi:<https://www.R-project.org/> (2019).
- 56 Dewhirst, F. E. *et al.* The human oral microbiome. *J Bacteriol* **192**, 5002-5017 (2010).
- 57 Warinner, C. *et al.* A robust framework for microbial archaeology. *Annual review of genomics and human genetics* **18**, 321-356 (2017).
- 58 Duchêne, S. *et al.* Genome-scale rates of evolutionary change in bacteria. *Microbial Genomics* **2** (2016).
- 59 Achtman, M. Insights from genomic comparisons of genetically monomorphic bacterial pathogens. *Philosophical Transactions of the Royal Society B: Biological Sciences* **367**, 860-867 (2012).
- 60 Barker, H. Laying the Corpses to Rest: Grain, Embargoes, and *Yersinia pestis* in the Black Sea, 1346–48. *Speculum* **96**, 97-126 (2021).

- 61 al-Khavafi., F. in *Fasikhov Svod* p. 73 (Tashkent: FAN, 1980).
- 62 Monaci, L. d. in *Rerum Italicarum* (ed Flaminius Corenlius) Ch. Laurentii de Monacis Veneti Cretae Cancellarii Chronoicon de Rebus Venetiis, col. 313 (Venice, 1758).
- 63 Caresini, R. *Raphayni de Caresinis Cancellarii Venetiarum Chronica aa. 1343–1388*. Vol. vol. 12:2 (Bologna, Nuova Edizione, 1922).
- 64 Khaydarov, T. F. in *Epokha "Chernoï Smerti" v Zolotoi Orde i Prilegayushchikh Regionakh (Konets XIII - Pervaya Polovina XV vv.)* Ch. pp. 191-2, (Kazan, 2018).
- 65 *Troitskaya Letopis'. Rekonstruktsiya Teksta*. (Moscow: Izdatel'stvo Akademii Nauk SSSR, 1950).
- 66 Makarikhin, V. P. a. T., A.I. in *Musul'manskaya Tsvivilizatsiya Volgo-Surskogo Regiona v Epokhu Feodalizma* (ed D.Z. Khairtdinov) pp. 24-31 (2009).
- 67 in *Polnoye Sobraniye Russkikh Letopisei 7.* Ch. Voskresenskaya Letopis' , p. 210 (St Petersburg: Eduard Prats, 1856).
- 68 Benedictow, O. J. *The Complete History of the Black Death*. (Boydell Press, 2021).
- 69 Ostrowski, D. City names of the western steppe at the time of the Mongol invasion. *Bulletin of the School of Oriental and African Studies* **61**, 465-475 (1998).

Reviewer Reports on the First Revision:

Referees' comments:

Referee #1 (Remarks to the Author):

I have read carefully all the answers that the authors provided to my earlier comments and I am pleased to confirm that the manuscript can now be accepted for publication. I congratulate the authors for the depth and clarity of their revisions.

Referee #4 (Remarks to the Author):

While I did not have the opportunity to read the original manuscript submission, I enjoyed reading this revised version. The authors describe the recovery of ancient DNA from 7 Kyrgystani remains dated to the early 14th century and the assembly of a *Y. pestis* genome from the concatenation of data from two individuals apparently infected with identical strains. The reconstructed *Y. pestis* genome, BSK001/003 appears to fall at the branching point of the polytomy for other extant branches of *Y. pestis* diversity and bears the ancestral SNP state for these branches wherever there was data to cover the SNP positions.

I found the discussion of the methodologies and results quite clear with appropriate quality control measures, and I gather by reading the previous reviewers' comments and the authors' rebuttal that this represents a considerable revision and improvement from the original submission. While I appreciate the contribution of a non-European ancient *Y. pestis* strain to the literature and commend the authors' approach to thoroughly describe the historical and anthropological context of the remains with excellent dating resolution, the position of this singular genome may well change with additional ancient data, and I think the assertion that it represents the diversification of all future *Y. pestis* diversity through to the 21st century a bit of a reach.

As it seems the authors' have already made significant revisions to both the text and the methods following the first round of review, I will offer no further suggestions on that front, but note that the text does have a few typographical errors that are easily fixed. For example:

Line 152: more narrow (narrower)

Line 428: again (against)

Table S3: coeficient (coefficient)

Line 488: the phrasing of "shown to exhibited similarity" could be clarified

Line 530: the parenthetical on this line is quite confusing

Author Rebuttals to First Revision:

Point-by-point response to Referees

Nature Manuscript 2021-07-11627B

We are grateful to Referees #1 and #4 for their positive remarks. Below we address all remaining comments through a point-by-point response (in blue). All changes and additions to the main text of our study are also highlighted in blue.

Referees' comments:

Referee #1 (Remarks to the Author):

I have read carefully all the answers that the authors provided to my earlier comments and I am pleased to confirm that the manuscript can now be accepted for publication. I congratulate the authors for the depth and clarity of their revisions.

We thank Referee #1 for the positive evaluation on our revised manuscript.

Referee #4 (Remarks to the Author):

While I did not have the opportunity to read the original manuscript submission, I enjoyed reading this revised version. The authors describe the recovery of ancient DNA from 7 Kyrgystani remains dated to the early 14th century and the assembly of a *Y. pestis* genome from the concatenation of data from two individuals apparently infected with identical strains. The reconstructed *Y. pestis* genome, BSK001/003 appears to fall at the branching point of the polytomy for other extant branches of *Y. pestis* diversity and bears the ancestral SNP state for these branches wherever there was data to cover the SNP positions.

I found the discussion of the methodologies and results quite clear with appropriate quality control measures, and I gather by reading the previous reviewers' comments and the authors' rebuttal that this represents a considerable revision and improvement from the original submission. While I appreciate the contribution of a non-European ancient *Y. pestis* strain to the literature and commend the authors' approach to thoroughly describe the historical and anthropological context of the remains with excellent dating resolution, the position of this singular genome may well change with additional ancient data, and I think the assertion that it represents the diversification of all future *Y. pestis* diversity through to the 21st century a bit of a reach.

We thank Referee #4 for these comments. Our current strategy for identifying and calling variants (SNPs) in modern and ancient *Y. pestis* genomes uses a 3-fold coverage, a 90% support threshold and a minimum genotyping quality of 30. To gain additional resolution in BSK001/003 we have now also checked all sites previously designated as missing data (Ns) and identified that all low-coverage or low-quality calls still support our assertions (see updated Supplementary Table 18).

Nevertheless, we agree with the referee's comment that more genomic data may provide unpredicted resolution and have now edited the concluding sentence of our phylogenetic results section to state: "*At our current resolution, we conclude that BSK001/003 represents the direct progenitor of the Branch 1-4 polytomy*".

As it seems the authors' have already made significant revisions to both the text and the methods following the first round of review, I will offer no further suggestions on that front, but note that the text does have a few typographical errors that are easily fixed. For example:

Line 152: more narrow (narrower)

Line 428: again (against)

Table S3: coefficient (coefficient)

Line 488: the phrasing of "shown to exhibited similarity" could be clarified

Line 530: the parenthetical on this line is quite confusing

We thank Referee #4 for these corrections. We have made all edits to the main text of our paper. For lines 488 and 530, we have changed the phrasing to the following:

"...masking the problematic pPCP1 region between nucleotides 3,000 and 4,200 that was shown to have high similarity to expression vectors used in laboratory reagents."

"Heterozygosity plots were created both prior and after MALT filtering (see section *post-capture Y. pestis data processing* for a detailed description), in order to investigate whether taxonomy-informed filtering could aid the elimination of contaminant sequences in the investigated datasets (Supplementary Fig. 7)."